# Riemannian L-systems: modelling growing forms in curved spaces

Christophe Godin[1] and Frédéric Boudon[2,3]

[1]Laboratoire Reproduction et Développement des Plantes, Univ. Lyon, ENS de Lyon, UCB Lyon1, CNRS, INRAE, Inria, Lyon, France; [2]CIRAD, UMR AGAP Institut, F-34398 Montpellier, France; [3]AGAP Institut, CIRAD, INRAE, Institut Agro, Université de Montpellier, Montpellier, France

## Theories

morphodynamic systems; differential geometry; patterning; tropism; intrinsic shapes.

**Corresponding author:**
Christophe Godin;
Email: christophe.godin@inria.fr

**Associate Editor:**
Dr. Richard Morris

### Abstract

In the past 50 years, the formalism of L-systems has been successfully used and developed to model the growth of filamentous and branching biological forms. These simulations take place in classical 2-D or 3-D Euclidean spaces. However, various biological forms actually grow in curved, non-Euclidean, spaces. This is, for example, the case of vein networks growing within curved leaf blades, of unicellular filaments, such as pollen tubes, growing on curved surfaces to fertilise distant ovules, of teeth patterns growing on folded epithelia of animals, of diffusion of chemical or mechanical signals at the surface of plant or animal tissues, etc. To model these forms growing in curved spaces, we thus extended the formalism of L-systems to non-Euclidean spaces. In a first step, we show that this extension can be carried out by integrating concepts of differential geometry in the notion of turtle geometry. We then illustrate how this extension can be applied to model and program the development of both mathematical and biological forms on curved surfaces embedded in our Euclidean space. We provide various examples applied to plant development. We finally show that this approach can be extended to more abstract spaces, called abstract Riemannian spaces, that are not embedded into any higher-dimensional space, while being intrinsically curved. We suggest that this abstract extension can be used to provide a new approach for effective modelling of growth of branching systems within non-uniform substrates and illustrate this idea on a few conceptual examples.

## 1. Introduction

Morphogenesis is the process by which biological and non-biological forms develop over time. Systems whose state changes in time according to some evolution rules are called *dynamical systems* (see, e.g., Strogatz, 2000). These rules are usually formalised as a differential equation, called the *evolution equation*, that expresses the rate of change of the system's state as a function of the current system's state and time:

$$\frac{d\mathbf{x}(t)}{dt} = F_\lambda(\mathbf{x}(t), \mathbf{z}(t), t), \tag{1}$$

where $\mathbf{x}(t)$ denotes the system's state, usually in $\mathbb{R}^n$, $n$ being the number of degrees of freedom of the system, and containing the positions and velocities of the system's parts, $\mathbf{z}(t)$ denotes any external variable that may affect the dynamics of the system, $F_\lambda$ specifies the rate of variation of the system's state $\mathbf{x}(t)$ ($\lambda$ refers to the model parameters). In practice, this equation is discretised in small time increments $dt$ to compute the system's evolution, and to iteratively update the system's state in time from a known initial state $\mathbf{x}(0)$:

$$\mathbf{x}(t + dt) = \mathbf{x}(t) + F_\lambda(\mathbf{x}(t), \mathbf{z}(t), t) \, dt. \tag{2}$$

In *morphodynamic systems*, the system is a form that evolves over time. Its state $\mathbf{x}(t)$ represents the spatial extent of the form as a mathematical structure, such as a graph, a grid, a parametric or an implicit surface, etc. This structure is usually augmented with additional information, called *fields* (Giavitto & Spicher, 2008), on its parts, representing locally for instance physical or chemical properties of the growing form. Yet, as operators, such as addition or multiplication

by a scalar, are in general not defined on forms, the evolution of morphodynamic systems cannot be directly modelled using equations in the form of Eqs. (1) and (2). To alleviate this difficulty, the evolution of forms can be formalised more generally by defining a global procedure specifying how the state of the whole form changes for a small amount of time $dt$:

$$\mathbf{x}(t+dt) = F_\lambda(\mathbf{x}(t), \mathbf{z}(t), t, dt). \qquad (3)$$

Here, $F_\lambda$ specifies the change that can be operated on the form $\mathbf{x}(t)$ during $dt$, without formalising the notion of form increment. Despite promising efforts (Giavitto & Michel, 2005; Mjolsness, 2010, 2019), there is yet no general theory to define expressions of $F_\lambda$ for general form growth. However, restrictions to specific form structures or dimensions have lead to the development of efficient formalisms to model form development (see Goriely, 2017; Prusinkiewicz & Runions, 2012 for reviews). This is particularly well illustrated by the formalism of L-systems (Lindenmayer, 1968a, 1968b, 1971; Prusinkiewicz & Lindenmayer, 1990), that was introduced to model the development of sequences and branching structures, very common in biology.

L-systems have been used successfully over more than 50 years to model the construction of fractal forms and the growth of plant branching structures, such as plant architectures, inflorescences, vein patterns, root systems, etc. L-systems make it possible to model the evolution of both the structure and the geometry of forms. For this, the state of a growing form, i.e., a sequence or branching structure augmented with field variables, is mapped to a three-dimensional (3-D) shape using turtle geometry (Abelson & diSessa, 1986). Remarkably, the formalism leads to the definition of a computer language that can be naturally used to program the development of forms in a declarative manner. Various computational implementation of this formalism have been proposed (e.g., Boudon et al., 2012; Hemmerling et al., 2008; Prusinkiewicz et al., 2007; Prusinkiewicz & Lindenmayer, 1990), as extensions of different programming languages, and putting emphasis on the development of different aspects of the theory.

Common to all these implementations is the assumption that the geometric interpretation of the forms is carried out in a Euclidean (flat) space as a vast majority of plant development models are conceived in flat spaces. Most models of plant architecture development for instance simulate the growth of a branching structure in 3-D Euclidean space (e.g., Boudon et al., 2020; Godin et al., 2005; Palubicki et al., 2009; Prusinkiewicz et al., 2001). Models of vein development in leaves simulate the growth of a branching vascular structure within 2-D shapes representing a flat leaf blade (e.g., Feugier et al., 2005; Merks et al., 2011; Runions et al., 2017). In plant tissues, models of hormone signaling or water flows usually assume transport laws expressed in 2-D or 3-D Euclidean representations of the tissue (e.g., Cheddadi et al., 2019; Grieneisen et al., 2007; Jönsson et al., 2006; Stoma et al., 2008). Likewise, the simulation of filamentous system growth, such as pollen tubes or root hairs, is carried out essentially within flat embedding substrate spaces (e.g., Dumais, 2021; Fayant et al., 2010).

Yet, a number of these phenomena actually take place in non-flat and so-called *curved* spaces. Climbing plants for instance may growth on tree trunks or grounds that are not flat surfaces. Vein networks can develop in leaf blades that are markedly curved. Pollen tubes grow on the pin-like structures of papillae that are not flat (Riglet et al., 2020). Microtubules polymerise/depolymerise dynamically within the cell cortex that in general is not flat (Allard et al., 2010).

Examples are numerous and occur on a variety of scales. However, the modelling of plant form growth in curved spaces has been up to now only scarcely investigated. At the level of macro-molecules, simulations of microtubule dynamics have been carried out in 3-D cell geometries to study the emerging properties of such a network of microfilaments subjected to local synthesis, decay and interaction rules (Mirabet et al., 2018). In this work, cell geometry is represented by a 3-D mesh, and microtubule trajectories are computed by assuming that microtubules are progressing in straight line in the 3-D Euclidean space. The resulting displacement is then projected the local tangent plane to account for potential curvature of the cell geometry. Likewise, at organ scale, a similar projection strategy is used in Hädrich et al. (2017) to model climbing plants or in Ringham et al. (2021) to model branching venation patterns on the surface of petals. In both cases, to approximate the formation of a pattern in a curved space, the pattern growth is first evaluated in the 3-D Euclidean ambient space. The resulting form is then projected on the discrete curved surface represented as a 3-D triangular mesh. A more direct use of the concepts of differential geometry was described in a different application context to create artistic patterns on the surface of 3-D objects by Li et al. (2010). This approach exploits user-defined vector fields on surface meshes to locally drive drawings of curves at the surface. This is different from the approach that we use here which is based on the possibility to follow geodesics in curved spaces to construct forms. However, similarly to what we do here, the authors construct a language that makes it possible to program patterns based on vector fields. A different approach, aimed at modelling the growth of plant lianas of their support and more generally the growth of parasite organisms on various types of hosts, analyses how to grow surfaces by accretion formalised in Moulton and Goriely (2014) with the constraint of keeping on a reference surface representing the host (Öncül et al., 2020). In this approach, the trajectories of the parasite is defined analytically on the host surface and the focus is on the construction of the surface representing the parasite envelope. In contrast, our approach is primarily focused on the construction of the trajectories representing the patterns growing in curved spaces. In a preliminary work, we explored the possibility of using L-systems to model fractal structures on simple spheres (Pulwicki & Godin, 2017). Here, we largely extend this initial exploration with a complete formalisation of the notion of Riemannian L-systems, that can be applied to both smooth curved surfaces and more abstract non-Euclidean smooth curved spaces.

Our approach couples L-systems and differential geometry. Importantly, this extension remains easy to use for modellers as it allows to program L-system models as if processes were locally taking place in a Euclidean space. To draw geometric patterns, the user mainly thinks in terms of elementary movements, 'go straight', 'turn right', etc. without paying attention most of the time to the curvature of the underlying space. In this way, we show that turtle geometry makes it possible to define a notion of *intrinsic shape*, that does not depend on the embedding space. Specific primitives allow the user to use geometric properties of curved spaces (curvature, excess angle, parallel transport, point-wise geodesic distance, etc.) to develop programs in which the geometry of the embedding space feeds back on the developing form. Through the article, we chose not to assume that the reader is familiar with concepts in differential geometry. We therefore introduce the necessary fundamental concepts and notations used in this domain to keep the text as self-contained as possible. We also provide various applications of how Riemannian L-systems can be used to illustrate key concepts of differential geometry and to explore of a variety of mathematical or

biological dynamical systems in curved spaces, such as the growing fractal forms, random walks, developing branching structures, tip growing filaments, vein pattern generation and so on.

## 2. L-systems overview

L-systems are basically rewriting systems on strings for which rewriting rules are applied in parallel to all the elements of the current string to compute the new string. In this section, we briefly introduce L-systems (Lindenmayer, 1971; Prusinkiewicz & Lindenmayer, 1990) and key associated notions.

### 2.1. Basic formalism

Let $V = \{a_1, a_2, a_3, ..., a_N\}$ be a finite set of elements (called *symbols* or *modules*). We call $V^*$ the set of all finite strings that can be constructed by concatenating any number of symbols from $V$ ($V^*$ is the free monoid constructed from $V$ for the binary operation of string concatenation). $V^*$ includes the empty string, denoted $\lambda$, that is the neutral element for string concatenation. Sequences of $V^*$ are called *words*. For example, if $V = \{A, B, a, b\}$, $x_1 = aaBAa$, and $x_2 = abbAaAabbbb$ are words of $V^*$.

**Definition 1** (String homomorphism). Let us consider a word $x \in V^*$ and a decomposition of this word into subwords: $x = x_1 x_2 \cdots x_K$, where $x_k$ are also words $\in V^*$, for $k = 1 \ldots K$. A *string homomorphism H* from $V^*$ to $V^*$ is a mapping such that:

$$H(x_1 x_2 \ldots x_K) = H(x_1) H(x_2) \ldots H(x_K).$$

From this definition, it follows that a string homomorphism is completely defined by the definition of the images of the symbols in $V$. In addition, this definition implies that $H(\lambda) = \lambda$.

Let us call $w_n$ the word image by $H$ of symbol $a_n \in V$, $H(a_n) = w_n$. In the context of L-systems, the pair $P_n = (a_n, w_n)$ is called a *production rule*. We denote $P = \{P_n\}_{n=1 \cdots N}$ the set of all production rules associated with $H$.

**Definition 2** (D0L-system). A D0L-system $\mathcal{L}$ is a 3-tuple $(V, P, A)$, where $V = \{a_1, a_2, a_3, ..., a_N\}$ is a finite set of symbols, $P$ is a set of production rules on $V$ and $A \in V^*$ is called the *axiom*.

**Definition 3** (Derivation). Let $V = \{a_1, a_2, a_3, \cdots, a_N\}$ be a finite set of symbols, $P$ a set of production rules on $V$ and $H$ the homomorphism associated with $P$. Let $x$ be a word in $V^*$, $H(x)$ is called the derivation of $x$ by $P$. If $x = x_1 x_2 \cdots x_K$ and $H(x_k) = w_k$ for $k = 1 \ldots K$, the derivation of $x$ is denoted by:

$$x_1 x_2 \cdots x_K \Rightarrow w_1 w_2 \cdots w_K.$$

For a D0L-system $\mathcal{L} = (V, P, A_0)$, successive derivations of the axiom $A_0$ can be obtained in a deterministic manner. Let us denote $A_i$ the $i$-th derivation of $A$, *i.e.*, $A_i = H^i(A_0)$, we have:

$$A_0 \Rightarrow A_1 \Rightarrow A_2 \Rightarrow \cdots \Rightarrow A_i.$$

A (D0)L-system may thus represent dynamical systems whose states can be abstracted as (discrete) strings of components, called *L-strings*, that dynamically change as derivations are applied. Derivations are often interpreted as the evolution of the system's state with time. They can also represent changes of the observer's viewpoint (such as zooming in a structure, which shows more elements, and thus result in a change of the system's representation).

**Example.** A simple example of a D0L-system is provided by the development of a multicellular filamentous organism *Anabenae*

(Prusinkiewicz & Lindenmayer, 1990) based on a model originally developed by Koster and Lindenmayer (Koster & Lindenmayer, 1987). This organism is organised as a file of cells that have different types $a, b, A, B$ corresponding to their differentiation and polarisation states. According to their types, cells have different developmental behaviours described by specific rules that can be modelled by an L-system (Prusinkiewicz & Lindenmayer, 1990).

Let $\mathcal{L} = (V, P, A_0)$, where $V = \{A, B, a, b\}$, $P = \{A \rightarrow Ba, B \rightarrow bA, a \rightarrow A, b \rightarrow B\}$ and $A_0 = A$. Starting from the axiom $A_0 = A$, the sequence of derivations of this L-system goes as follows:

$$A \Rightarrow Ba \Rightarrow bAA \Rightarrow BBaBa \Rightarrow bAbAAbAA \Rightarrow \cdots$$

Each string (L-string) of this derivation sequence represents a state of the growing organism at consecutive time steps.

### 2.2. D0L-system extensions

D0L-systems have been extended in various ways and have lead to develop powerful languages constructs to simulate dynamical systems. Here, we briefly recapitulate common key extensions (Prusinkiewicz & Lindenmayer, 1990).

**2.2.1. Branching systems.** First, D0L-systems have been extended to model *branching systems* rewriting and not only strings. This extension relies on the fact that branching systems can be simply encoded as *bracketed strings*. This makes it possible to define readily D0L-systems that rewrite branching systems. For this, the definition of the vocabulary $V$ is augmented by a pair of opening '[' and closing ']' square brackets. In addition, a restriction is imposed on the homomorphism on $V^*$: square brackets can only be mapped to themselves by $H$, and they can only appear on the right-hand side of a production rule if they form a well-formed bracketed string (all opening brackets must be properly balanced in the string and opening/closing brackets must be strictly nested).

In a plant for instance the apex A of a stem can produce a new portion of stem (internode I), a lateral apex A and a new apical apex. This can be represented by the production rule:

$$A \rightarrow I[A]A.$$

Starting from the axiom $A$, the first derivations yield:

$$A \Rightarrow I[A]A \Rightarrow I[I[A]A]I[A]A \Rightarrow \cdots$$

Note that it is usually assumed that every module for which a production rule is not specified (here $I$ for example) is rewritten unchanged in the new string (identity transformation).

**2.2.2. Parametric rules.** A second powerful extension is the possibility to add parameters to the modules. This makes it possible to write rules that propagate parameter values as the modules are rewritten and to make computation on them. For instance, the elongation of a rectangular cell represented by a module $C$ can be modelled by introducing a real parameter $y$ in a production rule such that:

$$C(y) \rightarrow C(y + \delta y),$$

where $\delta y$ is an increment of length. If the axiom consists of the string $C(0)$, then applying the above production rule to the axiom will yield the following $i$ first derivations:

$$C(0) \Rightarrow C(\delta y) \Rightarrow C(2\delta y) \Rightarrow \cdots \Rightarrow C(i\delta y).$$

**2.2.3. Context-sensitive rules.** Another useful extension is the notion of *context-sensitive* rules. Here, a production rule is applied to the

left-hand side module only if the context of this module matches some criterion in the current L-string. Traditionally, the left and right context of a module in a string are specified by using '<' and '>' markers.

For instance, the rule:

$$B < A \rightarrow B,$$

means that a symbol A must be rewritten into B only if its left-context (i.e., the module immediately to its left) in the original L-string is a B. Otherwise, the symbol is left unchanged. The action of this rule can be observed on the axiom $BAA[AA]AA$ that leads to the sequence of derivations:

$$BAA[AA]AAA \Rightarrow BBA[AA]AAA \Rightarrow BBB[AA]AAA$$
$$\Rightarrow BBB[BA]BAA \Rightarrow BBB[BB]BBA \Rightarrow BBB[BB]BBB.$$

This makes it possible for instance to model the propagation of a signal in a branching structure. Such a signal can propagate from the root of the branching structure to the leaves using left-context rules, or from the leaves to the root using right-context rules. One can note that the rule $B < A \Rightarrow B$ applies to L-string patterns, such as $..BA..$ or $..B[A]..$ and thus takes into account the branching organisation of the L-string.

**2.2.4. Declarative rules with procedural statements.** Instead of specifying once for all the right-hand side of production rules, it might be interesting to determine it procedurally at runtime. For example, the procedural production rule:

$$A(x) \rightarrow B \text{ if } x > 0.3 \text{ else } C$$

This rule yields different derivations for different axioms:

$$A(0.5) \Rightarrow B,$$
$$A(0.1) \Rightarrow C.$$

Such rules thus have a classical left hand-side, and a right-hand side that is a procedure ending with rewriting statements (here $\rightarrow B$ and $\rightarrow C$). Procedural rules make it possible for instance to simulate non-deterministic L-systems. In this case, the right-hand side rewriting statement is computed based on some random choice.

All the above extensions can be combined to model complex dynamical behaviours. Note that the resulting mathematical structures are no longer D0L-systems, but extensions of them that will be called hereafter using the generic term of *L-systems*.

Based on the pioneering language called cpfg developed by P. Prusinkiewicz (Prusinkiewicz & Lindenmayer, 1990), a number of language variants have been built over the years on the top of different programming languages, e.g., L+C (C++) (Prusinkiewicz et al., 2007), XL (Java) (Kniemeyer & Kurth, 2007) and L-Py (Python) (Boudon et al., 2012). In this article, the examples and extensions are developed in L-Py.

## 2.3. Turtle geometry: adding geometry in L-systems

L-strings are abstract representations of a system's state with no particular graphic representation. It is however often very useful to attach a geometric representation of the system's state in the 3-D space (to display the 3-D architecture of a modelled growing plant for example). For this, L-systems make use of turtle geometry (Abelson & diSessa, 1986).

**2.3.1. Turtle geometry.** Turtle geometry is a way to define complex geometric objects in 3-D as a sequence of elementary geometric instructions. Basically, a (virtual) turtle is able to move and draw in the 3-D space as it moves. For this, the turtle is given a sequence of elementary geometric instructions that it can read and interpret sequentially. Interestingly, as the definition of forms using turtle geometry relies on purely geometric primitives, the construction process is in general independent the selected coordinate system.

We assume that the turtle moves with respect to a global reference frame, denoted as $\mathcal{R}_0$ and that it is itself represented as a moving local reference frame, hereafter called the *turtle frame* $\mathcal{R} = \{\mathbf{H}, \mathbf{L}, \mathbf{U}\}$ (respectively, denoting unit vectors related to the turtle's body: Head, Left, Up). Each elementary instruction is interpreted by the turtle as an order to move or to draw (Abelson & diSessa, 1986; Prusinkiewicz & Lindenmayer, 1990). The elementary instructions are coded as strings of *geometric modules* having specific names. For example, the module `F(l)` instructs the turtle to move forward by a distance $l$ in the direction $\mathbf{H}$ of its head, and draw a line while moving, the module `+(a)` means that the turtle should turn left (around its Up axis $\mathbf{U}$) by an angle $a$ degrees, the module `;(c)` means that colour $c$ should be used for drawing from now on, etc. An L-system language usually contains a number of predefined geometric modules that make it possible to draw a large variety of simple and more complex geometrical shapes (see detailed list of turtle instructions in L-Py for example).

To operate on an input string, the turtle is associated with a state $S$ that records its current information: position $(x, y, z)$, orientation $\{\mathbf{H}, \mathbf{L}, \mathbf{U}\}$ expressed in the global reference frame $\mathcal{R}_0$, its current colour $c$, etc.,

$$S = (x, y, z, H, L, U, c, \cdots).$$

When reading a new instruction from an input sequence of turtle commands, the turtle executes the elementary action corresponding to the read string module, updates its state accordingly and proceeds to the next instruction in the sequence. Sequences of geometric modules, interpretable by a turtle, are called *T-strings*.

**2.3.2. Branching systems.** As explained above, branching systems can be modelled by using square brackets. During turtle interpretations, when reaching an opening square bracket in an L-string, the turtle saves its current state on the top of a stack, called the *interpretation stack*, and then proceeds with the string interpretation inside the bracket. When reaching a closing square bracket, the turtle pops the current top state of the interpretation stack and restores its current state with it before continuing to interpret the L-string. This push/pop turtle mechanism linked with the use of square brackets makes it possible to easily create branching patterns in L-systems. To insert a branch on another for example, a well-balanced bracketed list of modules representing the new branch must be inserted at the position corresponding to the bifurcation between a main branch and the new lateral branch.

**2.3.3. Interpretation rules: coupling L-strings with turtle geometry.** L-systems strings can be associated with a geometry using turtle geometry. This can be done by either directly integrating symbols that can be interpreted by the turtle in the finite set of symbols $V$ or by using *interpretation rules* to translate L-strings (corresponding to the dynamic system's state) into T-strings (corresponding to its geometrical interpretation).

Remarkably, these geometric interpretation rules can also be represented as a homomorphism $G$ mapping L-strings to T-strings (Kurth, 1994). This homomorphism is usually defined on set of

L-system symbols $V$. Let us call $z_n$ the word image by $G$ of symbol $b_n \in V$, $G(b_n) = z_n$. The pair $I_n = (b_n, z_n)$ is called an *interpretation rule*. We denote $I = \{I_n\}_{n=1\cdots M}$ the set of all the interpretation rules associated with $G$.

The interpretation rules are thus used to make a translation between the L-string modules that in general denote the components of the modelled system with their attributes and have no direct geometric interpretation, and the T-string geometric modules that can be directly interpreted by the turtle as geometrical instructions.

Interpretation rules can be recursive and are applied in a depth-first manner to the input L-string (recursions are fully developed before processing to the next module of the string). Most of the extensions that have been defined for L-system derivation rule above are also defined for interpretation rules (parametric and procedural), except for context sensitivity.

**2.3.4. Sensing the environment.** In L-systems, forms get developed in a 2-D or 3-D Euclidean embedding space. As they grow, it may sometimes be useful to locally probe the environment in a production rule to make a decision that will impact the growth. For example, one may want to determine if a new tentative segment extension of the growing form collides with some object in the environment or with previously constructed part of the same form to decide if the growth will actually take place.

For this, it is essential to access the position and orientation of the turtle in production rules. A specific mechanism has thus been designed in this aim (Prusinkiewicz et al., 1994). It relies on the use of specific query modules, such as ?P, ?H or ?U, that allow the modeller to access the turtle position and orientation at different locations of an L-string from within production rules. To access such information at a given derivation step, these query modules have to be produced in the L-string at the previous iteration step. They act as place-holders to record geometric information computed during turtle's interpretation at the previous step. Once filled, the parameters of these modules can be used in production rules of the next iteration to make development decisions.

### 2.4. L-system examples in L-Py

Let us illustrate the above concepts, on a few examples showing how simple forms can be computed in space or in both space and time

```
1 Axiom: A(1)
2 derivation length: 20
3 production:
4 A(n): nproduce F(n)+(60)A(n+1)
```

**Listing 1.** Archimedean spiral (see Figure 1a).

```
1 dl = 1.
2 Axiom: A(0)
3 derivation length: 1000
4 production:
5 A(n):
6   a = 360*random()
7   nproduce +(a)F(dl)A(n+1)
```

**Listing 2.** Random walk in 2-D (see Figure 1b).

```
1 Axiom: F(1)-(120)F(1)-(120)F(1)
2 derivation length: 5
3 production:
4 F(x) : nproduce F(x/3.0)+(60)F(x/3.0)-(120)F(x/3.0)+(60)F(x/3.0)
```

**Listing 3.** Fractal curve (von Koch flake) (see Figure 1c).

```
1  N = 10
2  offset = 2
3  iangle = 60
4
5  Axiom:  A(0)
6  derivation length: N
7  production:
8  A(n) :
9    if n<N:        # produces an internode, a lateral bud (in [...]) and an apical bud
10     nproduce I(n) [P(n)A(n+offset)] A(n+1)
11   else:          # produces a flower bud
12     nproduce B
13
14 interpretation:
15 I(n) : nproduce ;(1)F(N-n)
16 P(n) :            # Phyllotaxis angle
17   if n%2 == 0 : nproduce  +(iangle)
18   else: nproduce -(iangle)
19 A(n): nproduce ;(2)@O(1.5)
20 B : nproduce F(1);(3)@O(1.5)
```

**Listing 4.** Plant branching structure development (see Figure 1d).

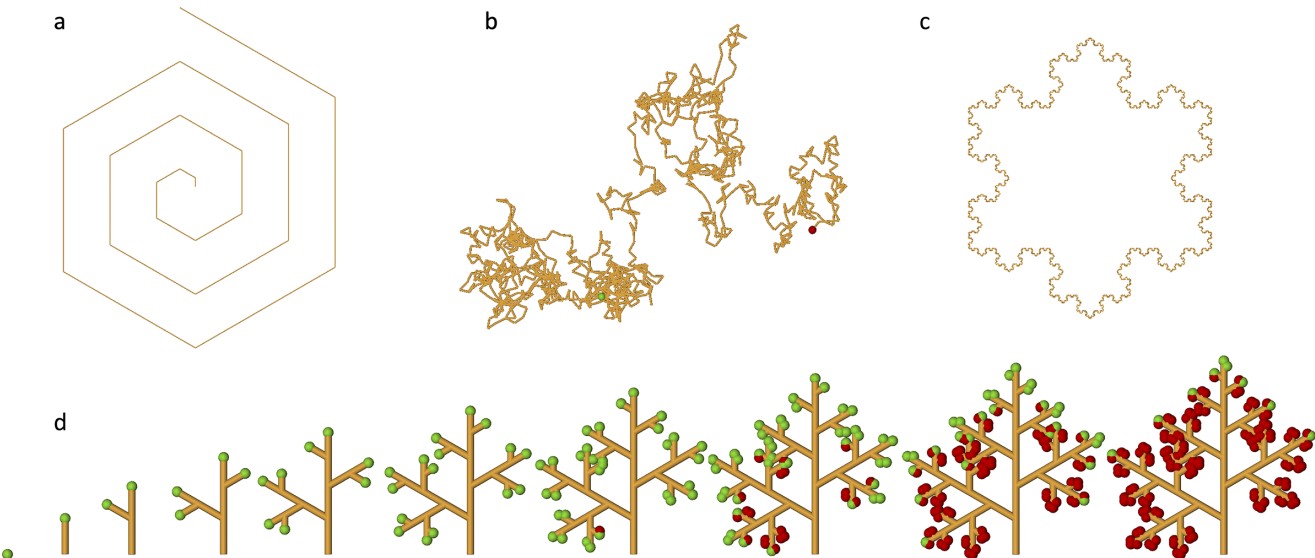

**Figure 1.** Examples of geometric forms that can be produced by L-systems (here using the computer language LPy). (a) A polygonal Archimedean spiral, (b) a random walk, (c) the von Koch flake (fractal curve) and (d) an idealised simple sequence of development of a plant branching system.

with L-systems, Figure 1, together with their L-system code (here using L-Py).

The first example (Listing 1) illustrates the notion of recursive production rule, where a module A containing a variable *n* produces a segment of size *n*, turns of a fixed angle and produces a new module A with variable $n + 1$. When repeated (here 20 times) this produces a polygonal Archimedean spiral, Figure 1a. On the same principle, the second example (Listing 2) shows how stochasticity can be introduced in the rules to generate a random walk for instance Figure 1b. The use of recursivity is at the core of the language. If extended beyond spatial recursivity, it can be used to model a wide variety of shapes. In Figure 1c, for example, recursivity is used over scales to produce a fractal form (Mandelbrot, 1982). Here (Listing 3), at each derivation, existing segments are recursively rewritten as several segments of shorter size. In this procedure, the turtle interpretation of the Lstring converges towards a fractal form (Prusinkiewicz & Lindenmayer, 1990). Finally, the recursive principle can also be used in space and time to simulate the development of a form. Figure 1d illustrates how the development of a plant branching system can be described very concisely using this principle (Listing 4).

Forms built using classical L-systems assume that the underlying 2-D or 3-D space is Euclidean, i.e., that the turtle graphical commands are interpreted as if the turtle were moving in a Euclidean space. Euclidean spaces are flat, i.e., the five Euclid postulates that found classical geometry hold. In particular, the fifth one, called the parallel postulate, states that, in a plane, through a point exterior to a given straight line, at most one line passes that never intersects the initial line. It can be shown that flat spaces are characterised by this fifth postulate, e.g. (Needham, 2021). In the sequel, we extend L-systems to operate in curved (non flat) spaces and provide a programming language to describe form and form development in these more general non-Euclidean spaces.

## 3. Moving on parametric surfaces

In this aim, we start by studying the extension of L-systems to 2-D curved surfaces embedded in a 3-D Euclidean space. For this, we explore how turtle geometry, which defines how forms are constructed in space, can be extended to 2-D surfaces. The extension of turtle geometry to non-parametric, mesh-like surfaces has been described in Abelson and diSessa (1986). Here, we investigate the extension of turtle geometry to parametric surfaces.

### 3.1. Parametric surfaces

A (smooth) parametric surface $\mathcal{S}$ is defined as a differentiable map from a 2-D parameter space $\mathbb{U} \subset \mathbb{R}^2$ to $\mathbb{R}^3$. Let us denote $(u^1, u^2)$ the coordinates of points in $\mathbb{U}$ and $(x^1, x^2, x^3)$ the coordinates of points in $\mathbb{R}^3$. With these notations, a 2-D surface can be defined by the equations:

$$
\begin{aligned}
x^1 &= \phi^1(u^1, u^2) \\
x^2 &= \phi^2(u^1, u^2) \\
x^3 &= \phi^3(u^1, u^2).
\end{aligned}
\tag{4}
$$

These three functions are often summarised by writing more simply $\mathbf{x}(\mathbf{u})$, or in coordinates $x^i(u^\alpha) = \phi^i(u^\alpha)$, $i = 1, 2, 3$, $\alpha = 1, 2$ reminding us that $x^i$'s are functions of the $u^\alpha$'s. If $u^1$ (resp., $u^2$) has a fixed value, the variation of the other parameter $u^2$ (resp., $u^1$) defines a so-called *coordinate line* on the surface, Figure 2a. Any pair of parameters $(u^1, u^2)$ thus defines a point $P$ on the surface. The spatial infinitesimal variations of this point $P(u^1, u^2)$ on the surface with respect to the parameter coordinates define two vectors:

$$
\mathbf{e}_\alpha = \frac{\partial P}{\partial u^\alpha},
\tag{5}
$$

that form the *covariant basis* at point $P$ (also called the *coordinate basis*). These vectors are tangent at point $P$ to the coordinate lines, and form a basis of the plane $T_P\mathcal{S}$ tangent to the surface at point $P$. A vector $\mathbf{X}$ in $T_P\mathcal{S}$ is thus a linear combination of the coordinate basis vectors $\mathbf{e}_\alpha$ at $P$ (Figure 2b), and we note, using Einstein's implicit summation convention:

$$
\mathbf{X} = \sum_\alpha X^\alpha \mathbf{e}_\alpha = X^\alpha \mathbf{e}_\alpha.
\tag{6}
$$

A vector field over the surface associates a vector in $T_P\mathcal{S}$ with each point $P$ of the surface. Each vector can thus be decomposed in the local covariant basis following Eq. (6).

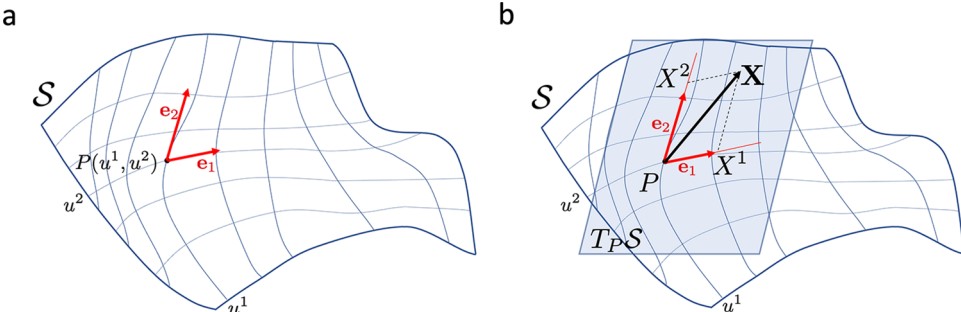

**Figure 2.** Manifold curvilinear coordinates illustrated on a manifold of dimension 2 embedded in a Euclidean pace of dimension 3. (a) Coordinate lines together with the covariant basis at a point $P$. (b) A vector $\mathbf{X}$ in the tangent plane $T_P\mathcal{S}$ at point $P$ can be decomposed in the covariant basis: $\mathbf{X} = X^\alpha \mathbf{e}_\alpha = X^1 \mathbf{e}_1 + X^2 \mathbf{e}_2$.

The embedding Euclidean space, $\mathbb{R}^3$, induces a metric on the surface by assigning a value to the dot product of each pair of vectors in the tangent plane $T_P\mathcal{S}$ at $P$:

$$< \mathbf{X}, \mathbf{Y} >=< X^\alpha \mathbf{e}_\alpha, Y^\beta \mathbf{e}_\beta >= X^\alpha Y^\beta < \mathbf{e}_\alpha, \mathbf{e}_\beta >= X^\alpha Y^\beta g_{\alpha\beta}, \quad (7)$$

where $< .,. >$ denotes the scalar product in $\mathbb{U}^2$ and $g_{\alpha\beta}$ defines the surface metric tensor as the dot product of $\mathbb{R}^3$ vectors:

$$g_{\alpha\beta} =< \mathbf{e}_\alpha, \mathbf{e}_\beta >= \mathbf{e}_\alpha . \mathbf{e}_\beta. \quad (8)$$

The inverse metric tensor $g^{\alpha\beta}$ is such that $g_{\alpha\gamma}g^{\gamma\beta} = \delta_\alpha^\beta$, where $\delta_\alpha^\beta$ is the Kronecker symbol (= 1 if $\alpha = \beta$ and 0 otherwise). At each point $P$ of the surface, we can also define a normal unit vector $\mathbf{n}$, perpendicular to all vectors in the tangent plane $T_P\mathcal{S}$:

$$\mathbf{n} = \frac{\mathbf{e}_1 \times \mathbf{e}_2}{|\mathbf{e}_1 \times \mathbf{e}_2|}. \quad (9)$$

We assume in this article that the surface is orientable, meaning that the normal vector at each point can be defined in a unique and consistent manner throughout the surface (this discards non-orientable surfaces such as the Moebius strip from our analysis). The orientation is made so that the three vectors $(\mathbf{e}_1, \mathbf{e}_2, \mathbf{n})$ have a direct orientation, i.e., $\det(\mathbf{e}_1, \mathbf{e}_2, \mathbf{n}) > 0$).

Surfaces are in general curved. Let us briefly recall what does this mean. Consider a point $P$ on a surface $\mathcal{S}$ and the normal vector $\mathbf{n}$ to the surface at $P$, Figure 3a,b. Let us consider a direction $\mathbf{t}_\theta$ at $P$ making an angle $\theta$ with a fixed arbitrary direction $\mathbf{t}_0$ in the tangent plane at $P$, and construct the 'vertical' plane passing by $\mathbf{t}_\theta$ and $\mathbf{n}$ at $P$ (in grey on Figure 3a,b). The intersection of this 'vertical' plane and the surface is a curve $\gamma_\theta$. We assume that a point moves at constant speed on this curve parameterised by $s$. The velocity of this point is colinear with $\mathbf{t}_\theta$ at $P$ and its rate of variation, defining the curve curvature at $P$, is thus a vector perpendicular to $\mathbf{t}_\theta$:

$$\frac{d\mathbf{t}}{ds} = \mathbf{k}_\theta. \quad (10)$$

This curvature vector lies in the 'vertical' plane which contains the curve $\gamma_\theta$. When the direction $\mathbf{t}_\theta$ is rotated around the vertical axis, the norm $k_\theta$ of $\mathbf{k}_\theta$ varies continuously. It can be shown that it passes by a minimum and a maximum value, called principal curvatures and denoted $k_{min}$ and $k_{max}$, in specific directions, $\mathbf{d}_{min}$ and $\mathbf{d}_{max}$ called the *principal curvature directions* at $P$. As the surface is oriented, the principal curvatures $k_{min}$ and $k_{max}$ may be either positive (curvature vector oriented like $\mathbf{n}$) or negative (curvature vector oriented in the other direction). Principal curvature direc-

tions have the property to always be perpendicular, e.g. (Gray, 1997).

From the principal curvatures at $P$, one can define the mean curvature $\kappa_M$ and the Gaussian curvature $\kappa_G$ as follows:

$$\kappa_M = \frac{1}{2}(k_{min} + k_{max}),$$
$$\kappa_G = k_{min}\, k_{max}. \quad (11)$$

The curved geometry of surfaces is characterised by their Gaussian and mean curvature. The Gaussian curvature characterises the type of local shape of the surface. If $\kappa_G > 0$ the surface locally bends in two identical ways along the principal directions at $P$ and the geometry is that of a dome, Figure 3a. If $\kappa_G < 0$, the surface bends in opposite ways along the principal directions. The surface locally looks like a saddle, Figure 3b. If $\kappa_G = 0$ (one of its principal curvatures at least is 0) the surface is locally looks like a piece of cylindrical surface. Remarkably, $\kappa_G$ only depends on the metric (and not on the embedding of the surface in the ambient Euclidean space): it is an *intrinsic property*, e.g. (do Carmo, 1980).

The mean curvature has also an important geometric interpretation. Contrary to the Gaussian curvature, it depends on the embedding of the surface in $\mathbb{R}^3$ (it is not an intrinsic concept). It expresses how much a piece of surface would deform locally if stretched to deform along the field of local normals, Figure 3c: a small area would be deformed by a factor $2\kappa_M$ in the first order (Struik, 1988, p. 183). The higher is the absolute value of the mean curvature locally, the higher is the surface deformation at this position (a contraction or a stretching depending on the sign of $\kappa_M$ and the orientation of the surface normal), $\kappa_M = 0$ meaning no deformation. Surfaces with constant minimal mean curvature for instance arise in various physical systems, such as soap bubbles or liquid droplets.

### 3.2. Turtle state on a curved surface

On parametric surfaces, positions and directions can be thus specified directly by defining positions and directions in the 2-D parameter space. For instance, a position can be defined by providing a pair of coordinates $P = (u^1, u^2)$ in the parametric space $\mathbb{U}$, while a direction at this point may be defined by providing a vector $\mathbf{V}$ of coordinates $V^\alpha$ in the local covariant basis, i.e., $\mathbf{V} = V^1 \mathbf{e}_1 + V^2 \mathbf{e}_2$. In $\mathbb{R}^3$, the corresponding point coordinates are:

$$\phi(u^1, u^2) = (x^1(u^1, u^2), x^2(u^1, u^2), x^3(u^1, u^2)), \quad (12)$$

and the direction vector has coordinates in $\mathbb{R}^3$:

$$v^i = J_\alpha^i V^\alpha, \quad (13)$$

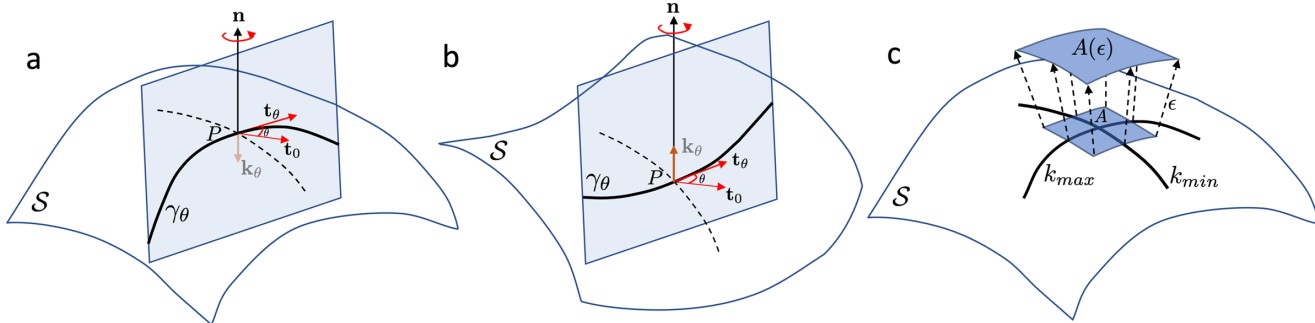

**Figure 3.** Principal curvatures on a surface. When the vertical planes (in grey) are rotated around the normal axis $\mathbf{n}$ (red arrow) with varying angles $\theta$, the plane intersects the surface at curves $\gamma_\theta$. The curvature vector $\mathbf{k}_\theta$ of these curves lies in the plane made by the normal and the tangent vector, and is perpendicular to the tangent vector $\mathbf{t}_\theta$. During rotation, the curvature intensity of these intersection curves passes by a minimum and maximum value, which define the principal curvatures. The corresponding directions are called the principal curvature directions (indicated here by the black and dashed curves) and are always perpendicular. (a) Surface with local positive Gaussian curvature: the principal curvatures have the same sign. (b) Surface with local negative Gaussian curvature: the principal curvatures have opposite signs. (c) Effect of the mean curvature on deforming a local surface element $A$ along the local normal directions over a distance $\epsilon$: $A(\epsilon) = A(1 + 2\epsilon\kappa_M)$ assuming $\kappa_M = (k_{min} + k_{max})/2 > 0$ here.

where $J_\alpha^i$ are the components of the Jacobian of $\phi$, $\mathbf{J}$:

$$J_\alpha^i = \frac{\partial x^i}{\partial u^\alpha}, \tag{14}$$

that represent the best local linear approximation of $\phi$ in the neighbourhood of point $P$. This is called *pushforward* operator, and often denoted $d\phi$. It is represented as a (3,2) matrix, made of the components of vectors $\mathbf{e}_1$ and $\mathbf{e}_2$ arranged in two columns, and maps vectors in the parameter space into corresponding vectors in the 3-D space.

Because $\phi$ defines a bijective differentiable correspondence between the points of $\mathbb{U}$ and those of the surface $\mathcal{S}$ that preserves point neibourhoods, $x^i$ and $u^\alpha$ are often considered as the coordinates of the surface point $P$ expressed in either $\mathbb{R}^3$ or $\mathbb{U}$, respectively. Similarly, vectors, such as $\mathbf{V}$ and $\mathbf{u}$, are mapped by the pushforward operator $d\phi$, and can be interpreted as the 'same vector' with coordinates expressed in two different spaces (e.g., Carroll, 2014, p. 424).

Therefore, the position and orientation of a turtle can be unambiguously defined on the surface by specifying their coordinates in the parametric space:

$$\begin{aligned} P &= (u^1, u^2) \\ \mathbf{t} &= t^1 \mathbf{e}_1 + t^2 \mathbf{e}_2. \end{aligned} \tag{15}$$

Together with the normal vector, the covariant basis forms a local direct basis of the 3-D space at point $P$, $(\mathbf{e}_1, \mathbf{e}_2, \mathbf{n})$, with basis vectors in the tangent plane or normal to it. As a consequence, one can compute a unique $\{\mathbf{H}, \mathbf{L}, \mathbf{U}\}$ in $\mathbb{R}^3$ on the curved surface. Assuming the turtle is at a position $u^\alpha$ on the surface, and points in the direction of vector $t^\alpha$ in the local tangent plane, the turtle's head, $\mathbf{H}$, is oriented in the direction of the tangent vector and is thus aligned along $\mathbf{t}$. The up direction is chosen to be oriented along the surface's normal $\mathbf{n}$ and the turtle's left arm points in the direction perpendicular to both $\mathbf{U}$ and $\mathbf{H}$, Figure 4:

$$\mathbf{H} = \frac{\mathbf{t}}{|\mathbf{t}|}, \qquad \mathbf{U} = \mathbf{n}, \qquad \mathbf{L} = \mathbf{U} \times \mathbf{H} \tag{16}$$

This makes it possible to redefine the turtle state in a curved space. In Euclidean space, this state was defined as

$$S = (x, y, z, H, L, U, \cdots, colour, \cdots). \tag{17}$$

In a curved surface, the turtle's state now becomes:

$$S = (\phi, u^1, u^2, t^1, t^2, \cdots, colour, \cdots), \tag{18}$$

where $\phi$ is the mapping $\mathbb{U} \to \mathbb{R}^3$, $x^i(u^\alpha)$, defining the surface, $(u^1, u^2)$ and $(t^1, t^2)$ are, respectively, the position and orientation of the turtle in the parameter space.

**3.2.1. SetSpace primitive.** To construct a language based on the formalism of Riemannian L-systems we need to associate language constructs with the main concepts introduced above. For instance, to define the space within which the L-system will operate and the turtle will move, the possibility to define parametric spaces must be available in the language. The language primitive SetSpace will allow us to select a specific parametric shape in a library of parametric surfaces provided by the language and on which the movements of the turtle will take place. This library contains standard geometric forms, such as spheres, torus or ellipsoids as well as more generic shapes, such as surfaces of revolution, sweeps and NURBS patches that make it possible to define more complex shapes.

```
1  R = 2.
2  Axiom: SetSpace(Sphere(R))
```

Once the parametric space is set, the turtle state takes the form indicated by Eq. (18). Then, the turtle states are manipulated exactly as in the case of classical Euclidean turtles, including the stacking and unstacking of turtle states when reading L-strings, that push in and pop out states.

### 3.3. Moving straight in a curved space

Moving straight in a curved space means moving along *geodesics*. The definition of geodesics relies on the notion of parallel transport, that specifies what it means for vectors of a vector field to keep parallel as one moves along a curve in the curved space.

**3.3.1. Covariant derivative.** To analyze the spatial variation of the vectors of a vector field as one moves within a curved space, it is convenient to define a corresponding notion of derivative. For this, the derivative of a vector at a point $P$ in a given direction must not only integrate the change in coordinates of the vector but also the change of the local covariant basis. This leads to define a so-called *covariant derivative* on vector fields.

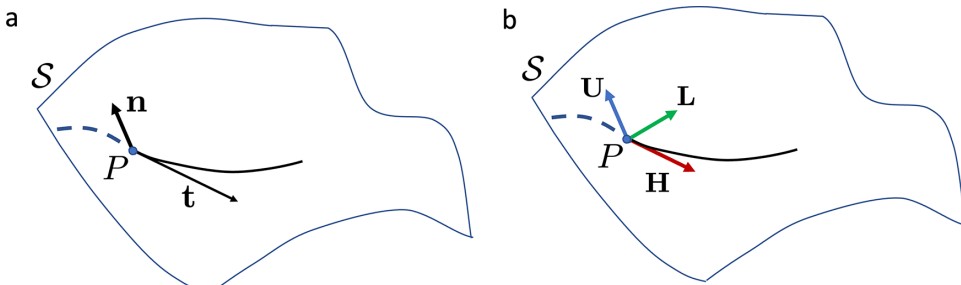

**Figure 4.** Definition of the $\{\mathbf{H}, \mathbf{L}, \mathbf{U}\}$ frame on a curved surface. (a) A direction $\mathbf{t}$ is defined at point $P$. Together with the surface normal $\mathbf{n}$ at $P$, they define a local reference frame that makes it possible to define (b) the turtle's frame: $\mathbf{H}$ is locally defined in the tangent plane aligned with the vector $\mathbf{t}$ while the turtle's upward direction $\mathbf{U}$ is imposed by $\mathbf{n}$ and $\mathbf{L}$ is the direct vector product of $\mathbf{U}$ and $\mathbf{H}$.

To find an expression of covariant derivatives, let us consider the variation of a vector in the tangent plane at a point $P$ along the coordinates lines in the 3-D space:

$$\frac{\partial \mathbf{X}}{\partial u^{\alpha}} = \frac{\partial (X^{\beta} \mathbf{e}_{\beta})}{\partial u^{\alpha}}$$
$$= \frac{\partial X^{\beta}}{\partial u^{\alpha}} \mathbf{e}_{\beta} + X^{\beta} \frac{\partial \mathbf{e}_{\beta}}{\partial u^{\alpha}}. \tag{19}$$

This expression brings out the derivatives of the basis vectors, that can be computed in the ambient space:

$$\frac{\partial \mathbf{e}_{\beta}}{\partial u^{\alpha}} = \Gamma^{\gamma}_{\alpha\beta} \mathbf{e}_{\gamma} + \Lambda_{\alpha\beta} \mathbf{n}, \tag{20}$$

where:

$$\Gamma^{\gamma}_{\alpha\beta} = \frac{\partial \mathbf{e}_{\alpha}}{\partial u^{\beta}} \cdot \mathbf{e}^{\gamma}, \tag{21}$$

are the so-called *Christoffel symbols* (of second kind), $\mathbf{e}^{\beta} = g^{\alpha\beta} \mathbf{e}_{\alpha}$ are the *contravariant basis* vectors. The Christoffel symbols define the rate of change of the covariant basis vectors in each direction of the local contravariant basis. Similarly,

$$\Lambda_{\alpha\beta} = \frac{\partial \mathbf{e}_{\beta}}{\partial u^{\alpha}} \cdot \mathbf{n}, \tag{22}$$

are the coefficients characterising the variation of the covariant basis vectors along the surface normal. Therefore, altogether we have:

$$\frac{\partial \mathbf{X}}{\partial u^{\alpha}} = \left( \frac{\partial X^{\gamma}}{\partial u^{\alpha}} + X^{\beta} \Gamma^{\gamma}_{\alpha\beta} \right) e_{\gamma} + X^{\beta} \Lambda_{\alpha\beta} \mathbf{n}. \tag{23}$$

Let us keep only the part of this expression lying in the tangent plane and define:

$$\nabla_{\alpha} \mathbf{X} = \frac{\partial \mathbf{X}}{\partial u^{\alpha}} - X^{\beta} \Lambda_{\alpha\beta} \mathbf{n}$$
$$= \left( \frac{\partial X^{\gamma}}{\partial u^{\alpha}} + X^{\beta} \Gamma^{\gamma}_{\alpha\beta} \right) e_{\gamma}. \tag{24}$$

This quantity is called the *covariant derivative* of $\mathbf{X}$ on the surface. It corresponds to the orthogonal projection of the usual partial derivative of the vector $\mathbf{X}$ in the Euclidean space onto the local tangent plane, e.g. (Rouvière, 2016, p. 40). It can be shown that $\nabla_{\alpha} \mathbf{X}$ has the properties of a derivative operator (linearity and product rule) and that it defines an intrinsic differential operator on the surface, i.e., an operator that depends only on quantities that can be measured on the surface. Its components are denoted $\nabla_{\alpha} X^{\beta}$:

$$\nabla_{\alpha} \mathbf{X} = (\nabla_{\alpha} X^{\beta}) \mathbf{e}_{\beta}. \tag{25}$$

More generally, the covariant derivative of a vector $\mathbf{X}$ in the direction of an arbitrary vector $\mathbf{Y} = Y^{\alpha} \mathbf{e}_{\alpha}$ is defined as

$$\nabla_{\mathbf{Y}} \mathbf{X} = Y^{\alpha} \nabla_{\alpha} \mathbf{X}. \tag{26}$$

This leads us to the definition of the notion of *parallel transport*. Let $\gamma$ be a curve embedded within the surface defined from an interval $\mathbb{I} \subset \mathbb{R}$ to the surface:

$$\mathbb{I} \to \mathcal{S}$$
$$t \to \gamma(t), \tag{27}$$

and let $\mathbf{X}$ be a vector field on $\mathcal{S}$. $\mathbf{X}$ is parallel transported along the curve $\gamma$ if at each point $\gamma(t)$ of the curve:

$$\nabla_{\dot{\boldsymbol{\gamma}}(t)} \mathbf{X} = 0, \tag{28}$$

where $\dot{\boldsymbol{\gamma}}(t) = \frac{d\gamma(t)}{dt}$ is the tangent to the curve at point $\gamma(t)$. This means that the vector field keeps constant seen from within the tangent planes when one moves in the direction of the curve. More generally, two vectors fields $\mathbf{X}$ and $\mathbf{Y}$ parallel transported along a curve $\gamma(t)$ keep a constant scalar product (i.e., a constant angle in the tangent planes along $\gamma$ as $t$ varies), e.g. (Carroll, 2014).

**3.3.2. Geodesics.** Consider a smooth curve $\gamma$ embedded in a curved space $\mathcal{S}$ (Figure 5a). We assume the curve $\gamma$ is parameterised by the arc-length parameter $s$ using a smooth mapping $\gamma(s) = x^{i}(u^{\alpha}(s))$ from a real interval $\mathbb{I}$ on the curved space $\mathcal{S}$, such that, as $s$ varies, the point $\gamma(s)$ travels at a constant and unit velocity:

$$\mathbf{t} = \frac{dP(s)}{ds} \quad \text{with} \quad |\mathbf{t}| = 1. \tag{29}$$

The tangent vector being of constant norm, its rate of change along the curve,

$$\mathbf{k} = \frac{d\mathbf{t}}{ds}, \tag{30}$$

reflects changes at every point $P$ in direction only and defines the curvature vector $\mathbf{k}$ perpendicular to $\mathbf{t}$ (note that $\mathbf{k}$ is not necessarily in a vertical plane passing through $\mathbf{t}$ and $\mathbf{n}$ at $P$ as was the case with the curvature vector $\mathbf{k}_{\theta}$ in Eq. (10)). The norm of this vector,

$$\kappa = |\mathbf{k}|, \tag{31}$$

defines the curve's *curvature* and indicates the intensity of the curve's bending in the 3-D space at each point. Interestingly, on a curved surface $\mathcal{S}$, the curvature vector $\mathbf{k}$ can be further decomposed locally (Figure 5a). Let $\mathbf{n}$ and $T_{P}\mathcal{S}$ denote, respectively, the normal and the tangent plane to $\mathcal{S}$ at a point $P$ (assuming a local

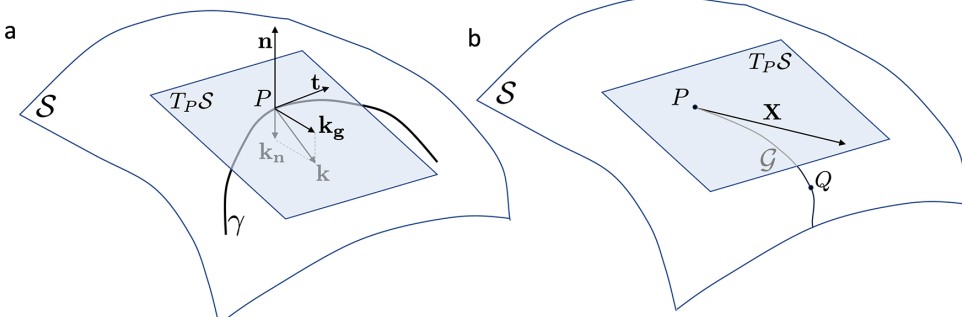

**Figure 5.** Curvature and geodesic. (a) Normal and geodesic curvature of a curve lying on a surface. (b) Let $\mathcal{G}$ be the unique geodesic curve starting at a point $P$ in the direction $\mathbf{X}$. $Q$ is the image on the surface of $\mathbf{X}$ by the exponential map at $P$: $Q = \exp_P(\mathbf{X})$. Reciprocally, $\mathbf{X} = Log_P Q$.

orientation of the surface). Then, the curvature vector $\mathbf{k}$ can be decomposed locally into a normal and a tangent components, $\mathbf{k_n}$ and $\mathbf{k_g}$,

$$\mathbf{k} = \mathbf{k_n} + \mathbf{k_g}. \tag{32}$$

$\mathbf{k_n}$ is the *normal curvature* (rate of change of the tangent vector normal to the surface) and $\mathbf{k_g}$ is the *geodesic curvature*. Geodesic curvature intensity indicates how the curve tends to bend locally in plane, i.e., to deviate from a straight move in the tangent plane. Curves for which the geodesic curvature is null at every point are called *geodesics*. As for geodesics the change of tangent direction (acceleration) is purely normal to the surface, a turtle living on the surface will not notice any lateral movement when moving, and will have the impression to locally move straight. Geodesics are the equivalent in curved spaces to straight lines in flat Euclidean spaces.

An important consequence of the definition is that, in the direction of the tangent vector, the covariant derivative of the tangent vector of a geodesic should have no components on the in-plane local basis vectors (otherwise, there would be a detectable geodesic curvature):

$$\nabla_{\mathbf{t}(s)} \mathbf{t}(s) = \mathbf{k_g} = 0. \tag{33}$$

This equation can be used to define geodesic curves on $\mathcal{S}$. Using the definition of the tangent to the curve $\mathbf{t}$, Eq. (29), expressed in the covariant basis, i.e., $\mathbf{t} = \frac{du^\alpha(s)}{ds} \mathbf{e}_\alpha$ and developing Eq. (33), one obtains a set of two second order, non-linear, coupled differential equations, one for each value of $\alpha$ (e.g., (Carroll, 2014, p. 106)):

$$\frac{d^2 u^\alpha}{ds^2} + \Gamma^\alpha_{\beta\gamma} \frac{du^\beta}{ds} \frac{du^\gamma}{ds} = 0. \tag{34}$$

These equations can be used to compute geodesic trajectories on the surface from different perspectives, depending on the choice of boundary conditions (see below).

**3.3.2.1. Exponential maps.** Smooth surfaces have a remarkable property: At any point of the surface and in a given direction, there exists a unique geodesic that passes through this point and whose tangent at this point points in the given direction, e.g. (Struik, 1988, p. 133). This property makes it possible to define a map between vectors from the tangent plane $T_P\mathcal{S}$ at $P$ and the points of the surface, called the *exponential map*, e.g., (do Carmo, 1980, p. 287). For this, let $P \in \mathcal{S}$ and $\mathbf{X} \in T_P\mathcal{S}$ and denote $\mathcal{G}_P(t, \mathbf{X})$ the unique geodesic originating at $P$ with an initial velocity $\mathbf{X}$ (i.e., $\dot{\mathcal{G}}_P(0, \mathbf{X}) = \mathbf{X}$). Note that on a geodesic parameterised by a parameter $t$, the norm of the

velocity must stay constant all along the curve (e.g., Rouvière, 2016, p. 47). The exponential map is defined by do Carmo (1980, p. 288):

$$\exp_P \mathbf{X} = \mathcal{G}_P(1, \mathbf{X}), \tag{35}$$

i.e., the exponential function returns the point reached after travelling on the geodesic for a time unit, at constant speed $\|\mathbf{X}\|$. Therefore,

$$Q = \exp_P \mathbf{X}, \tag{36}$$

is the unique point on the geodesic $\mathcal{G}_P(t, \mathbf{X})$ at a unit time reach from $P$ when moving at constant velocity $\|\mathbf{X}\|$ in the direction X, Figure 5b. At least in sufficiently small neighbourhoods of $P$, this map is a diffeomorphism (do Carmo, 1980, p. 288). It is thus possible to define a reciprocal map, called the logarithmic map at $P$, such that:

$$\mathbf{X} = \log_P Q, \tag{37}$$

that returns, for any point $Q$ of the surface (in the region where $\exp_P$ is bijective), the direction $\mathbf{X}$ in the plane $T_P\mathcal{S}$ which initiates a geodesic from $P$ that passes through $Q$ and the geodesic distance between $P$ and $Q$ has the norm of $\mathbf{X}$ (e.g., Sommer et al., 2020, pp. 17–30).

Exponential and logarithmic maps are essential tools in Riemannian L-systems as they provide the natural concepts to simulate turtle movements locally, i.e., in the local referential transported with the turtle during the movement.

**3.3.3. Moving forward in a direction: an initial value problem.** Consider a turtle positioned at a point $P$ on a curved surface, and heading in direction $\mathbf{H}$. To advance the turtle by a distance $l$ on the surface, one needs to compute the geodesic starting at $P = (u, v)$ in the direction $\mathbf{H} = (p, q)$ over a length $l$ on the surface, Figure 6a. This computation determines the new position of the turtle, corresponding to $P' = \exp_P(l\mathbf{H})$ and a new value of the head vector $\mathbf{H}$ corresponding to the tangent of the geodesic at the destination point $P'$.

For this, we can integrate (34) over a determined length $l$ to obtain the points of the unique geodesic solving this *initial value problem* (IVP). A classical strategy to solve an IVP system of second-order differential equations similar to Eq. (34), consists of considering $\frac{du}{ds} = p$ and $\frac{dv}{ds} = q$ as two new independent variables and rewrite Eq. (34) as a system of four coupled first-order differential

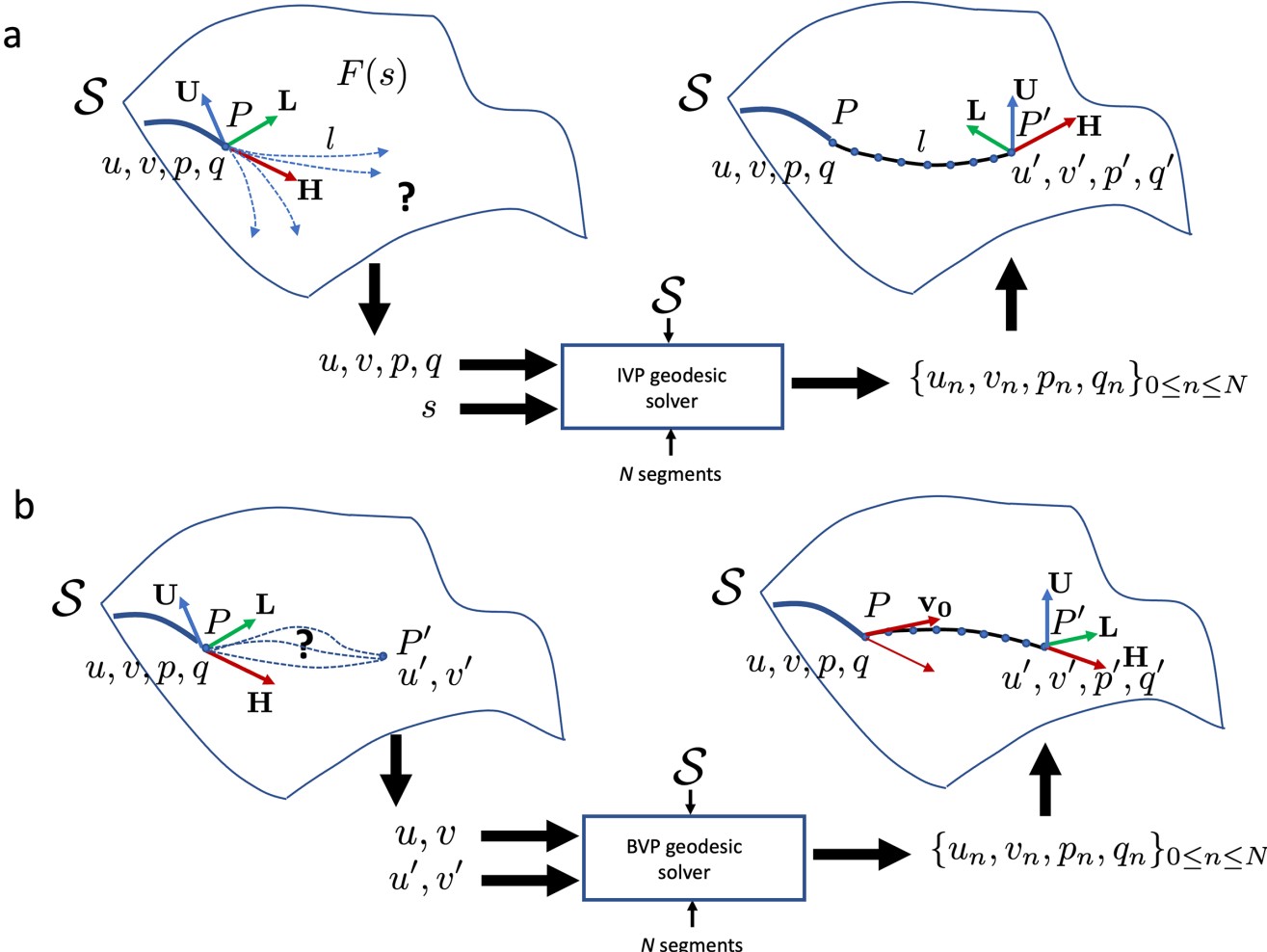

**Figure 6.** Different ways to specify straight displacement in a curved space. (a) Forward algorithm: an IVP. (b) LineTo algorithm: a BVP.

equations (see, e.g., Gray, 1997):

$$\begin{cases} \dfrac{du}{ds} = p \\[2mm] \dfrac{dv}{ds} = q \\[2mm] \dfrac{dp}{ds} + \Gamma_{11}^1 p^2 + 2\Gamma_{12}^1 pq + \Gamma_{22}^1 q^2 = 0 \\[2mm] \dfrac{dq}{ds} + \Gamma_{11}^2 p^2 + 2\Gamma_{12}^2 pq + \Gamma_{22}^2 q^2 = 0, \end{cases} \tag{38}$$

with the initial conditions:

$$P = \begin{bmatrix} u_P \\ v_P \end{bmatrix}, \mathbf{t}_0 = \begin{bmatrix} p_P \\ q_P \end{bmatrix}. \tag{39}$$

The variables of Eq. (34) have been explicitly rewritten with renaming $u^1 \to u$ and $u^2 \to v$ to avoid over indexation in the final Eq. (38). This new system of first-order differential equations can be integrated with classical ODE solvers. Note that for a given parametric surface, the Christoffel symbols appearing in the equation are the only quantities that need be computed. They are functions of $p$ and $q$, which fully couples the equation system. They can be evaluated by using Eq. (21).

In a curved space, a forward step is specified by the primitive $\mathrm{F}(l)$ (move forward at the surface over a distance $l$) as in a Euclidean space, except that the primitive will now make a call to a geodesic solver that takes as an input the surface parametric equation, the current position of the turtle $P = (u,v)$, the heading direction prescribed by the current value of the turtle direction $\mathbf{H}$, both provided by their coordinates in the parametric domain, as well as the number of elementary steps $N$ requested to produce the line, Figure 6a. The solver returns a list of $N$ points $P_n = (u_n, v_n)$, with their corresponding tangents $\mathbf{t}_n = (p_n, q_n)$ on the geodesic that the turtle can use to move forward, and draw a geodesic line on the surface as points of coordinates $\{P_n = x^1(u_n, v_n), x^2(u_n, v_n),, x^3(u_n, v_n)\}_{n=0,...,N}$, where $P_N = P' = \exp_P(l\mathbf{H})$.

Altogether, in the language, all this integration process is hidden from the user and results in very intuitive moves. The user can think of programming a form as if moving in a Euclidean space. After the declaration of the parametric surface to be used, moving forward by a distance $l$ length units takes exactly the same form as moving forward with the turtle in a flat space:

```
1 Axiom:
2   nproduce SetSpace(Sphere(1))
3   nproduce InitTurtle((0,0,1,0)) # Initializes
    turtle position at u=0,v=0,p=1,q=0
4   nproduce F(0.5) # Draws a geodesic of length=0.5
    from point (0,0) in direction (1,0)
```

**3.3.4. Moving forward to a given position on the curved space: a boundary value problem.** To prescribe turtle movements, an alternative method consists of specifying a target point $P'$ that the turtle should reach, from the current position $P$, in the straightest possible way at its next step, Figure 6b. Here, only a target point is prescribed instead of a direction and a length as in the previous IVP case, and one seeks for the geodesics that corresponds to the straightest path between $P$ and $P'$. This can be viewed as a reciprocal problem, corresponding to evaluate the logarithmic map for a given point $P'$, i.e., find the direction $\mathbf{t}$ in $T_P\mathcal{S}$ and the corresponding geodesic that leads to $P'$ from $P$, i.e., such that $\mathbf{t} = \log_P P'$.

Interestingly, the geodesic equation (38) can also be used to solve this problem. However, the problem is now constrained by the endpoints, Figure 6b: given an initial point $P$ and a target point $P'$, find a geodesic that connects these two points. For a smooth surface, there exists at least one geodesic between two points. Finding such a geodesic belongs to the class of boundary value problems (BVPs). There are two main ways to solve these BVP problems, using i) shooting methods or ii) improving progressively an initial solution passing through the two endpoints (see, e.g., Maekawa, 1996).

**3.3.4.1. Shooting method.** The first method can be formalised as an optimisation problem by choosing a shooting direction $\mathbf{t}$ as well as a length $l$, and integrating Eq. (38) over the length $l$ to find a geodesic and a length that would lead exactly to the target point. Let $\gamma_{P,\mathbf{t}}$ be the unique geodesic starting from $P$ in direction $\mathbf{t}$. For at least one specific direction $\mathbf{t}^*$ ($\|\mathbf{t}^*\| = 1$), this geodesic passes through the target point $P'$. Then, for a certain optimal shooting length $l^*$, we get:

$$\exp_P(l^*\mathbf{t}^*) = P'. \tag{40}$$

Then, by shooting in different directions and using different lengths, one can evaluate the quality of each choice $(\mathbf{t}, l)$ by computing a distance $D$ between the point reached and the target $P$. The shooting problem can thus be cast into an optimisation problem:

$$(\mathbf{t}^*, l^*) = \underset{\mathbf{t} \in T_P(\mathcal{S}), l>0}{\operatorname{argmin}} D(\exp_P(l\mathbf{t}), P'). \tag{41}$$

The distance $D(.,.)$ should in principle be the distance on the surface. But as computing this distance would already require the problem to be solved, in practice, we use Euclidean distances in either the parameter space or the embedding space that provide simple computations.

To solve this problem, classical numerical optimisation techniques can be used. Here, we implemented this method using a nonlinear least-squares algorithm with bounds on the variables (Press et al., 2001). The bounds on the variables make it possible to fix the endpoints to the required values, $P$ and $P'$, respectively.

The `RiemannLineToShoot` primitive can be called by passing the target point $(u,v)$ coordinates as an argument:

```
1  Axiom:
2    nproduce SetSpace(Sphere(1))
3    nproduce InitTurtle((0,0,0,1)) # Iniializes
     turtle position at u=0, v=0, p=0, q=1
4    nproduce RiemannLineToShoot((1,0))  # Draw a
     geodesic between points (0,0) and (1,0)
```

**3.3.4.2. Geodesic residuals (GR) method.** Maekawa (1996) proposed an alternative optimisation method to solve this problem on parametric surfaces. Let $\Gamma(P,P')$ be the set of parametric curves with extremities fixed at $P$ and $P'$ and $\gamma$ be a curve in $\Gamma(P,P')$. The idea is to define a quantitative criterion $C(\gamma)$ to assess how much $\gamma$ departs

from a geodesic, ($C(\gamma) = 0$ means the curve is a geodesic) and then to minimise this criterion:

$$\gamma^* = \underset{\gamma \in \Gamma(P,P')}{\operatorname{argmin}} C(\gamma), \tag{42}$$

where $\gamma^*$ is the sought geodesic between endpoint $P$ and $P'$. To define the criterion, let us remark that Eq. (38) can be written in the form:

$$\frac{d\mathbf{Q}}{ds} = G(\mathbf{Q}, s), \tag{43}$$

where $\mathbf{Q}$ is the vector $[u,v,p,q]^T$ corresponding to the concatenation of the position and velocity of a point on the geodesic. In addition, the geodesic curve must respect the boundary conditions:

$$\mathbf{Q}_{initial} = [u = u_P, v = v_P, -, -]^T \quad \text{and} \quad \mathbf{Q}_{final} = [u = u_{P'}, v = v_{P'}, -, -]^T, \tag{44}$$

where $(u_P, v_P)$ and $(u_{P'}, v_{P'})$ are the coordinates of the two endpoints and a dash $-$ means that these values are unconstrained at the boundary. This equation is a first-order differential equation that suggests that recurrence relations exist binding the variables along the curve as $s$ varies. This can be made explicit by discretising the curve into $m - 1$ segments bounded by points of curvilinear abscissa $[s_k, s_{k+1}], k = 0, \ldots, m - 1$. Denoting $\mathbf{Q}_k = \mathbf{Q}(s_k)$ and $\mathbf{G}_k = \mathbf{G}(\mathbf{Q}_k, s_k)$, and using the trapezoidal rule to approximate the derivative, one obtains the following recurrence relations:

$$\frac{\mathbf{Q}_k - \mathbf{Q}_{k-1}}{s_k - s_{k-1}} = \frac{1}{2}(G_k + G_{k-1}), \qquad \forall k = 1, \ldots, m - 1, \tag{45}$$

with the boundary conditions defined in Eq. (44) becoming $\mathbf{Q}_0 = \mathbf{Q}_{initial}$ and $\mathbf{Q}_m = \mathbf{Q}_{final}$. Note that the distance $s_k - s_{k-1}$ between curvilinear abscissa needs to be approximated by a Euclidean distance in $\mathbb{R}^3$. However, provided that the segments are sufficiently small compared to the local curvature, this approximation is in general accurate. From this equations, let us define the residuals:

$$R_k = \frac{\mathbf{Q}_k - \mathbf{Q}_{k-1}}{s_k - s_{k-1}} - \frac{1}{2}(G_k + G_{k-1}), \qquad \forall k = 1, \ldots, m - 1, \tag{46}$$

and

$$R_0 = [u_0 - u_P, v_0 - v_P]^T \quad \text{and} \quad R_m = [u_m - u_{P'}, v_m - v_{P'}]^T. \tag{47}$$

For points falling exactly on a geodesic, these residuals should be 0. The problem can thus be turned into an optimisation problem where we seek for the $\mathbf{Q}_k$ and $s_k$ that cancel the residuals:

$$\begin{aligned} R_k &= [0,0,0,0]^T \qquad k = 1, \ldots, m - 1, \\ R_0 &= R_m = [0,0]^T. \end{aligned} \tag{48}$$

This defines a non-linear system of $4m = 4(m-1) + 2 + 2$ equations for $4m = 4(m-1) + 2 + 2$ variables. Here again, such a system can be solved using classical methods to find the roots of a function. Following Maekawa (1996) who gives an explicit expression for the Jacobian of the residuals, we used a Newton method that remains very efficient if a good initial guess of the solution can be found. In our case, a natural initial guess corresponds to the linear interpolation in the parameter space of the endpoint variables $(u_P, v_P)$ and $(u_{P'}, v_{P'})$.

In Riemannian L-systems, the GR method can be called using a `RiemannLineTo` primitive and passing the target point $(u,v)$ coordinates as an argument. This computes the geodesic between

the current turtle position and the target point argument, and moves the turtle's position at the target point, with a heading direction **H** tangent to the geodesic extremity.

```
1 Axiom:
2   nproduce SetSpace(Sphere(1))
3   nproduce InitTurtle((0,0,0,1)) # Iniiializes
    turtle position at u=0,v=0,p=0,q=1
4   nproduce RiemannLineTo((1,0))  # Draw a geodesic
    between points (0,0) and (1,0)
```

**3.3.5. Geodesic distances on the surface.** Distances can be defined between points of a curved space by using the length of the shortest path between two points. In addition to being the straightest lines between points, geodesic also have the property of being local minima of the length of trajectories between any two points. We can thus define a notion of distance between points of the surface by integrating the length of the different segments composing a geodesic. Let us call $D_\mathcal{S}$ this geodesic-based distance on the surface $\mathcal{S}$ and call $\{(u_k, v_k)\}_{k=0,m}$ the points in the parameter space defining a geodesic between $A$ and $B$, then:

$$D_\mathcal{S}(A,B) = \sum_{k=1}^{m} \|\mathbf{x}(u_k, v_k) - \mathbf{x}(u_{k-1}, v_{k-1})\|, \qquad (49)$$

where $\|.\|$ is the usual $L_2$ norm in $\mathbb{R}^3$. Here, we assume that the distance in $\mathbb{R}^3$ approximates sufficiently well the geodesic length of short segments on the surface. Note that in general the geodesic between two points of the surface is not necessarily unique. In principle, if several geodesic exist one should base the above distance on the one that minimises their lengths.

The distance between two points can be computed by using the function geodesic_distance-to_point, where the turtle state is available (mainly in interpretation rules).

```
1 dist,_,_ = geodesic_distance_to_point(turtle.space,
  (u,v),(ut,vt))
```

### 3.4. Turning on curved surfaces and holonomy

Riemannian spaces are by definition smooth spaces that can be locally approximated by Euclidean spaces. As turning (i.e., rotating) in a Riemannian space is a purely local operation, i.e., an operation that takes place in the tangent plane associated with the current position, it is thus no wonder that turning in such curved spaces is essentially equivalent to turning in Euclidean spaces. However, one must be cautious as rotation transformations must usually be operated in orthonormal bases. This is not the case in our Riemannian spaces as the local basis, e.g., the covariant basis, is not in general orthonormal. Before applying a rotation, one must thus move temporarily to some local orthonormal basis and move back to the original basis afterwards.

The basis vectors $\mathbf{e}'_\alpha$ of the new coordinate system can be related to the old basis vectors $\mathbf{e}_\beta$ by a matrix $\mathbf{G} = \{G^\beta_\alpha\}$ such that, $\mathbf{e}'_\alpha = G^\beta_\alpha \mathbf{e}_\beta$ and $\mathbf{e}'_\alpha . \mathbf{e}'_\beta = \delta_{\alpha\beta}$. Then to rotate in the local tangent plane a vector $\mathbf{t}$ with components $[t^1, t^2]^T$ in the covariant basis by an angle $\theta$, one must first transport the components of $\mathbf{t}$ in a local orthonormal basis, then do the rotation yielding the vector $\mathbf{t}'$ with components expressed in the orthonormal basis, and finally get back to the original, non-orthonormal, basis where $\mathbf{t}'$ has components $[t'^1, t'^2]^T$, Figure 7. If $\mathbf{R}_\theta$ denotes the rotation matrix by an angle $\theta$ in the orthonormal basis $\mathbf{e}'_\alpha$, then:

$$\begin{bmatrix} t'^1 \\ t'^2 \end{bmatrix} = \mathbf{G}\mathbf{R}_\theta\mathbf{G}^{-1}\begin{bmatrix} t^1 \\ t^2 \end{bmatrix}. \qquad (50)$$

Hence for turning by an angle $\theta$, one must first compute locally a matrix $\mathbf{G}$ (here, we used a classical Gram–Schmidt orthogonalisation process; Franklin, 2003), and then rotate turtle's head **H** and left arm **L** according to Eq. (50), while keeping the **U** vector unchanged in the direction of the surface normal. All these operations are hidden in the language and the user only specifies rotations by providing angles (expressed in degrees):

```
1 Axiom:
2   nproduce SetSpace(Sphere(1))
3   nproduce InitTurtle((0,0,0,1))
4   nproduce F(1) +(30) F(1) +(45) F(2)
```

**3.4.1. Parallel transport.** The possibility to turn while moving in a curved space is essential to reveal the curved nature of the space in which the movement takes place (e.g., Abelson & diSessa, 1986; Arnold, 1989, p. 301). To see this, let us use the notion of *parallel transport* introduced above and see how it can be illustrated using the turtle movements. When a vector is parallel transported along a geodesic, its orientation with respect to the geodesic tangent and its norm does not change as one moves along the geodesic.

Consider the classical example of a walker moving on a sphere and transporting a vector from its local tangent plane. The walker starts at the equator in the direction of the North pole and holds her vector pointing ahead (e.g., Abelson & diSessa, 1986, Figure 8a,b red arrow pointing upward at green point). While walking straight, the walker keeps the vector always positioned identically with respect to her. She is parallel-transporting the vector with her. The vector is thus aligned with the tangent of the curve and keeps aligned all the way through. At the pole, the vector is thus horizontal and points to the back. The walker then turns to the left without turning the vector. She then continues her path while still parallel transporting the vector, Figure 8a. As the orientation of the vector with respect to the geodesic tangent does not change along the way, the vector arrives at the equator pointing to the back. The walker then turns again by 90 degrees counter clockwise, still without moving the transported vector orientation, and continues her trip along the equator back to her starting point. When arriving, the transported vector now points west in the initial tangent plane. Still without changing the orientation of her transported vector, the walker makes a final turn by 90 degrees to get back to its exact starting orientation and completely close the loop. One can observe that, although during the trip, orientation of the transported vector never got modified, its final orientation (to the west) does not match its original orientation (to the north).

This property of implicitly rotating parallel transported vectors along closed trajectories is characteristic of curved spaces. It is called *holonomy*. The difference of angle between the vectors before and after the parallel transport along a loop is called the *angle defect*. In flat spaces, the angle defect is 0 for any loop and any parallel-transported vector. In curved spaces, the angle defect is tightly linked with the curvature of the space (e.g., Abelson & diSessa, 1986).

Holonomy and angle defect can be visualised with Riemannian L-system. For this, the turtle keeps track of the accumulated rotation angle, $\alpha$, since an origin point on a turtle path. The origin point is defined using the module ParallelTransportReset that reinitialises the cumulated turtle rotation at any moment ($\alpha = 0$), and thus defines a new origin for parallel transport at the current turtle's position, say $(u_0 = 0, v_0 = 0)$. For a unit vector **v** positioned in the tangent plane at the origin point and making an angle $\beta$ with the turtle's head at the origin, we can thus

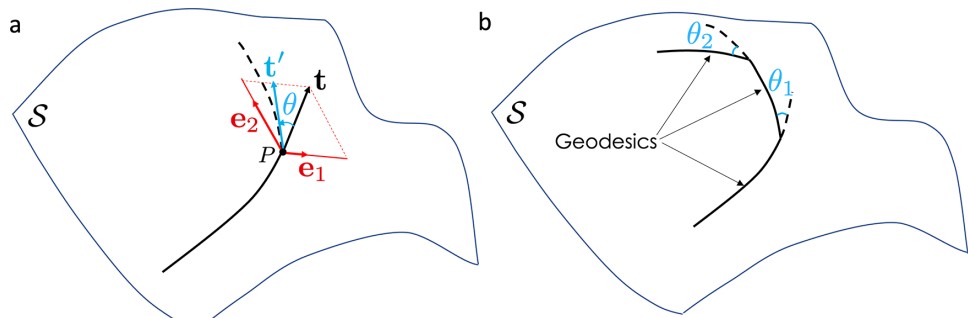

**Figure 7.** Turning on a surface. (a) On a geodesic trajectory (black curve), the turtle is instructed to turn by an angle $\theta$ at a point $P$. As the tangent $\mathbf{t}$ (black) before the turn is expressed in the local covariant basis (red), an orthonormal basis (not shown) must be computed to perform the rotation, leading to a new tangent vector $\mathbf{t}'$ expressed in the orthonormal basis. Then, the new tangent vector is expressed in the covariant basis and moves along geodesics can continue. (b) In this way, the turtle can draw curved polylines on the surface, by alternating geodesic segments (in black) and rotations (angles $\theta$ in blue).

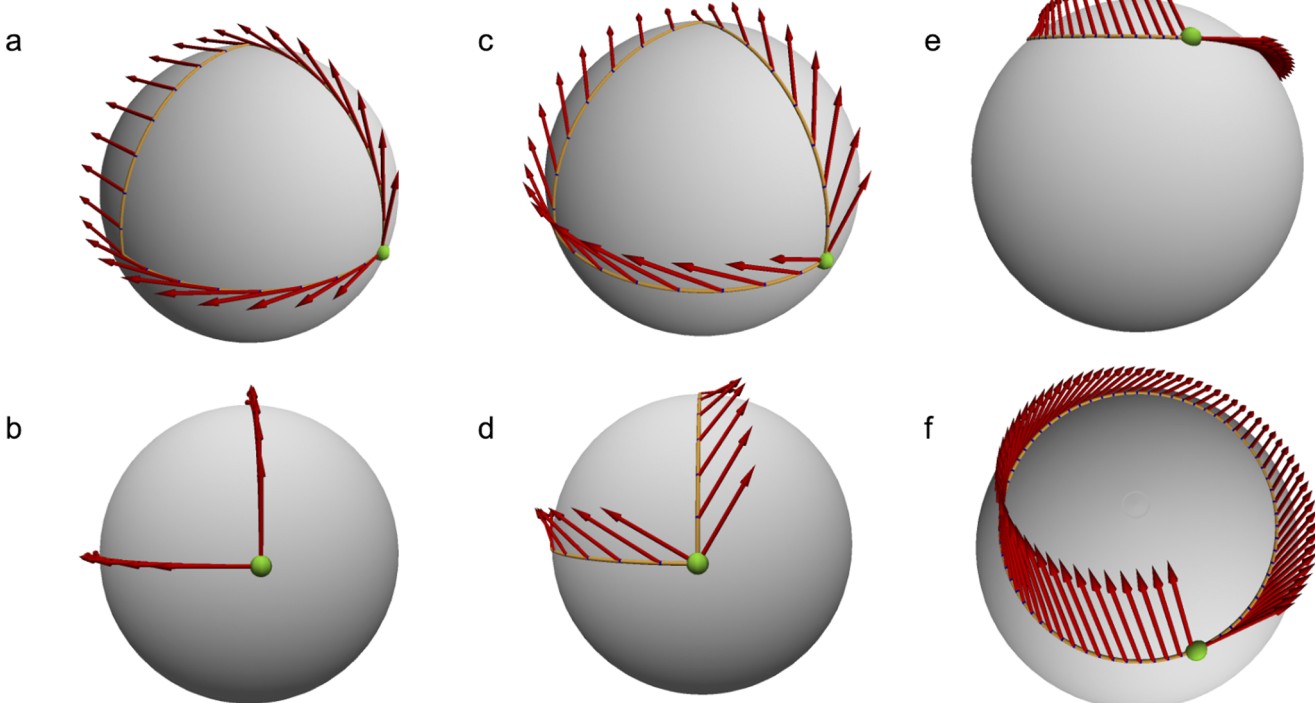

**Figure 8.** Holonomy and parallel transport using Riemannian L-systems. (a) and (b) Parallel transport of a vector along a polygon made of geodesics. The vector is initially tangent to the first geodesic, then is perpendicular to the tangent on the second geodesic, then points backward on the third geodesic segment. (c) and (d) Parallel transport of a vector not tangent to the first geodesic. The vector keeps a constant angle with the tangent vector, but this angle changes each time the turtle turns. (e) and (f) Parallel transport along a curve that is not a geodesic: the angle between the transporting curve and the transported vector varies continuously.

compute and draw its corresponding transported vector at any subsequent position of the turtle. This is done by using the module `ParallelTransportedArrow(vect_angle,vect_size)`, with *vect_angle* = $\beta$, and *vect_size* being a scaling factor for visualisation. At the new position, the parallel transported vector $\mathbf{v}$ makes an angle $\beta'$ with the current turtle's head in the current tangent plane corresponding to the original angle corrected by the cumulated angle (i.e., $\alpha$) by which the turtle turned on the path to the current point:

$$\beta' = \beta - \alpha \qquad (51)$$

The module `ParallelTransportedArrow(vect_angle, vect_size)` thus draws the transported vector $\mathbf{v}$ as a vector making an angle $\beta'$ with the turtle's head at the current turtle's position.

To illustrate how to program this with Riemannian L-systems, let us start with a program tracing a triangle at the surface of a sphere as in Figure 8c:

```
1  Axiom:
2    nproduce SetSpace(Sphere(1))
3    nproduce InitTurtle((0,0,0,1))
4    nproduce F(R*pi/2) +(90) F(R*pi/2) +(90)
           F(R*pi/2) +(90)
```

This program can be modified to show the different stages of a transported vector along the triangular path. For this, after setting the origin point of parallel transport at 0 using `ParallelTransportReset`, we insert the module `ParallelTransportedArrow()` at each vertex positions (List. 5, Figure 8c and d):

```
1  alpha = -30       # in degrees
2  R = 1
3  Axiom:
4    nproduce SetSpace(Sphere(R))
5    nproduce InitTurtle((0,0,0,1))
6    nproduce ParallelTransportReset
7    nproduce ParallelTransportedArrow(alpha,0.5)
8    nproduce F(R*pi/2)
9    nproduce ParallelTransportedArrow(alpha,0.5)
10   nproduce +(90) F(R*pi/2)
11   nproduce ParallelTransportedArrow(alpha,0.5)
12   nproduce +(90) F(R*pi/2)
13   nproduce ParallelTransportedArrow(alpha,0.5)
```

**Listing 5.** Parallel transport (see Figure 8c and d).

Figure 8e and f, illustrating the transport of a vector along a non-geodesic curve can be obtained in a similar manner by making short segments of size $dl \ll R$, each followed by small rotations $d\alpha \ll \pi$ and drawing the transported vector at each incremental step.

### 3.4.2. Interpretation of turtle movements in terms of differential operators.
According to the definition of parallel transport, we see that the head vector of the turtle **H** is always parallel transported by an F statement as it remains parallel to the tangent vector of the geodesic trajectory produced by F. Hence, parallel transport of a vector **X** in the current tangent plane $T_P\mathcal{S}$ of the turtle, through an F statement, is the vector **X**$'$ such that, if $\alpha$ denotes the angle between **X** and **H** in $T_P\mathcal{S}$, $P'$ and **H**$'$ are, respectively, the new position and head of the turtle after the execution of the F statement, then **X**$'$ is the vector in $T_{P'}\mathcal{S}$ that makes an angle $\alpha$ with **H**$'$ and such that $\|\mathbf{X}'\| = \|\mathbf{X}\|$.

Combined together, the Riemanian definition of a forward (F) and rotate (+) statement, implement the exponential map at the current turtle's position $P$. Let us take the turtle's orientation **H** as a reference orientation in current turtle's tangent plane. Then any vector **X** in this plane can be defined by a rotation $\alpha$ with respect to **H**, and a scaling factor $\|\mathbf{X}\| = l$. Then, the instruction,

```
1    nproduce +(alpha) F(l)
```

moves the turtle to the point $P' = \exp_P(\mathbf{X})$ with a new direction **H**$'$ at $P'$ that is the parallel transport of vector **X** at $P$ along the geodesic from $P$ to $P'$.

Reciprocally, given the current turtle's position $P$ and a different point $P'$ on the surface, the definition of the statement `RiemannLineTo` implements the logarithmic map at the turtle's position. The instruction:

```
1    nproduce RiemannLineTo(P_prime)
```

computes a geodesic as a ordered list of points on the geodesics $\{(u_n,v_n,p_n,q_n)\}_{n=0,\ldots,N}$, where $P_N = P' = \exp_P(l\mathbf{H})$. The vector $\mathbf{X_0} = (p_0,q_0)$ belongs to the tangent plane at $P$ and represents the initial direction of the velocity in the parameter space, and thus $\mathbf{X_0} = \log_P(P')$.

### 3.4.3. Closed polygons on a surface.
As illustrated by holonomy, various usual geometric properties valid in Euclidean spaces are no longer valid in curved spaces. In particular, the sum of the inner angles of a triangle is not 180 degrees nor even a constant number in general. This makes it difficult to draw closed polygonal curves just based of local operations such as moving forward by a length $l$ or turning by a certain angle $\alpha$, Figure 9a. In general, is not possible to map flat space onto a curved space map while preserving both

the angles and lengths. This means that we have to accept to drop some properties of polygons when mapping them in a curved space. However, it can be practically important in some cases to draw close figures on the surface and even to fill them with a particular colour or texture.

This issue related to holonomy can be circumvented in different ways in Riemaniann L-systems. A first option is to draw a polygonal line using the F module (thus making geodesic segments), Figure 9a, and to close the polygon by creating a final geodesic segment using a `RiemannLineTo` module from the current position of the turtle to its initial position. However, this solution may induce undesired biases on the length and orientation of the final segment if the total curvature enclosed by the polygonal line is significant.

A second option consists of tracing the polygonal figure in the parametric space rather than on the surface directly, and then project all the points of this polygonal line on the surface (Figure 9b). This can be considered as an indirect interpretation of the turtle instructions, which are momentarily executed in the parameter space rather than on the surface. This can be carried out by using the modules `StartIndirectInterpretation` and `StopIndirectInterpretation`.

The backside of this indirect interpretation is that the segments are in general not geodesics. It is possible to define a spherical square as a polygon made of geodesic edges with equal lengths and equal inner angles between consecutive edges (although different from 90 degrees). In Riemannian L-systems, one would position the turtle at the centre of the square, then send geodesic rays of equal length $d$ from this point with an angle of 90 degrees between each other (Figure 9c thin blue lines). This defines four points at the same distance of the square centre, which can then be joined by a geodesic using a RiemannianLineto primitive. This procedure can be used on more chaotic surfaces (Figure 9d), but with an additional loss of geometric symmetries of the square.

### 3.4.4. Drawing smooth curves on a curved surface.
In the Euclidean space, important families of parametric curves can be constructed by the use of straight lines. This is the case for instance of *basis spline curves*, or *B-Splines*, whose points are a weighted linear combinations of so-called control points (Piegl & Tiller, 1997). This linear combination is based on piece-wise polynomial basis functions whose degree controls the smoothness of the curve. An approximation of such curve can be achieved through algorithms that recursively subdivide the control polygon (polygon formed by the control points), such as the de Casteljau (Piegl & Tiller, 1997) or the Lane–Riesenfeld (Lane & Riesenfeld, 1980) algorithms. Here, we show how such algorithms can easily be extended with Riemannian L-systems to control smooth curves in curved spaces (Figure 10a–h).

Starting from the initial control polygon made up of geodesics segments, the number of controls points is first doubled by inserting new control points in the middle of each segment (duplication operation) (Figure 10b). The centroid of the geodesic is estimated as the point of the geodesic at equal geodesic distance between the two control points. Then, each initial control point is moved towards the midpoint of the adjacent segment (move operation) (Figure 10c and d). Each subdivision step is thus composed of duplication and move operations. Subdivision steps are reiterated until the desired level of approximation of the B-Spline curve is reached (Figure 10f–h).

Let us illustrate how this principle can be used to generate an oak leaf-like shape using Riemannian L-systems (List. 6). First, a set of control points is specified as the terminal nodes of a simple

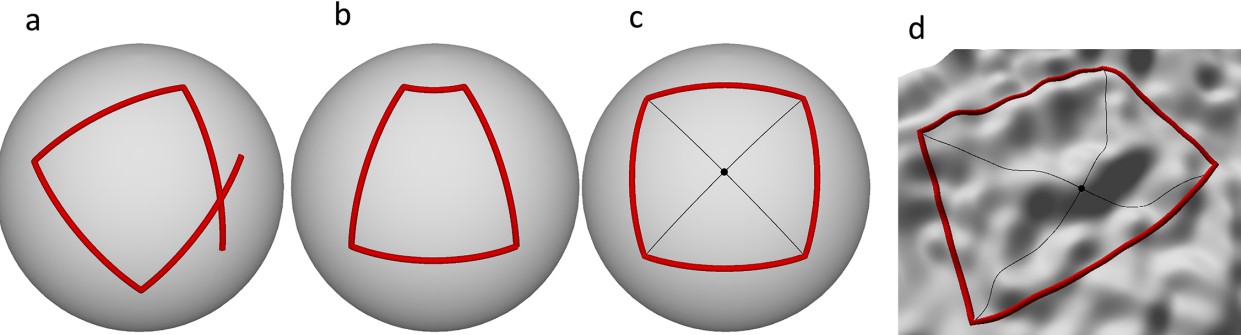

**Figure 9.** Drawing closed polygons on curved surfaces. (a) Failure to close a square by forcing consecutive sides to be at 90 degrees from one another on a curved surface. (b) Square drawn in the parameter space, and then pushed forward on the surface (c) Alternative geometric construction of the square preserving the right angle at the intersection of the diagonals and their length. (d) The construction in c can be used on more chaotic surfaces.

**Figure 10.** B-splines on a curved surface. (a) Initial quadrilateral control polygon on a sphere. (b) Duplication operation with new control points (in purple) inserted in the midpoint of each segment. (c) and (d) All original control points (in red) are moved towards the midpoint of their adjacent segments (in purple). For a B-Spline of degree 3, 2 move operations are applied. (e) Resulting control polygon after a complete subdivision step. (f)– (h) Successive control polygons (in green) after 1, 2, and 3 subdivision steps, respectively. (i)–(k) The B-Spline curve is defined by control points positioned using the Riemannian L-system (List. 6) that generates a simple branching structure. Resulting curve on (i) a flat surface, (j) an ellipsoid and (k) a bumpy ellipsoid. (l) The control polygon of a salamander shape. (m) and (n) The interpretation of the control polygon as B-Splines of degree 2 and 8, respectively. The degree of the curve controls the number of control points that influence each point of the curve.

tree structure using the `BSplinePoint` module, encapsulated within `StartBSpline` and `EndBSpline` modules. These control points must be provided in an order that respects the curve parameterisation. For this, the left branches are first generated (line 17), followed by the primary branch (line 18) and subsequently the right ones (line 19). The subdivision algorithm encoded in the

BSpline primitive of the system (lines 6 and 8) then generates the outline of an oak-like leaf. Depending on the embedding space, the shape is drawn on a flat surface (Figure 10i), an ellipsoid (Figure 10j) or a bumpy ellipsoid, respectively, in Figure 10k. The representation of the leaf contour dynamically adjusts to the local irregularity of the underlying space. In particular, on the bumpy

```
1  maxlength = 12
2  dl = 1.5
3
4  Axiom:
5    nproduce SetSpace(EllipsoidOfRevolution(5,8)) InitTurtle([0,-1,0,1])
6    nproduce StartBSpline(2)   # The degree of the B-Spline is given as parameter
7    nproduce BSplinePoint() [ A(0) ] BSplinePoint()
8    nproduce EndBSpline()
9
10 derivation length: 8
11 production:
12
13 A(clength) :
14   if clength < maxlength:
15     clength += dl
16     lateral_length = maxlength*lateralratios(clength/maxlength)
17     nproduce F(dl)
18     nproduce [+(60) F(0.1+lateral_length) BSplinePoint()]
19     nproduce [ F(0.1) A(clength)]
20     nproduce [-(60) F(0.1+lateral_length) BSplinePoint()]
21
22 interpretation:
23 A(l) --> BSplinePoint()
```

**Listing 6.** B-Spline curve built by positioning B-Spline control points at the end of a simple tree structure (see Figure 10j). The lengths of the lateral branches depend on a graphically defined function called `lateralratios`.

```
1  alpha = -45              # in degrees
2  ra = 1.                 # dimensions of the ellipsoid
3  rb = 0.5
4  lg = 10 * (2*pi*ra)     # length of the geodesic
5  Axiom:
6    nproduce SetSpace(EllipsoidOfRevolution(ra,rb))
7    nproduce InitTurtle((0,0,1,0))  # head points in the horizontal direction [1,0]
8    nproduce +(alpha)    # initial inclination with respect to horizontal
9    nproduce F(lg)       # draws a geodesic
```

**Listing 7.** Geodesic on an ellipsoid of revolution.

ellipsoid, the main axis of the skeleton is deflected towards the right, due to the specific curvature of the space.

Various types of shapes can be drawn on curved spaces using the same principle as illustrated by Figure 10l–n. Here, the control polygon represents a salamander sketched in Escher style. The original control polygon (Figure 10l) is recursively subdivided to achieve approximation of B-Spline curves of degree 2 and 8, respectively (Figure 10m and n). The higher the degree, the smoother the curve and more rough the details.

While the resulting curve appears visually smooth and approximates its control points in a manner consistent with the B-spline definition, it is not strictly a B-spline. This is because the smoothness of an embedded curve is fundamentally constrained by the differentiability class of the surface it resides on. Consequently, the curve's degree of differentiability may not meet the requirements of a true B-spline. The geometric properties of such embedded curves remain an open question for future investigation.

## 4. Freely growing forms on curved surfaces

Using the turtle primitives introduced above to move on curved surfaces, we now explore how to program the development of filamentous or branching forms on curved surfaces using Riemannian L-systems.

### 4.1. Geodesic trajectories

In the absence of any reason for deviating from a straight movement, a mobile moving in a curved space would nat-urally follow geodesics. On the sphere for example, starting at a point $P$ and heading in a direction $\mathbf{v}$ will produce a geodesic that corresponds to the great circle passing through $P$ in the direction $\mathbf{v}$. In more complex spaces, geodesics may show remarkable properties that can lead to non-intuitive patterns.

Let us consider, for example, geodesic trajectories on an ellipsoid of revolution, starting at equatorial points in directions given by different initial angles above the equator $\alpha$, Figure 11a. In Riemannian L-systems, these geodesics can be simulated with the simple program described in List. 7.

Here, only an axiom is defined, with no production rules, and directly produces modules that can be interpreted by a Riemannian turtle (no need of interpretation rules either). The axiom first triggers the use of Riemannian L-systems by defining a curved space in which the turtle should operate (line 6), then its initial position and reference orientation (pointing right in the horizontal direction) in this space (line 7), and finally after a local rotation with respect to the reference orientation (line 8), a geodesic line of length `lg` is drawn from the initial position heading at angle `alpha` below the horizontal direction (line 9). By varying `alpha`, we can observe the behaviour of geodesics on an ellipsoid, Figure 11a. They tend to stay in an equatorial band whose width depends on the inclination of the initial direction with respect to the equator: the smaller the initial inclination $\alpha$, the narrower the band. Interestingly, depending on the initial inclination angle, the geodesic will more or less densely cover this equatorial band.

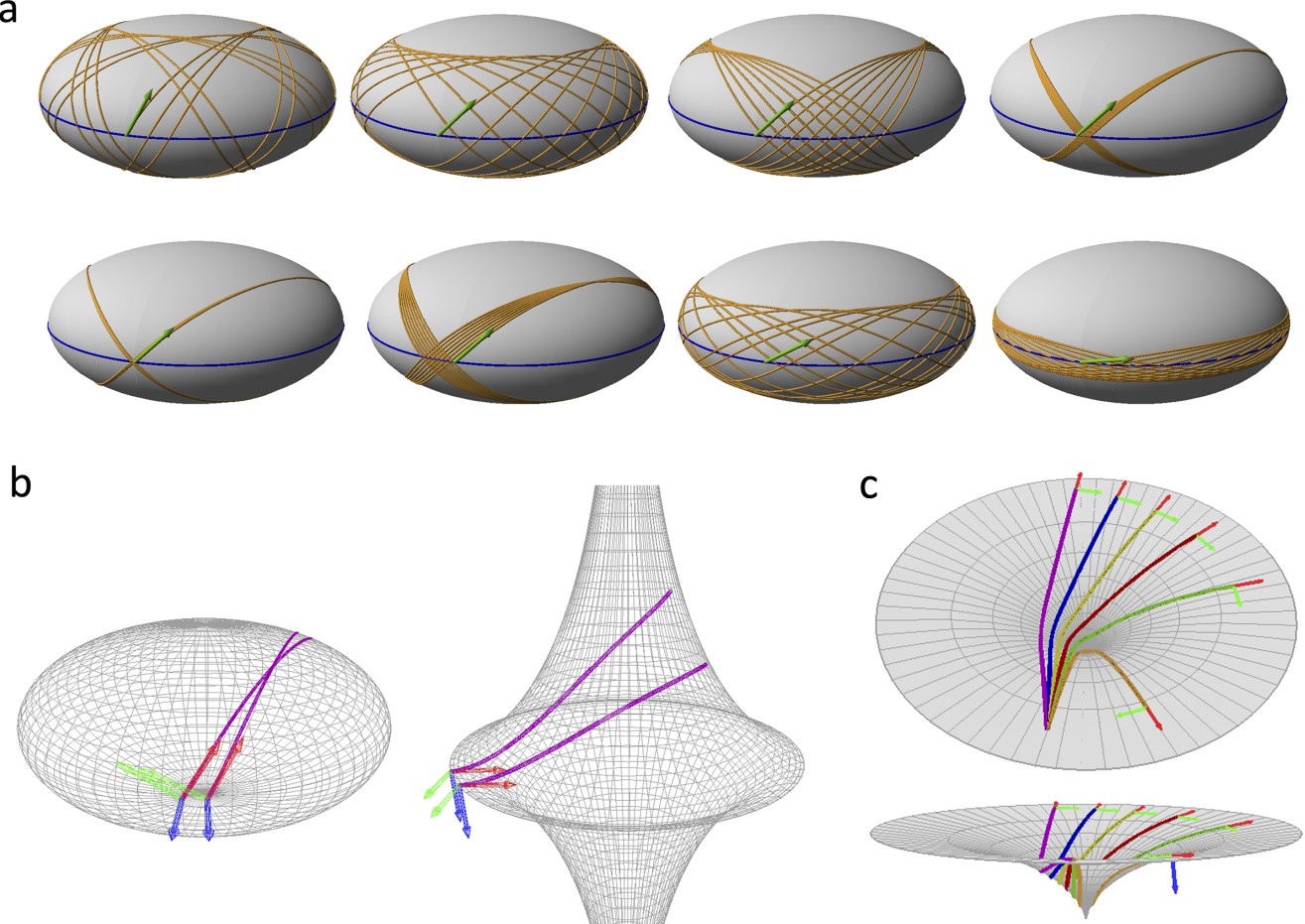

**Figure 11.** Geodesics on surfaces of revolution. (a) Geodesic on an ellipsoid of revolution, with a length 10× equator circumference (equator indicated in blue), and with different initial orientation (green arrow): from top-left to bottom-right: 60, 45, 44, 43.3, 43.2, 43, 30, 10 degrees inclination with respect to equator. (b) Comparison of the behaviour of close geodesic trajectories in spaces with positive (left: ellipsoid) and negative (right: pseudo-sphere) curvatures. In both examples, geodesics start with parallel orientation (red arrow). (c) Geodesics in a space with negative Gaussian curvature starting with varied initial orientations. Geodesics are all the more deflected by the space curvature that they pass closer to the centre of the shape.

The curvature of the space can in general be characterised by looking at how neighbouring geodesics behave (e.g., Carroll, 2014, p. 144). Geodesics initially parallel will tend to converge in spaces with positive Gaussian curvature and diverge in space with negative Gaussian curvature, Fig. 11b. It can be shown that this property, called *geodesic deviation*, characterises locally space curvature: the rate at which parallel neighbouring geodesics get closer or farther away from each other is proportional to the local curvature (Carroll, 2014).

Figure 11c shows an example of geodesics on a surface with negative curvature. Several geodesics displayed with different colours are emitted in slightly different directions towards the centre of the surface (central singularity). The closer they pass to the axis of the central singularity, the more the geodesics are deflected. Geodesics passing too close to the singularity are even reflected back (orange curve).

These behaviours are the consequences of a mathematical theorem about geodesics on surfaces of revolution, known as Clairaut's theorem (do Carmo, 1980, p. 259) for *Clairaut parameterisations* of general surfaces (O'Neill, 1966, p. 353):

**Theorem** (Clairaut). Consider a geodesic on a surface of revolution and denote $\alpha$ the inclination angle of the geodesic at a point

$P$ with respect to the latitude passing by $P$ and $r$ the radius of revolution at $P$. Then, along the geodesic we have:

$$r\cos\alpha = c_0, \tag{52}$$

where $c_0$ is a real constant.

Therefore, choosing an initial point and direction for a geodesic determines a value $c_0$ that will remain subsequently constant along the geodesic trajectory. As one moves along the geodesic, $r$ varies, and $\alpha$ must vary accordingly to respect Eq. (52). If $r$ decreases, $\cos\alpha$ must increase. At some point, if $r$ continues to decrease, $\cos\alpha$ reaches the value 1 (i.e., the curve is tangent to the local latitude at $P$), and the trajectory will be reflected back to increasing values of $r$ to avoid that $r$ decreases more and to keep up with Eq. (52). This means that the geodesics with this $c_0$ will be trapped in a part of the surface of revolution. For the ellipsoid (Figure 11a) for example, the geodesics keep in an equatorial band whose width is determined by the value of $c_0$ at the initial point and for the chosen initial direction. For the surface with negative curvature (Figure 11c), trajectories passing close to the central singularity have a radius which markedly decreases and thus induce a reversal of the angle $\alpha$ variation at some critical radius, thus bouncing back the geodesic curve.

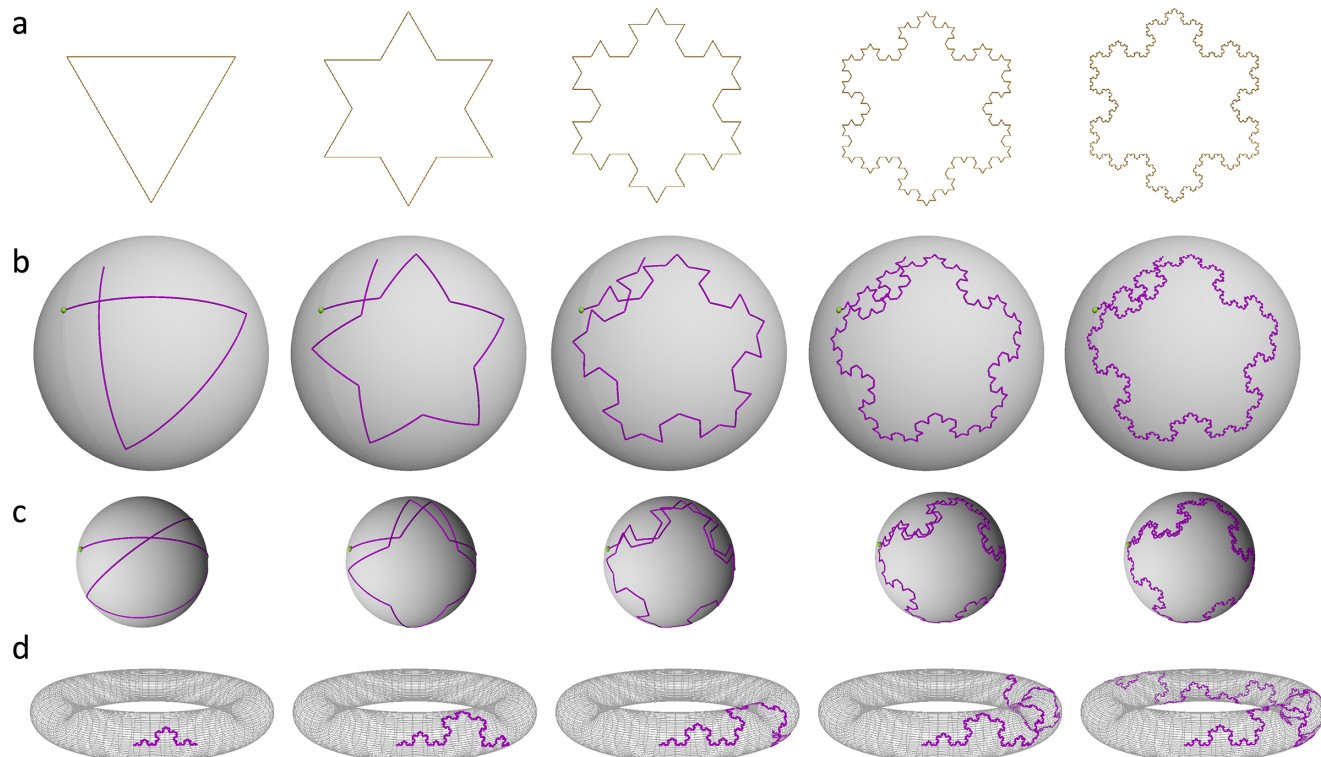

**Figure 12.** Using the turtle to draw fractals on curved spaces. (a) Prefractal sequence of the von Koch Curve in a flat space. (b) Prefractal sequence obtained by the same procedure as in a on a sphere of radius 1, and (c) on sphere of radius 1/2. (d) von Koch curves with increasing step length on a torus.

These examples illustrate how Riemannian L-systems can help explore mathematical properties in differential geometry with the efficiency and simplicity provided by a high-level programming language (see List. 7).

### 4.2. Turning and **branching**

**4.2.1. Fractals.** Being particular cases of manifolds, smooth parametric surfaces are locally flat and look like a Euclidean plane. However, at larger scales, the surface curvature is not negligible and can be revealed by trajectory holonomy. Fractals living on curved surfaces exhibit various levels of detail all along their entire structure. We may expect that the fine details of the structure, typically much smaller than the local surface principal radii of curvature, are not affected by the surface curvature. However, at coarser scales (of the order of magnitude of the radii of curvature), the fractal form must be affected in proportion of the local curvature. This is illustrated by the series of prefractal forms in Figure 12. The reference prefractal forms converge towards the well-known von Koch flake in a flat space, Figure 12a. When traced on a sphere, we see that the curve does not close anymore due to holonomy at large scale that tends to fold the curve faster than in the flat reference case, Figure 12b. However, one can notice that the curve pattern is almost not affected at fine scales where one easily locally recognises the pattern of the von Koch curve. Interestingly, if the radius of the sphere is reduced (the curvature is increased), the overall topology of the flake is dramatically affected, and contains only four main arms, Figure 12c, instead of six in the reference case. Similarly, Figure 12d shows a von Koch curve growing on a torus. When the curve size is small compared to the two main radii of curvature of the torus (left), the curve form is not affected significantly. However, as the curve grows, the overall curve geometry is markedly deformed (right).

These figures have been obtained with the program listed below (List. 8). One can observe that the modification of the code with respect to the reference flat case (List. 3) is minimal: only the specification of the turtle space has been added in the axiom. The rest of the code is unchanged. For the torus, one just needs to change line 2 by `nproduce SetSpace(Torus(1,0.3))`.

**4.2.2. Branching patterns.** Like in classical L-systems, branching patterns can be created easily with Riemannian L-systems using well-formed strings of modules. The turtle's interpretation mechanism is entirely conserved but runs on Riemannian states (Eq. (18)) instead of the classical turtle's states used in Euclidean geometry (Eq. (17)). The turtle thus draws branching patterns whose segments are geodesics, and branching angles are defined either explicitly by the programmer in the local tangent plane at the bifurcation point using a + statement before an F, Figure 13a–c or implicitly inferred through a `RiemannianLineTo` primitive. The generic program to produce these trees is as follows (List. 9):

As before, to change the space (Sphere, Pseudo-sphere, . . . ) and the initial position of the turtle in the space, only lines 6 and 8 have to be updated. The rest of the code remains unchanged.

Branching patterns, that are made of geodesic segments, are affected differently by the surface depending on both its extrinsic and intrinsic geometric properties, Figure 13a–c. Branches tend to get more dense on the sphere (constant positive Gaussian curvature), Figure 13a, less dense on the pseudo-sphere (constant negative Gaussian curvature), Figure 13b, and show a mixed effect on the torus that has an external and inner regions of, respectively, positive and negative Gaussian curvature, Figure 13c. On spheres, branching patterns are more bent towards each other as the sphere radius decreases (and the curvature augments). On the pseudo-sphere, the same tree, represented at three different

```
1  Axiom:
2    nproduce SetSpace(Sphere(radius=1))
3    nproduce F(1)-(120)F(1)-(120)F(1)
4  derivation length: 5
5  production:
6  F(x) : nproduce  F(x/3.0)+(60)F(x/3.0)-(120)F(x/3.0)+(60)F(x/3.0)
```

**Listing 8.** Fractal curve (von Koch flake) on curved surfaces (see Figure 12b-d).

```
1  N = 7           # Depth of the tree recursion
2  iangle = 45   # Insertion angle
3  ilen = 0.2    # Length of a segment between two branches
4
5  Axiom:
6    nproduce SetSpace(Sphere(radius=1))
7    nproduce InitTurtle([0.,0.,0.,1.]) # turtle's head pointing upwards
8    nproduce A(0)
9
10 derivation length: N
11 production:
12 A(n) :
13   a =  iangle if n % 2 else -iangle
14   if n<N:
15     nproduce  F(ilen) [+(a)A(n+1)] A(n+1)
```

**Listing 9.** Tree patterns on surfaces (see Figure 13a–c).

```
1  N = 10                                    # Number of target points
2  target_pts = [[0.,0.3],[0.4,0.],...]    # Array of (u,v) coords of target points
3  ilen = 0.1                                # Length of a segment between two branches
4
5  Axiom:
6    nproduce SetSpace(Patch(leafblade))   # Nurbs patch in the form of a leaf blade
7    nproduce InitTurtle([0.,0.,1.,0.])    # initial (u,v,p,q) coords of the turtle
8    nproduce A(0)
9
10 derivation length: N+1                    # Number of segments on the main stem
11 production:
12 A(n) :
13   if n<N:
14     nproduce  F(ilen)
15     nproduce [RiemannLineTo(target_pts[n],20)]  # segment made of 20 subsegments
16     nproduce A(n+1)
```

**Listing 10.** Tree patterns based on BVP problem (see Figure 13d).

altitudes, shows very contrasted shapes. As the pseudo-sphere is of constant curvature, these variations are not due to a change in the local intrinsic geometry (that keeps constant everywhere), but due to the change in the extrinsic geometric component only (indeed, while their product is constant, the principal radii of curvature are not constant throughout the pseudo-sphere surface).

Branching patterns can also be produced by joining the current position of the turtle to target points during the construction of a central stem, Figure 13d. For this, the turtle must be placed at an initial position on the surface, pointing in an initial direction (corresponding roughly to the direction of the future main stem). Assume that a set of $N$ target points is defined by their $(u_n, v_n)$ coordinates on the surface. We aim at creating a main stem composed of $N$ segments that follow a geodesic of the surface and such that, at the end of each segment $n$, a lateral segment is drawn to the target point $(u_n, v_n)$, (List. 10):

Segments on the main stem are iteratively computed as a series of IVP using an F statement (line 14). The length of the each segment is determined by the user. At each step $n$, a lateral segment is computed as a BVP to the $n$-th target point using the RiemannianLineTo primitive (line 15). This determines

automatically all the points of the lateral segment, and in particular, a specific insertion angle on the main stem (log map, see above), Figure 13d.

### 4.3. Applications

Let us now illustrate on some examples how Riemannian L-systems can be applied to the modelling of some biological organisms and patterns.

**4.3.1. Filamentous growth.** Pollen grains are transported from flowers to flowers by wind or animals. They can germinate if they land on specific elongated cells, called papillae located at the tip of the stigma, the female organ of the flower. When they germinate, a pollen tube starts to grow out downward the papilla, and keeping at the papilla surface (Riglet et al., 2020). Papillae have roughly a pin-like structure, but may vary in shape within or between species and present either convex or non-convex forms. Biologists try to understand the possible physical or chemical cues that guide the growth of the pollen-tube downwards. One of the hypothesis is that the precise geometry of the papillae may play an important role in the guidance of the tube and that the tube could follow geodesics of the papillae surface (Riglet et al., 2025).

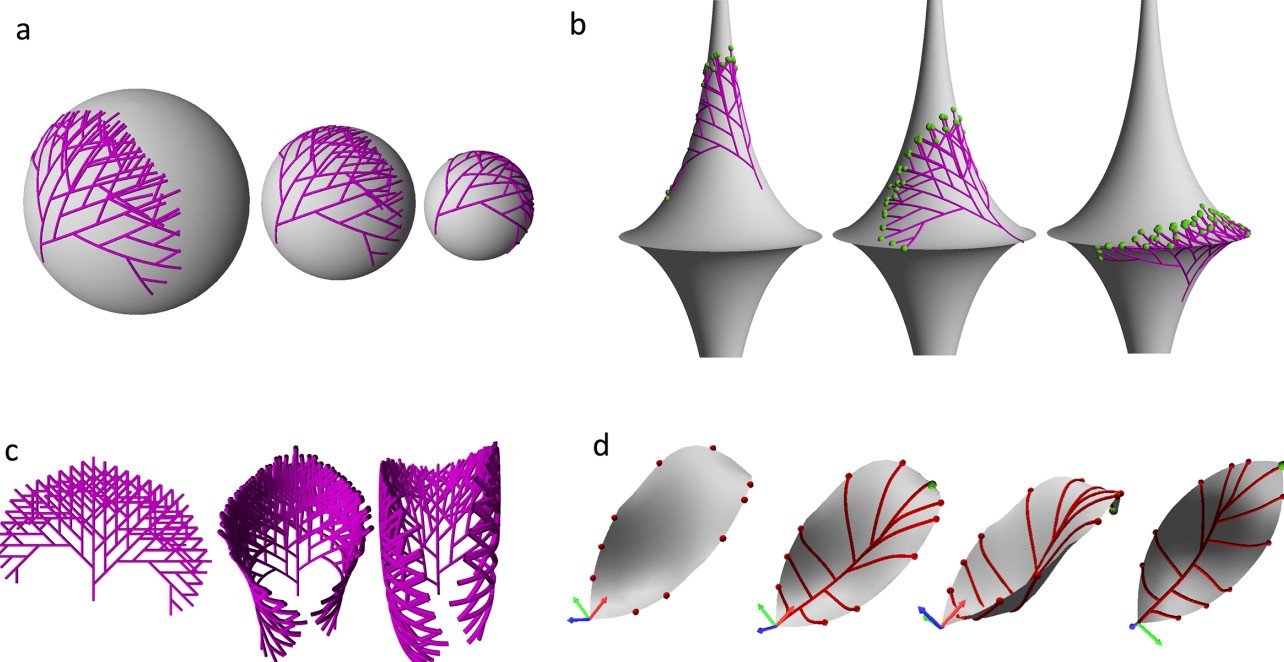

**Figure 13.** Growing tree structures on curved surfaces. (a)–(c) Trees created with a shooting algorithm to solve an IVP (F primitive). (a) Tree on spheres (constant Gaussian positive curvature) with decreasing radii (1, 0.7, 0.5). (b) Tree on a pseudo-sphere surface (constant Gaussian negative curvature) at different altitudes showing the effect of a local change of the extrinsic geometry on tree structures. (c) Tree growth on a torus. Left: reference tree grown in a flat space. Middle: the tree trunk is aligned along the external great circle (region of positive Gaussian curvature). Remark in the central region at the tip that the small branches form a very densely organised fan. Compare with Right: the tree trunk is aligned along the inner great circle (region of negative Gaussian curvature). In the central region at the tip that the small branches form a less dense fan. (d) Tree representing the veins of a leaf, created by joining pre-specified (red) points on the rim (left) to a main branching system using a RiemannianLineTo primitive (BVP). Next to right: resulting branching system in the same view as left, followed on the right by a view slightly tilted, and to the right-end, the back of the leaf.

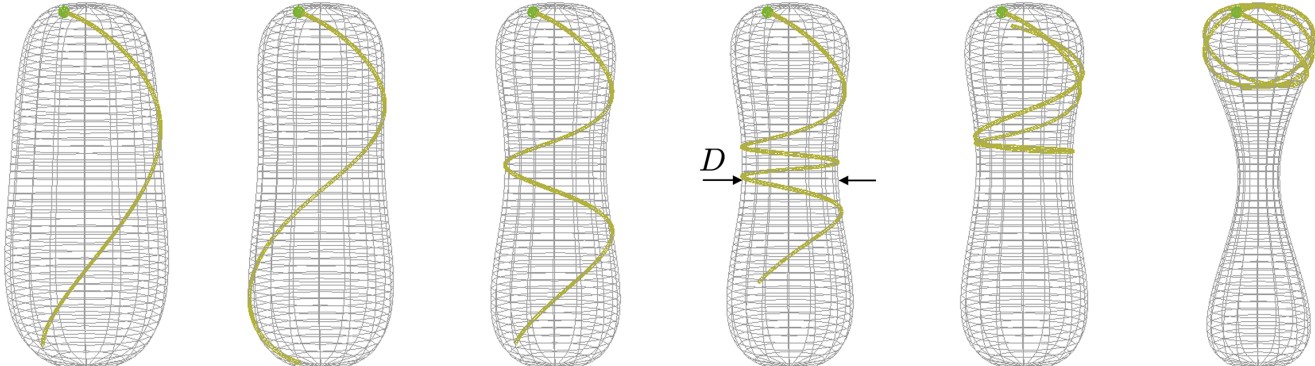

**Figure 14.** Geodesics on a pin-shaped surface. A geodesic of constant length *l* in yellow is initiated at a constant angle (28 degrees downward) with respect to the latitude at the level of the pollen grain position (green point). From left to right: the mid-height neck of the pin surface is progressively reduced from left to right (the value of *D* decreases). As a result, the geodesic coils increasingly around the neck, up to a point where it cannot pass the neck anymore and coils in the top region.

Riemannian L-systems can be used to analyse this growth process and for example to explore how the shape of the papillae impacts the possible geodesic trajectories that a tube might follow. Let us consider, for example, a pin-like structure and decrease progressively its neck (i.e., its diameter at mid-height) to pass from a convex to a marked non-convex shape, Figure 14. Starting from a point *P* at the tip of the structure representing the position of the pollen grain and a given orientation of the initial germination of the tube from the grain, we observe that the trajectory of geodesics is highly affected by the change in the surface shape: as the neck narrows down, the geodesic increasingly spirals when approaching the neck and continuously goes downward. Surprisingly, below a certain neck diameter, the geodesic is reflected and coils back without crossing the neck any longer, Figure 14 last two right

examples. This is again a consequence of the Clairaut theorem (Eq. (52)). Altogether, this suggests that geometry is potentially a key, possibly genetically regulated, actor in the driving of the pollen tube. This hypothesis was explored in a recent work using these tools (see Riglet et al., 2025).

**4.3.2. Branching system growth.** Leaves' vascular networks are essential to gas, water and sugar exchanges in plants. They are an integral part of the development of the leaf and result from developmental mechanisms that are not yet well understood. In the last decade however, a few studies have made progress on the understanding of the genetic regulation of leaf development and its connection with the construction of the vascular network (Katifori, 2018; Runions et al., 2017). In these works, leaves are considered as flat medium

```
1  N = 10                                   # Number of branches on main stem
2  dl = 0.2                                 # segment length between 2 branches
3  Axiom:
4    nproduce SetSpace(leaf_patch)          # space that forms the leaf blade
5    nproduce PlotSpace()                   # Plots the NURBS patch
6    nproduce InitTurtle([0.001,0.27,1,0.45])
7    nproduce -(6.5) A(N)                   # correct slightly turtle's head
8
9  # Main apex
10 A(n)                                     # n=seg cpt-down on this axis
11   if n > 0:
12     r = BASERADIUS*n/N
13     nproduce F(dl,r)
14     d = 8 + 0.1*n**2                     # deflection angle from geodesic @next order
15     a = 5*n/N                            # insertion angle adjustment
16     nproduce [+(30+a)B(5,r,d) ]
17     nproduce [-(30)  B(5,r,-d)]
18     nproduce A(n-1)
19
20 # Branch apices
21 B(n,r,d):                                # n=seg cpt-down,r=radius, d=deflexion angle
22   if n > 0:
23     nproduce +(d) F(dl,r*n/N)
24     if n == 4 :                          # the second segment forks
25       a = -8 if d < 0 else 8            # new deflection angle
26       nproduce [+(5)B(n-1,r,a)]
27       nproduce [-(7)B(n-1,r,a)]
28     else:
29       nproduce B(n-1,r,d)
```

**Listing 11.** Cabbage leaf as an IVP (see Figure 15c and d). Note that in the following application examples some code details have delibarately been simplified to highlight code structure. Complete listing codes can be found in the notebooks describing the figures at https://github.com/fredboudon/RiemannianLsystems.

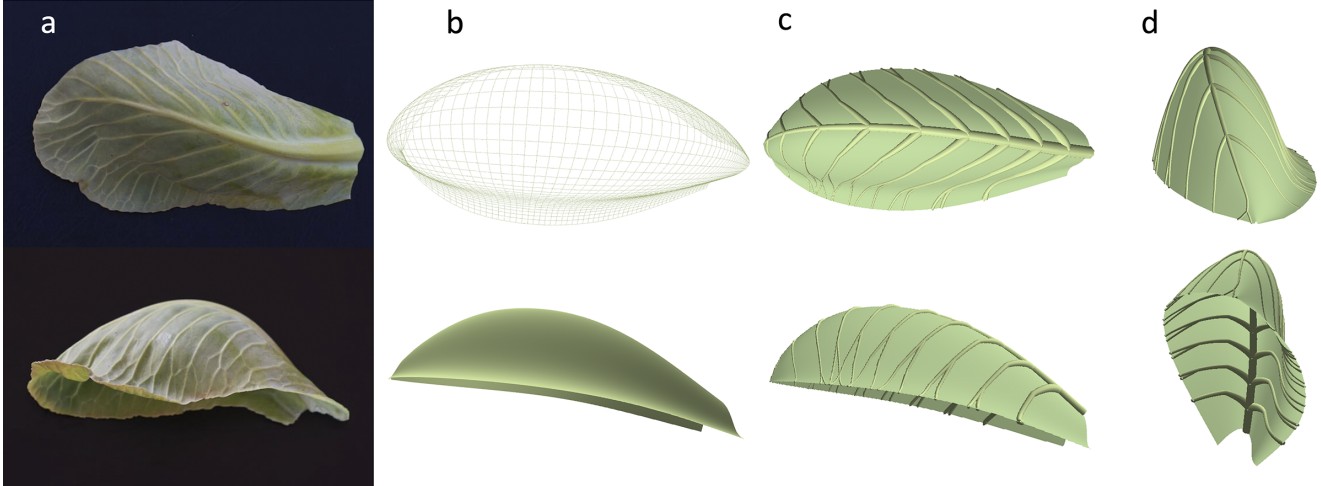

**Figure 15.** Cabbage leaf model. (a) Photos of a white cabbage leaf (up: top view, below: side view). (b) Approximated NURBS model of the cabbage leaf. (c) and (d) Different views of the vascular network constructed with Riemannian L-systems, with veins corresponding to geodesics computed as IVPs.

represented by a triangulated mesh in which the vascular network is embedded. From a geometrical point of view, "*the modeling of leaves that are curved as well as lobed or serrated remains an open problem.* [...]. *Challenges are posed by* [...] *the need to replace straight segments with their more complicated counterparts defined on curved surfaces: the geodesic curves*" (Runions et al., 2017). Riemannian L-system provide a first generic tool to address this problem with high-level programming constructs. We illustrate this on the vein network of the curved leaf of white cabbage, Figure 15a.

For this, a geometric model of the curved leaf blade is first constructed using NURBS parametric surfaces (also called NURBS

patches; Piegl & Tiller, 1997) embedded in the software platform L-Py (Boudon et al., 2012), Figure 15b. Then, the turtle is positioned at the bottom of the leaf, with its head vector **H** pointing roughly in the direction of the ridge of the surface (lines 4–7 in List. 11).

A main apex produces segments of equal size following a geodesics (i.e., using an F primitive) and then producing two opposite lateral branches with insertion angle close to 30 degrees (lines 13–18). Each lateral branch is composed of segments, the second of which forks into two branches (lines 23–29).

The different parameters of this model have been adjusted by hand to visually reproduce main geometric traits of the image in Figure 15a. However, it can be noticed that the number of

```
1  Axiom: SetSpace(trunkshape) InitTurtle([0.05,0.001,1,0]) A(0)
2
3  production:
4  A(n):
5    if n < NMAX:
6      dl = l/
7      nbleaf
8      for i in range(nbleaf):
9          nproduce F(dl) [ Leaf ]
10     nproduce [ +(60 * pow(-1,n)) A(n+1) ]
11     nproduce A(n+1)
12
13 interpretation:
14 Leaf:
15   hang = uniform(-45,45)
16   vang = uniform(60,90))
17   u,v, _, _ = turtle.uvpq
18   nml = trunkshape.getNormalAt(u,v)
19   tgt = trunkshape.getVTangentAt(u,v)
20   produce EndSpace SetHead(nml,tgt) RollToVert() +(hang) F(0.1) &(vang) LeafSymbol
```

**Listing 12.** Sketch of the L-systems generating an Ivy structure over a generalised cylinder representing the trunk of a tree (see Figure 16).

parameters remains limited and they only necessitate fine grain tuning (at the level of branch insertion angles in particular). The core branching system makes use of geodesic lines and is straightforward to program. Only deflections with respect to this geodesic pattern need to be adjusted. In particular, the main stem was not adjusted at all (line 13) and the corresponding geodesic follows the ridge of the curved leaf blade, despite the fact that it is slightly twisted near the tip of the leaf as can be observed in the top of Figure 15d.

### 4.3.3. Plant branching system.

Some plants, such as lianas, cannot support their own weight as they grow. They then often grow on a supporting structure, which could be another plant, a rock or a building for instance. The supporting structure, in turn, may possess intricate geometric characteristics. If we assume that, in the absence of any extra external force, the axes of the plants grow straight, then we can model the growth of the axes by following the geodesics of the supporting structure. Conventional approaches rely on collision detection techniques using a voxel representation of the 3-D space (Greene, 1989) or bounding volume hierarchies (Wong & Chen, 2016) to determine the path on the structure that should be followed. In the case of complex geometry of the supporting structure, substantial refinements of the voxel representation are required to capture the detailed changes in curvature leading to computationally intensive generation. Riemannian L-systems present an efficient means to simplify such modelling process by embedding the generated plant shape directly onto the supporting structure, thus mitigating the complexities associated with the curved geometry of the supporting structure.

In the example depicted in Figure 16, the climbing of an ivy plant is directly simulated on the external surface of a tree trunk. This process involves representing the trunk as a generalised cylinder characterised by an S-shaped central axis, and with a linearly decreasing radius along the axis. The cylinder cross-section incorporates concavities indicative of the differential radial growth of the trunk and its fusion with aerial components of the roots. The initiation of the ivy growth occurs at the base of the trunk, with an upward orientation. A sketch of the code used to simulate the ivy growth is given in List. 12. Ramifications are regularly generated at a constant insertion angle along the axes, resulting in the coverage of the trunk surface by the ivy structure (line 9).

Interestingly, while ivy stems remain confined to the surface of a supporting structure due to the adhesion of their adventitious rootlets, ivy leaves are not subject to this constraint. Their petioles allow them to extend outward into free space, optimizing light capture while the stem remains anchored to the substrate. To model such dual behaviour, the embedding of the generated shape onto the surface can be stopped using the `EndSpace` command (see line 20). When incorporated into the L-string, the geometric interpretation of the subsequent modules of the string will recover the conventional behaviour of L-systems within the Euclidean 3-D space embedding. For the ivy example, the normal vector to the surface at the leaf insertion point is estimated, and the turtle is reoriented accordingly. Subsequently, a roll rotation is then executed to establish a horizontal orientation for the leaves. The petioles and the leaf blades are then generated. While geodesic are intrinsic objects (only depending on the surface's metric), this example illustrates the possibility in Riemannian L-systems to use as well primitives related to the embedding of the surface in the 3-D Euclidean space, and thus to take into account the full geometric information of the trunk surface. It also illustrates how multiple embeddings can be combined simply within the same Riemannian L-system. This could be generalised further by considering the generation of a plant onto multiple support structures. For this, the different embedding spaces can be set by inserting several `SetSpace` or `EndSpace` commands into the L-string.

## 5. Feedbacks between surface and embedded forms

Be they of mathematical, physical or biological nature, forms are built according to construction rules that operate within some substrate space, e.g., a plane or a curved surface. These rules may be completely independent of the space in which the construction process operates. This was the case in all the previous examples. However, in many situations, morphogenesis relies on external cues that are used during development by the construction rules to make decisions and orient form development (by changing growth orientation or speed, by creating branches, etc.). In particular, in such cases, geodesics can be seen as default trajectories and construction rules indicate how to deviate from these reference trajectories to construct the target form based on local information.

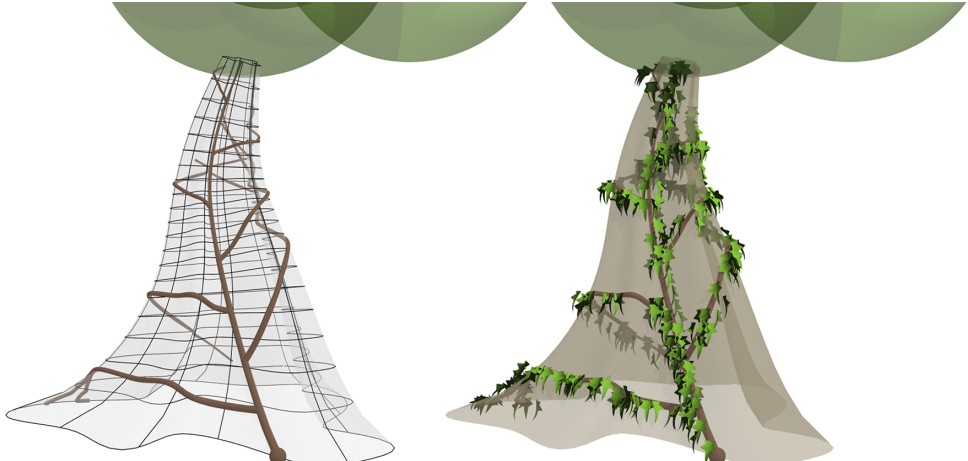

**Figure 16.** Climbing ivy. A tree trunk is modelled as a generalised cylinder on which the growth of an ivy is simulated. On the left, the wireframe representation of the trunk and the branching system of the ivy. On the right, semi-transparent polygonal representation of the trunk with the full leafy ivy structure.

Spatial informational cues may be considered as fields living in the substrate space, and that may be constant or change in time. In general, these fields have scalar, vectorial or more generally tensorial types, and may represent either geometric (e.g., curvature and principal directions), physical (e.g., obstacles, molecular concentrations, stresses and material properties) or more abstract (e.g., energy density, directional anisotropy and signals) local quantities.

Forms thus, in general, result from the interaction between three key factors: A substrate space on or in which the form develops, information fields defined on this space that may or not be (partially) produced by the form itself, and the growing form (here a filament or a branching network). Hereafter, we explore how sensing external fields may contribute to shaping forms using Riemannian L-systems, first in a fixed embedding space, then in a dynamically deforming embedding space.

### 5.1. Information feedback

In Riemannian L-systems, to make use of external fields living in the substrate space (here a surface) during development, we must allow the form being constructed to probe its space environment at anytime. This will be carried out by allowing the modeller to access turtle geometric information from within the production rules.

For this, we extended the L-system mechanism of sensitivity to the environment (see Section 2.3). By using a new special query module ?T inserted in the L-string constructed at a given

derivation step, the modeller can retrieve the actual embedding space and the turtle's current position and orientation on it during the interpretation phase, similarly to the ?P, ?H, ?L and ?U modules used in context-sensitive systems for retrieving position and orientation of the turtle in the Euclidean space – see Section 2.3.4. At the next derivation step, the modeller can then access the turtle's state recorded in the variable of module ?T and take decisions based on the information contained in this state.

In this way, the space can be queried to retrieve all types of information attached to the surface: geometric primitives (covariant basis, surface normal, principal directions, Gaussian and mean curvature, principal curvature and directions …) using dedicated primitives, as well as any type of field value stored by external processes on the surface.

**5.1.1. Scalar fields.** Scalar fields can express either local geometric properties of a surface or spatial distributions of physical or biological quantities defined on the surface. Let us consider, for example, how the random movement of a set of turtles can be canalised by their ability to read out locally the geometric characteristics of the embedding surface and to use it to make move decisions, Figure 17a and b. We assume that the turtles locally sense the surface curvature and avoid to go in flat (0 Gaussian curvature) or saddle-like regions (negative Gaussian curvature). These random walks are thus restricted to regions of (strictly) positive curvature (indicated in red in Figure 17c). To model this situation in Riemannian L-systems, one would basically modify the random walk algorithm described in List. 2 as follows (List. 13):

```
1  epsilon = 0.002
2  production:
3  ?T(state) A(n) :
4    uvpq = state.uvpq         # Current turtle's position (u,v) & orientation (p,q)
5    surface = state.space
6    found = False
7    while not found
8      a = 360*random()
9      uvpq_rot = riemannian_turtle_turn(uvpq, surface, a)
10     uvpq_seq = riemannian_turtle_move_forward(state.uvpq, surface, step)
11     uvpq_new = uvpq_seq[-1]         # Last point of the tentative move sequence
12     K,H,kmin,kmax = state.space.localCurvatures(uvpq_new[0],uvpq_new[1])
13     if K > epsilon: found = True  # Test if Gaussian curvature is positive
14   nproduce +(a) P(uvpq_seq) ?T A(n+1)
```

**Listing 13.** Random walk keeping on regions of positive Gaussian curvature (see Figure 17a–c).

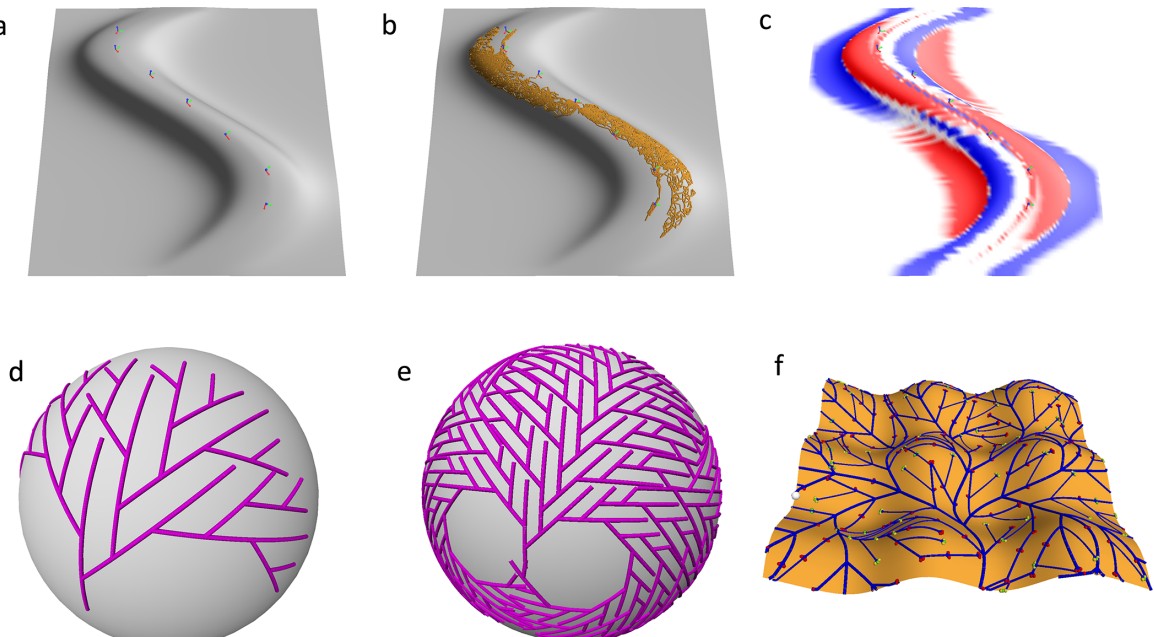

**Figure 17.** Making use of information available in the embedding space. (a)–(c) Random walks in regions of positive Gaussian curvature. (a) Curved space with seven random walkers initially positioned at the indicated frames. (b) Canalised random walks after 200 steps for each walker. (c) Map of Gaussian curvatures $K$: red regions with $K > \epsilon$, and white to blue for $K \le \epsilon$, $\epsilon = 0.002$. (d)–(f) Self-avoiding branching structures (d) on a sphere, (e) filling the sphere and (f) on an egg-box like landscape. Green resp. red points represent growing, resp. blocked, apices.

The query module ?T is inserted just before each moving apex in the L-string at each derivation (line 14). At the next derivation, the turtle's state (line 3), automatically updated at the last interpretation step, can be used to get the current position and orientation of the turtle (line 4) as well as the current embedding surface (line 5). Then, the model tries to make moves in different random position until one of this moves happens to be in a region of positive curvature (lines 7–13). Different primitives make it possible to simulate the turtle's rotation by an angle $a$ (line 9), to generate the sequence of coordinates corresponding to a potential step in this new direction (line 10), to compute the Gaussian curvature $K$ at the destination point (lines 11 and 12) and to test whether its value is positive (line 13). A sequence of coordinates associated with a positive $K$ is thus selected and used to draw the turtle's trajectory using the P primitive (line 14). As a result, one can observe that the random walks get canalised in the regions of positive Gaussian curvature, without being able to cross areas of flat or negative curvature, Figure 17b and c.

A scalar information can also result from a read out of the physical environment. A geometric form, for example, can be physically constrained in its development by its already existing parts. As an example, consider the construction of self-avoiding branching patterns, Figure 17d–f. A specialised data structure and associated primitives have been developed to record segments produced as the form grows and to test the presence/absence of already existing segments on the surface at specific locations, List. 14. An empty data structure is first created (line 10). Then, the apices of branching structure grow (line 16). Like before, a query module is used to recover the current state of the turtle before each apex. Instead of creating a segment, an apex first computes the path on the surface that would correspond to this new segment (line 17). It then tests whether an intersection is detected with previously created segments (line 18). If yes the apex does not grow (line 19). If no, the new segment is added to the list of already existing segments

(line 21), and the segment is drawn (line 22), together with a new lateral bud (line 24) and a renewed main apex (line 25).

This procedure is generic and does not depend on the underlying space. Figure 17f, for example, illustrates the same self-avoiding branching algorithm applied to an egg-box surface with a more complex geometry with both positive and negative Gaussian curvatures.

**5.1.2. Vector fields.** Scalar fields provide information that can indirectly be used to affect turtle's trajectory. A more explicit directional information may be provided by a vector field. At any point of the surface, the turtle can compute the angle $\alpha$ between its current heading direction $\mathbf{H}$ and the local vector value of the field $\mathbf{V}$, and orient its trajectory locally in a direction related to the vector field. For example, a deflection of the otherwise geodesic trajectory in the direction of a vector field is in general called *a tropism*. Formally, let us denote $\alpha$ the current angle between the turtle heading direction $\mathbf{H}$ and the local value of the vector field $\mathbf{V}$ at the turtle's position, $\alpha = \widehat{\mathbf{H}, \mathbf{V}}$, and $\Delta s$ the small step length that the turtle must make at the next derivation step. We assume that the tropism will deflect the turtle's initial direction from a geodesic by imposing an in plane geodesic curvature proportional to the angle $\alpha$ and that $\Delta s$ is sufficiently small so that the geodesic curvature is considered constant over the step length $\Delta s$. Then, we have:

$$\frac{\Delta \alpha}{\Delta s} = \sigma \alpha, \tag{53}$$

where $\sigma$ is a sensitivity to the 'tropism force'. This provides a direct expression of the deflection angle $\Delta \alpha$ by which the turtle's movement must be affected at the next iteration step to progress over a distance $\Delta s$ due to tropism.

To illustrate this, consider a geodesic starting at the equator of an ellipsoid of revolution and heading east and slightly north, Figure 18a. In the absence of a field (or of a feedback between a

```
1  R = 1.         # Radius od the Sphere
2  N = 7          # Depth of the tree recursion
3  iangle = 45    # Insertion angle
4  ilen = 0.2     # Length of a segment between two branches
5  trajectories = None # Will contain the set of already existing tree branches
6
7  Axiom:
8    space = Sphere(R)
9    nproduce SetSpace(space)
10   trajectories = LineSet(space) # Initializes the set of existing branches (lines)
11   nproduce InitTurtle([0.,0.,0.,1.])
12   nproduce A(0)
13
14 derivation length: N
15 production:
16 ?T(state) A(n) :
17   uvpq_s = forward(state,ilen)  # Pre-computes a new segment (sequence of points)
18   if trajectories.test_intersection(uvpq_s): # Does the new segment intersect
       previous ones?
19     nproduce ?T A(n+1)
20   else:
21     line_id = trajectories.add_line_from_point(state.uvpq,uvpq_s) # updates set of
       trajectories
22     nproduce P(uvpq_s) # Draw the precomputed new segment sequence of points
23     a =  iangle if n % 2 else -iangle
24     nproduce [+(a) ?T A(n+1)]
25     nproduce ?T A(n+1)
```

**Listing 14.** Non self-intersecting tree (see Figure 17d–f).

field and the turtle's displacement), the turtle follows a geodesic trajectory, Figure 18a. In the presence of a field (Figure 18b), the turtle may react to the field and use this information to locally modify its natural trajectory. Here, we consider a vector field resulting from the gradient of a scalar field corresponding for example to the diffusion of a substance from the north pole from high (red) to low (yellow) concentrations of the substance). By reading this gradient, the turtle can deflect its trajectory in the corresponding direction. As a result, the trajectory is attracted at the north pole and trapped in a circular attractor (Figure 18b). Interestingly, this attractor results from an equilibrium between the effect of the surface geometry that, in the absence of tropism 'force', would keep the turtle on a geodesic and thus make it possible to escape the north region after having visited it (Figure 18a), and the tropism force that constantly deflects the trajectory towards the north pole.

Programming such a feedback function follows the design pattern illustrated in List. 15. A query module ?T before the growing apex makes it possible to retrieve at each step the current state of the turtle at the apex (lines 15 and 26). At the position $(u,v)$ of the apex, the value of the tropism vector field is evaluated (here as the gradient of a scalar field, lines 17–19). The angle $\alpha$ between the tropism vector and the heading direction of the turtle is then computed (lines 21 and 22) as well as the deflection angle due to tropism over a step length *slen* according to the tropism attraction described in Eq. (53) (line 23). The sign of this angle depends on the relative orientations of the heading and tropism vectors (line 24). Finally, the turtle turns according to the deflection angle and draws a portion of geodesic over length *slen* (line 26), thus deflecting its trajectory towards the tropism vector.

This design pattern can be applied to model similar situations in biological applications. For example, Figure 18c and d illustrates the possible effect of a surface tropism on a tip-growing filament at the surface of a pin-like structure. This could for instance represent the growth of a pollen tube that would be attracted by a chemical gradient. In the absence of tropism, the filament follows

the geodesics of the pin-like surface, Figure 18c and d-left. In the presence of a tropism due to the surface gradient (indicated by a variation of colours from red to blue), the filament is deflected from geodesic curves and heads more rapidly towards the bottom of the surface. This phenomenon is amplified by an increase of the coupling between the filament growth and the directional cue (increase of $\sigma$ from left to right in Figure 18c and d).

### 5.2. Growing forms on dynamic surfaces

Forms can be constructed on surfaces that change in time. This change may affect the embedded form in different ways that we explore in this section. Here, we restrict our analysis to form construction processes that are significantly more rapid than the change of the surface geometry so that, during the form construction, the surface geometry can be considered steady. In this way, after one growth step of the surface, several steps of growth for the form can be simulated repeatedly until convergence.

**5.2.1. Convected versus floating forms.** Once a form has been constructed on a surface, two extreme subsequent types of evolution of this form can be considered.

Let us first assume that the points on the surface can be tracked in time. They can be considered as material points whose movement define the surface evolution in the 3-D space. At the initial time, the form passes through a set of material points to which the form is considered to be 'attached'. While the surface changes in time, the relative positions of the material points change, which induces a corresponding deformation of the form geometry on the surface. As the form depends on the definition of material points and their movements, we say that such form definition is *extrinsic*. An extrinsic form is being *convected* by the material flow on the deforming surface.

This situation is illustrated in Figure 19a and b using a fractal form. A Sierpinski carpet (e.g., Peitgen et al., 1992, p. 81), whose

```
1  a, b = 1., 0.5
2  N = 200
3  sigma = 1.          # sensitivity to gradient
4  slen = 0.1          # Length of a step at each derivation
5  def field(u,v):     # function returns a scalar value as function of u,v
6    ...               # --> not detailed
7
8  Axiom:
9    nproduce SetSpace(Ellipsoid(a,b))
10   nproduce InitTurtle([0.,0.,1.,0.])
11   nproduce ?T A(field)
12
13 derivation length: N
14 production:
15 ?T(state) A(sf) :
16   u,v,p,q = state.uvpq  # retreives current pos, dir of turtle
17   gradf_uv = gradient(state, field, u,v) # gradient of the scalar field at u,v
18   pushfwd = state.pushforward(u,v) # local pushforward operator
19   gradf = state.pushforward(gradf_uv) # gradient vector on the surface in 3D
20
21   heading = state.heading     # turtle's head in 3D
22   alpha = angle_between(gradf, heading) # deviation of turtle's head from gradient
23   defl_angle = sigma * alpha * slen # deflection angle to correct turtle's dir
24   sgn = 1 if state.space.positive_orientation(heading,gradf)) else -1
25
26   nproduce +(sgn*to_deg(defl_angle) F(slen) ?T A(sf)
```

**Listing 15.** Tropism on ellipsoid of revolution (see Figure 18a and b).

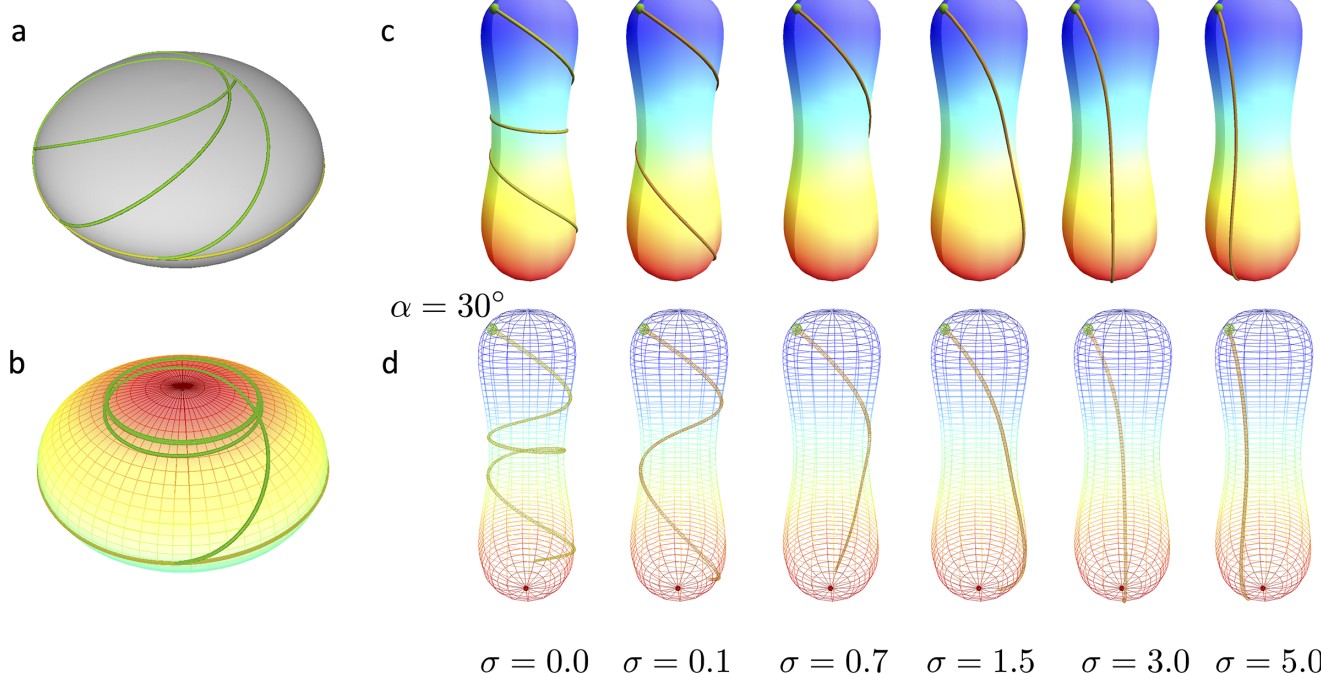

**Figure 18.** Deflection of geodesic trajectories using surface vector fields. (a) and (b) Tropism on an ellipsoid of revolution. (a) No field: the trajectory (green) is a geodesic starting at the equator and heading east, bending 30 degrees north at the origin. (b) Presence of a scalar field (red = high, yellow = low values). The trajectory, with identical initial conditions, converges to a circle in the north region. (c) and (d) Simulated tip-growing filament trajectories on pin-shaped structure with a scalar field at the surface (colour gradient from red (high) to dark blue (low values)). (c) Left-most: Geodesic trajectory (growth not interfering with the scalar field). To the right: effect of tropism resulting of an interaction with the scalar field. Geodesics are deflected in the direction of the gradient of the scalar field, with an increasing intensity $\sigma$ from 0 to 5. (d) The wireframe structures in the bottom row show whole trajectories.

construction process is depicted in Figure 19a, is embedded in an initially planar surface that will progressively take the form of an egg-box. This can be modelled by allowing the parameters of a parametric surface, here a NURBS patch $x^i(u,v,\lambda(t))$, where $\lambda(t)$ represents the surface parameters at time $t$, to change smoothly

in time, and to consider that the uv-coordinates correspond to coordinates of material points. We can observe that the deformation of the Sierpinski carpet does not change its topological structure, and only its geometry is smoothly affected, reflecting the smooth deformation of the embedding surface, Figure 19b.

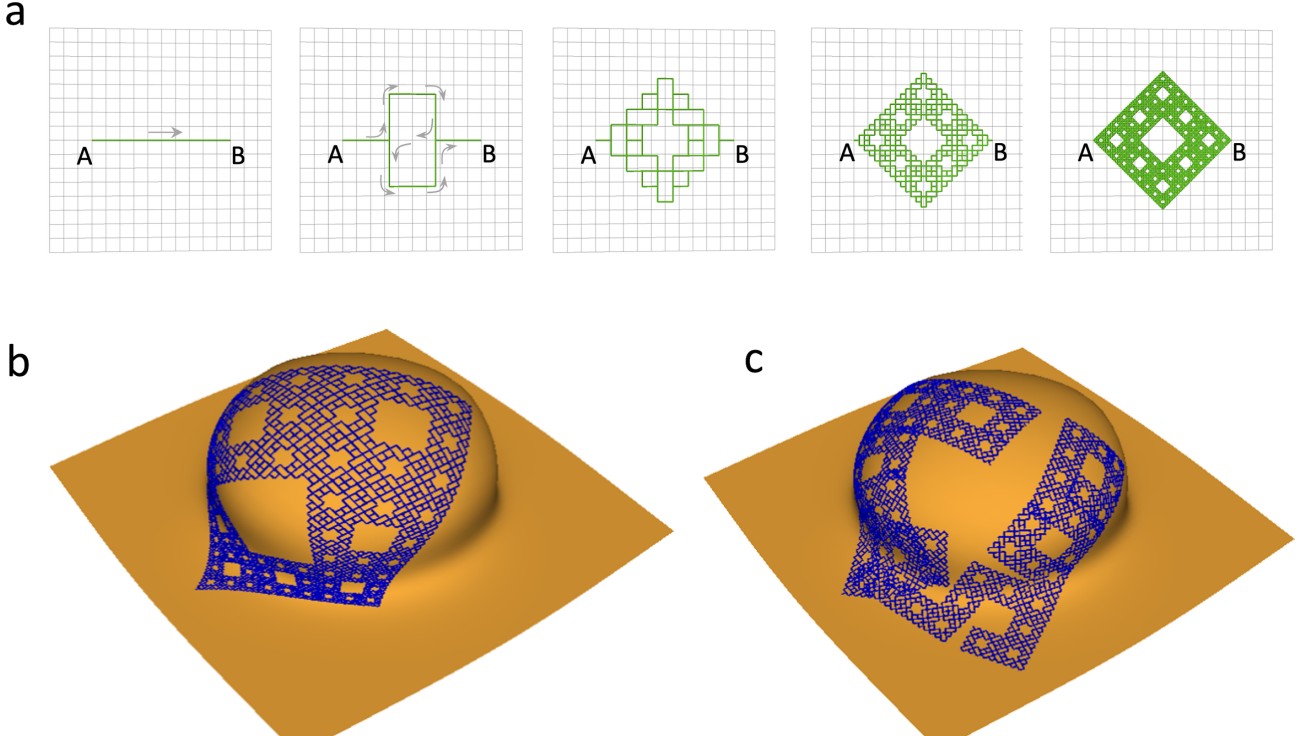

**Figure 19.** Feedback of surface dynamics on forms constructed at the surface. (a) Steps of the construction of a Sierpinski carpet in the Euclidean plane corresponding constructed with an L-system. At each step, the pre-fractal form is obtained by the trajectory of a turtle moving from A to B along convoluted paths. At scale 0 (left), the form is approximated by a simple segment. The turtle draws this segment by going straight from point A to point B. Then, at scale 1, this segment is refined into eight smaller segments of length 1/3 of the original segment length each, as illustrated on the next diagram. The turtle draws this pattern by following the trajectory indicated by the grey arrows. The refinement process then continues at higher scales by decomposing further the segments into smaller segments using the same refinement rule. The form obtained by increasing the scale is called a pre-fractal and contains an increasing number of details at finer and finer resolutions. At every scale, the form is obtained by a single trajectory of the turtle moving from A and to B with increasingly convoluted paths. (b) Convected Sierpinski carpet at a reference scale (indirect interpretation). (c) Floating Sierpinski carpet (direct interpretation, all curves are geodesics): due to holonomy, its topology is not preserved.

```
1  Axiom:
2      nproduce SetSpace(shape(t=0)) InitTurtle([0.3,0.4,1,0])
3      nproduce _(0.05);(2)+(180)f(5)-(180)Seg(0.5,0)
4
5  derivation length: 5
6  production:
7  # Evolving embedding shape
8  SetSpace(cshape) --> SetSpace(shape(t))
9
10 Seg(x, depth):
11     # The fractal shape is decomposed until a maximum depth
12     if depth < MAX_DEPTH::
13         produce Seg(x/3.0, depth+1)+(90)Seg(x/3.0, depth+1)-(90)Seg(x/3.0, depth+1)
           -(90)Seg(x/3.0, depth+1)-(90)Seg(x/3.0, depth+1)+(90)Seg(x/3.0, depth+1)+(90)
           Seg(x/3.0, depth+1)+(90)Seg(x/3.0, depth+1)-(90)Seg(x/3.0, depth+1)
14     elif CONVECTED :
15         # if convected it is replaced by a StaticF to be attached to the surface
16         produce StaticF(x)
17     else:
18         # else it will produce new path on the deformed surface at each step
19         produce F(x)
```

**Listing 16.** Convected versus floating (see Figure 19).

In Riemannian L-systems, this form convection can be achieved using indirect interpretation of L-strings (see Section 3.4). In this mode, the form is not directly drawn as a sequence of geodesics on the surface. Rather, it is drawn in the uv-parameter space and then projected on the surface, reminiscent of texture mapping in computer graphics (e.g., Foley et al., 1996). In this way, the form drawn in the uv-coordinates is that of Figure 19a. The line segments projected on the surface are in general not geodesics, but the projection preserves material points neighbourhood. As a result, the form on the surface appears simply to be convected by the smooth surface material deformation that preserves its integrity and topology. To keep track of the uv-coordinates of a previous

drawing, we introduced a primitive `StaticF` that behaves like a classical forward instruction `F(x)`, i.e., it computes a geodesic of length $x$ from the current position and heading of the turtle. However, the first time it is called, it memorises the uv-positions of the computed geodesic. Then, at next derivation steps, on further calls, it will not recompute a geodesic on the modified surface. Instead it will use the previously cached geodesic information. List. 16 illustrates how to switch between floating and convected shapes using `StaticF` into Riemannian L-systems.

On the other hand, forms lying on a surface may be completely independent of the movement of material points at the surface, or may even be defined in the absence of material points. For this, the construction procedure must be purely *geometric* and must not depend on the actual surface parameterisation, if any. In this case, the form is said to have an *intrinsic* geometry on the surface (it does not depend on the embedding of the surface in a space of higher dimension, nor on the coordinate system used on the surface). Due to holonomy, during the surface evolution, the geometry of an intrinsic form is affected by changes in surface curvature. As a consequence, the form appears to be *floating* on the surface (Figure 19c and d), with varying degrees of geometric and topological distortions through time. By providing general geometric primitives to draw in curved spaces, Riemannian L-systems naturally produce intrinsic geometric forms. As fractals contain geometric details at different scales, they provide a natural way to probe the impact of changes in surface curvature in time within a range of scales. In Figure 19c and d, one can observe that only the scales that are commensurate with the local radius of curvature of the surface are significantly impacted. At smaller scales, the surface can

be assimilated to a plane and the corresponding small details are hardly affected. Supplementary Movie 1 shows how the progressive change of the embedding surface curvature dynamically affects the different regions of the floating fractal form.

**5.2.2. Feedback of the embedding surface growth on the form.** In the previous example, the deformation of the surface alters the constructed shape, while the construction process itself remains unchanged. However, the deformation of the surface could in principle also feedback on the construction process itself. Such a feedback may be caused for example by changes in physical fields living on the surface due to the surface growth (e.g., dilution of molecule concentrations, relaxation of mechanical stresses, etc.). They can also be induced by the very change of surface geometry. In biology for instance, patterns at the surface of an organ are commonly refined in response to growth. In plants, cells grow at the surface of an organ and divide as soon as their size reaches a certain threshold (Jones et al., 2019). Likewise, in leaves, new veinlets appear as the leaf blade expands (Sawchuk et al., 2013). A first approach of this type was proposed in the context of simple planar (affine) deformation of a material 2-D flat space (Prusinkiewicz et al., 2014). Here, we consider 3-D substrate deformations, where the curvature of object may also change in time. Let us explore how such feedback mechanisms can be modelled using Riemannian L-systems.

We first consider the case of forms made of segments convected deformed by growth, Figure 20. The convection deforms the segments and we assume that as soon as a segment reaches a maximum length, it gets replaced by a series of smaller segments using

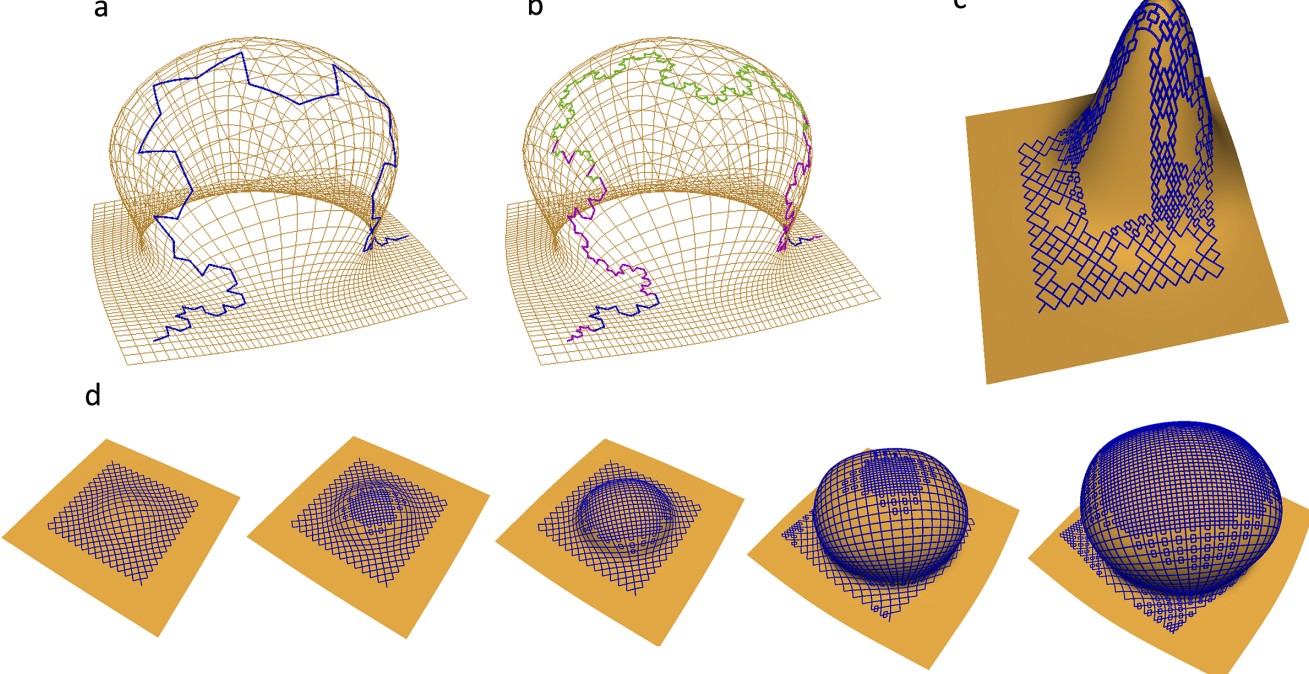

**Figure 20.** Feedback of surface growth on patterns living at the surface: example of subdivisions (a) and (b) von Koch prefractal curve initially developed at level 3 (dark blue segments) gets deformed by the growth of a flat surface, without subdivision feedback (a) and with feedback (segment are coloured purple (detail level 4) and green (detail level 5) (b). Note additional, non homogeneous fractal details on (b) due to the subdivision of segments that reached a length threshold during growth. (c) subdivision feedback on a Sierpinski carpet. Details are added only in places where the initial motif has been significantly stretched by surface growth. (d) Deformation of a Peano prefractal curve. The Peano curve is developed up to level 3 on a flat surface starting to grow out (left). As soon as segments are stretched above a given threshold, they divide into nine smaller segments (Peano subdivision rule, that is derived from the Sierpinski carpet rule illustrated on Figure 19a by tracing the fifth (middle) segment instead of skipping it). Due to growth, waves of subdivisions can be observed at the surface. The rightmost image shows the result after two rounds of divisions on the topmost part of the growing surface, see Supplementary Movie 2.

some refinement rule (i.e., von Koch, Sierpinski carpet and Peano rules, respectively, for Figure 20a–d). Segment decomposition is made indirectly in the parameter space before curve segments are projected onto the surface. This ensures to keep curves continuous. However, the segments drawn in the parameter space are not geodesics. Note that by contrast, a form freely floating on a surface (i.e., not convected) would not be affected so drastically in its size, as segments are not 'attached' to the surface and can freely accommodate surface deformations, while keeping their angles and lengths unchanged. These forms would therefore only be able to sense local changes of curvature of the embedding surface, but not its growth *per se*.

### 5.2.3. Dynamic surfaces with boundary growth.
The previous fractal forms capture in essence what can be called *ubiquitous growth*: during organism development, growth may occur everywhere in the organism by local subdivisions and or expansion of existing atomic regions. This is observed in various animal or plant tissues at cellular level. However, other types of growth may be identified at organ or individual level. At macroscopic scales for instance, plants develop their branching structures by apical or edge growth. Rather than subdividing already existing structures (axes,

leaf blades, etc.), plants add new components at the extremities or boundaries of these structures if space allows. Such a *boundary growth* has efficiently been modelled with L-systems in 3-D Euclidean spaces (e.g., Godin et al., 2005; Prusinkiewicz & Lindenmayer, 1990; Prusinkiewicz et al., 2018). Riemmanian L-system generalises this approach to the modelling of boundary growth on various types of curved embedding spaces.

This may be illustrated by the modelling of leaf growth. The young leaf of the kidney fern (*Hymenophyllum nephrophyllum*), Figure 21a, has a curved and thin blade that embeds a conspicuous dichotomic venation pattern. As the fern grows, this fractal-like pattern gets branching over and over, suggesting that the growth mainly occurs at the apical edge of the fern and that the venation pattern is progressively built bottom-up as the edge growth progresses.

Here, we model this process using a couple of assumptions. First, we assume that the blade is a growing surface that extends only at the periphery (rim). For this, a NURBS patch is used to model the surface with its growth begin emulated by extending through progressive increase of its parameterisation range. At an initial time $t$, a number of $N(t)$ of target points are regularly distributed over the blade rim perimeter, from which originates $N(t)$ veins. In a

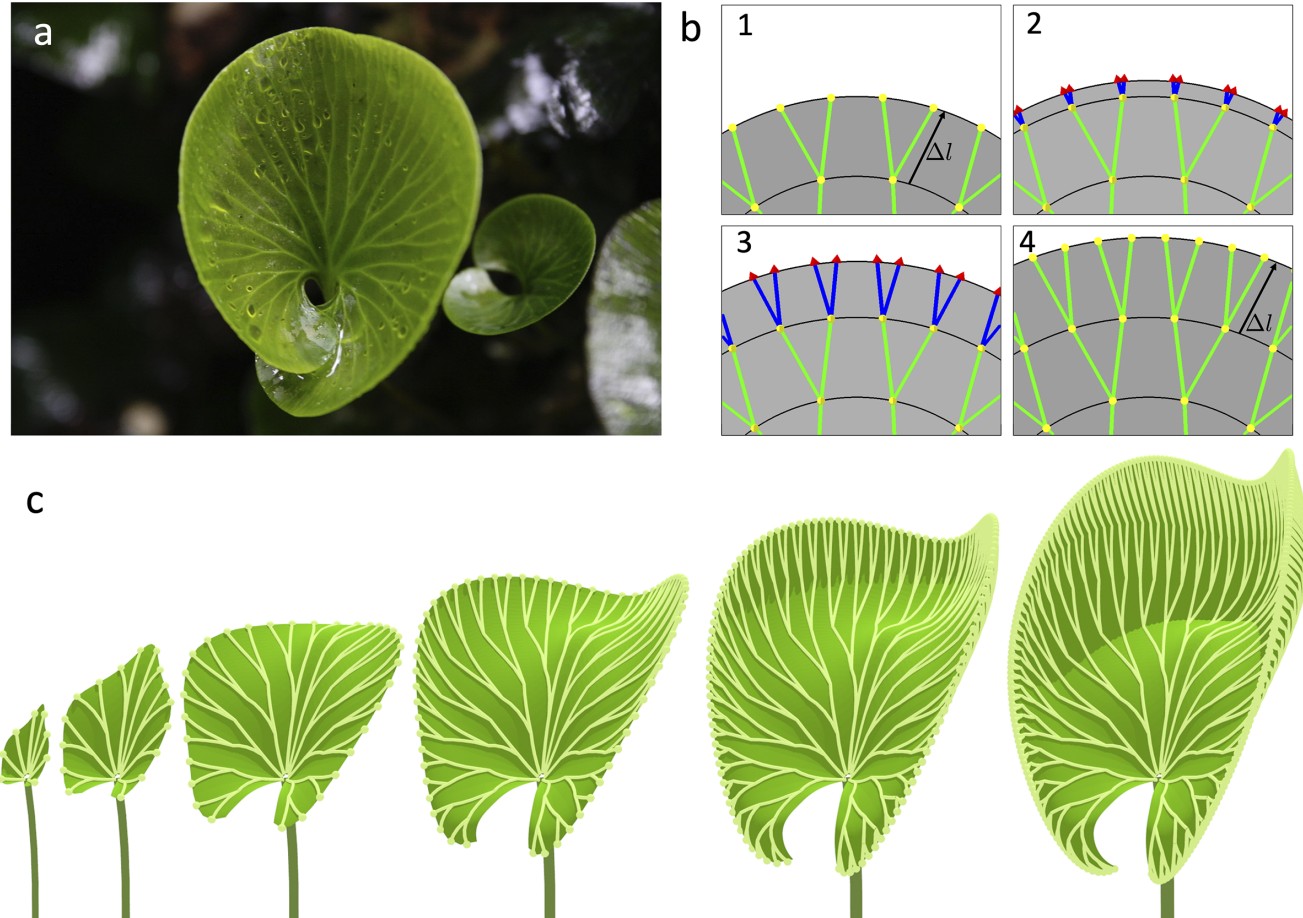

**Figure 21.** Kidney fern model. (a) *Hymenophyllum nephrophyllum* (Kidney fern). (b) Steps of the growth algorithm. The algorithm proceeds by growing recursively the fern blade (grey) and its vasculature (green). At some time, a binary structure of veins has already been constructed (green) with branching points indicated in yellow. 1. Initial step of the recursion: the last blade growth band (between the last two black lines), containing the active points (yellow) regularly positioned on the rim line (black), reaches a size $\Delta L$. 2. This triggers the formation of a new generation of active points (red arrowheads) on a new rim line at the next time step, with twice as many points as in the previous one (yellow points). Geodesic paths (blue lines) are constructed from the last row of branching points (yellow) towards the new target points (the direction of the red arrowheads indicate the direction of the geodesic construction). This produces binary branchings. 3. The blade continues to grow. Geodesic lines (blue) are recomputed to adapt to the rim growth and changing geometry. 4. Final step of the iteration: a new blade band has been constructed with new branching segments ending at regularly spaced points and the recursion can proceed similarly for a new blade band. (c) Consecutive stages of the simulated developmental model.

small amount of time $\Delta t$, the rim grows and convects the $N(t)$ source points with it. As it extends, the $N(t)$ source points get further away from each other. At a given distance from the original source points, each source point is divided into two new source points on the rim, leading to a double number of source points $N(t + \Delta t)$. This process can be repeated over time.

Our simulation based on Riemaniann L-systems emulates the successive stages of vein ramification over the growing surface. At each time step $t$, $N(t)$ sources points are uniformly positioned along the rim $R(t)$ at equidistant intervals (as depicted in Figure 21b). Sequential pairs of these source points are associated with corresponding points on the preceding rim $R(t-\Delta t)$. The geodesic paths are determined using the `RiemannLineTo` command. As the preceding part of the structure has become fixed with respect to the existing surface at previous time step, the `StaticF` command can be used to anchor it and apply the surface deformation. Consequently, the shape formation integrates both convected and floating mechanisms. The different steps produced by the simulation, delineating the various stages of vein ramification, are given in Figure 21c, see also Supplementary Movie 3.

## 6. L-systems in abstract Riemannian spaces

Surfaces embedded in $\mathbb{R}^3$ are a particular example of the general concept of *Riemannian manifold* (e.g., Boothby, 2003; Carroll, 2014). Intuitively, a Riemannian manifold of dimension $n$ is an abstract space that locally looks everywhere like the Euclidean space $\mathbb{R}^n$ for some $n$ (i.e., a space of points where one can locally measure distances between points and angles between directions). According to this definition, Euclidean spaces $\mathbb{R}^n$ themselves are trivial manifolds. A smooth surface in $\mathbb{R}^3$ is another example of a 2-D manifold as it locally looks like a plane around each surface point. Likewise, a curve on this surface, in the plane or in $\mathbb{R}^3$, is a 1-D manifold being locally akin to a straight line.

What is specific of these examples, is that they consider Riemannian manifolds embedded in higher-dimensional Euclidean spaces. This makes it possible to exploit the metric of the embedding space to define locally their own metric to measure distances and angles. However, general Riemannian manifolds can be defined on spaces that are not necessarily embedded in higher-dimensional spaces. For this, it suffices to define a smooth metric, i.e., a way to measure distances and angles at every point, directly on the manifold structure. General relativity (GR) for instance postulates that the space-time in which we live is such an abstract Riemannian manifold with four dimensions, not embedded in a higher-dimensional space and whose metric is locally imposed by the spatial distribution of matter through Einstein's field equations, e.g. (Carroll, 2014; D'Inverno, 1992).

In this section, we explore the possibility to construct forms using L-systems in such abstract Riemannian spaces. For this, we will be led to find ways to define metrics on these spaces, and revisit the way key differential geometric quantities are evaluated in order to compute geodesics and construct forms in these spaces.

### 6.1. Riemannian manifolds

We briefly introduce here essential definitions and notations classically used to manipulate manifolds. For a more extensive introduction to Riemannian manifolds, readers can refer to, e.g., Boothby (2003), Carroll (2014), D'Inverno (1992), Frankel (2017), Rouvière (2016), and Schutz (1980).

**6.1.1. Definition.** The key feature that a manifold locally looks like $\mathbb{R}^n$ leads naturally to (loosely) define a manifold $\mathcal{M}$ as a set of points that locally possesses coordinates $(u^1, u^2, \cdots, u^n) = u^\alpha, \alpha = 1, n$ that vary smoothly over the manifold, with values in a connected region of $\mathbb{R}^n$[1], e.g., (D'Inverno, 1992). A local vector basis, called *covariant basis*, can be attached to each point $P$ of coordinates $u^\alpha$ of $\mathcal{M}$. Vectors of the covariant basis are usually denoted $\partial_\alpha$[2]). In two dimensions, a picture of the coordinate lines and the associated covariant basis is given by Figure 2a where the basis vectors $e_1, e_2$ should be substituted by $\partial_1, \partial_2$. A *Riemannian manifold* is a manifold equipped with a (positive definite) metric (e.g., Carroll, 2014). As in this abstract case, the metric cannot be inherited from a higher-dimensional space, one needs to define a proper and smoothly varying metric, i.e., scalar product, at every point $P$ of the manifold:

$$g_{\alpha\beta} = <\partial_\alpha, \partial_\beta>. \tag{54}$$

Note that the dependency of the metric on the point $P$ is not indicated here. The metric allows to define how lengths should be measured locally around each point. Consider for example a small segment in the manifold corresponding to a variation $du^\alpha$ of coordinates. Then, the length $ds$ of this small segment is defined by the metric:

$$ds^2 = g_{\alpha\beta} du^\alpha du^\beta. \tag{55}$$

It is important to note that without a metric, the notion of distance is not defined and one cannot rely on the coordinates for that. Two points with close coordinates may actually, depending on the metric, be very far away from each other in terms of distance according to Eq. (55).

**6.1.2. Connections and derivatives.** Interestingly, however, if a metric measures and compares vectors in the same tangent plane, it does not give *per se* any means to compare vectors living in different tangent planes. For this, one needs to introduce the additional notion of *connection* (e.g., Carroll, 2014, p. 95). A connection defines the rate at which the difference of coordinates of two vectors, considered parallel in neighbouring tangent spaces, vary as one tangent space gets closer to the other one and the coordinate difference tends to 0. It can be defined by a series of coefficients, $\Gamma^\gamma_{\alpha\beta}$, called *connection coefficients*, specifying how to compute the coordinates of vectors parallel to a given vector in its neighbourhood (D'Inverno, 1992, p. 72). Let $\mathbf{X} = X^\alpha \partial_\alpha$ be a vector field evaluated at $P$ of coordinates $u (= u^\gamma)$. We define the parallel vector $\bar{\mathbf{X}} = \bar{X}^\alpha \partial_\alpha$ at a neighbouring point of coordinates $u + \delta u$ as

$$\bar{\mathbf{X}}(u + \delta u) = \mathbf{X}(u) - \Gamma^\alpha_{\beta\gamma} X^\beta \delta u^\gamma \partial_\alpha. \tag{56}$$

This expression defines how the components of a given vector must change to stay parallel to the original vector when one

---

[1]Note that it is in general not possible to use a single coordinate system to cover a manifold in this way and, that then several regions of the manifold can be covered with different coordinate systems, that make up *an atlas*. A manifold is then defined in general as a set endowed with an atlas of coordinate systems (Boothby, 2003; Carroll, 2014). However, we do not consider the full definition hereafter.

[2]In abstract manifolds, tangent spaces cannot be represented physically and vectors in the tangent plane are defined by differential operators that can operate at each point $P$ on scalar functions $f(P)$ defined on the manifold. The notation $\partial_\alpha = \frac{\partial}{\partial u^\alpha}$ comes from the fact that basis vectors actually correspond to partial directional derivatives operators along each coordinate line. Together they form a basis of the local tangent plane $T_P\mathcal{M}$ (e.g., Boothby, 2003).

moves on the manifold. Here again, these connection coefficients should be understood as depending smoothly on the point $P$ in the manifold. Using connections, neighbouring vectors can thus be compared by first parallel transporting one vector in the tangent plane of the other (Eq. (56)), and then comparing them. This parallel transport thus makes it possible to define a notion of derivative on the manifold, called *covariant derivative*. Covariant derivatives quantify the rates of variations of quantities living on the manifold, such as vectors and tensors, in specific directions, independently of the chosen coordinate system. This generalises the notion of directional derivative. If $\mathbf{X} = X^\alpha \partial_\alpha$ denotes a vector field, its covariant derivative $\nabla_\alpha \mathbf{X}$ in the direction of the basis vector $\partial_\alpha$ at $P$ of coordinate $u$ is defined by D'Inverno (1992):

$$\nabla_\alpha \mathbf{X} = \lim_{\delta u^\alpha \to 0} \frac{1}{\delta u^\alpha} \left( \mathbf{X}(u + \delta u) - \bar{\mathbf{X}}(u + \delta u) \right), \qquad (57)$$

which leads for abstract Riemannian spaces to an expression similar to Eq. (24) for surfaces embedded in $\mathbb{R}^3$:

$$\nabla_\alpha \mathbf{X} = (\partial_\alpha X^\beta + X^\gamma \Gamma^\beta_{\alpha\gamma}) \partial_\beta. \qquad (58)$$

This expression can be generalised by linearity to the derivative of vector $\mathbf{X}$ in any direction $\mathbf{Y} = Y^\beta \partial_\beta$ in the same tangent plane:

$$\nabla_{\mathbf{Y}} \mathbf{X} = Y^\beta \nabla_\beta \mathbf{X}. \qquad (59)$$

**6.1.3. Choice of a specific connection compatible with the metric.** The situation is thus as follows: on the one hand we have a metric, defined by coefficients $g_{\alpha\beta}$, that specifies how to compare vectors in common tangent spaces. On the other hand, we have connection coefficients that correspond to selecting a notion of parallelism on the manifold, i.e., what it means for vectors living in different tangent planes to be parallel. These two notions can be defined independently. However, they interact and it is possible to define connection coefficients so that length and angles of vectors parallel transported along the same curve remains constant. This condition, called *metric compatibility*, formally links the two notions $\nabla$ and $g$ by imposing:

$$\nabla g = 0. \qquad (60)$$

In addition, if it assumes that the connection coefficients are symmetric in their lower indexes, $\Gamma^\gamma_{\alpha\beta} = \Gamma^\gamma_{\beta\alpha}$ (the connection is said to be *torsion free*), then it can be shown that there exists a unique connection, called the *Levi-Civita connection*, that is both metric compatible and torsion free. The connection coefficients, are then called Christoffel symbols, and are completely determined by the metric:

$$\Gamma^\gamma_{\alpha\beta} = \frac{1}{2} g^{\gamma\lambda} (\partial_\beta g_{\lambda\alpha} + \partial_\alpha g_{\lambda\beta} - \partial_\lambda g_{\alpha\beta}) \qquad (61)$$

This formula is remarkable in that it shows that the Christoffel symbols can be derived by using only the metric and its first derivatives. Applied to surfaces as a particular case of Riemannian spaces, it shows that the Christoffel symbols can be computed by using only intrinsic properties of the surface (while we actually used the (non-intrinsic) Euclidean scalar product previously in Eq. (21)).

**6.1.4. Summary.** Altogether, once a metric is defined on a manifold, it becomes a Riemannian manifold. However, a suitable notion of differentiability still needs to be added so that spatial variation rates of geometric or physical quantities living on the manifold can be computed independently of the coordinate system. This is called covariant derivative and requires the definition of the additional notion of 'connection' between tangent spaces. Interestingly, among all the possible options, the definition of the manifold metric already induces a unique one, the Levi-Civita connection, that has the very natural property to preserve angles and length of vectors parallel transported along the same curve. For this reason, the Levi-Civita connection is often used to define covariant differentiability on manifolds[3].

**6.1.5. Consequences for Riemannian L-systems.** The extension of Riemannian L-systems defined on curved surfaces presented in Sections 3–5 to abstract Riemannian spaces relies on this more general intrinsic expression of the Christoffel symbols. In particular, moving the turtle forward by a distance $l$ can still be computed as the solution of an IVP defined by Eq. (34) where the Christoffel symbols are now computed using Eq. (61).

Turning in abstract Riemannian space also needs some adjustment. Indeed, Eq. (50) used to take place in the surface tangent plane and use the surface normal to define a rotation axis. In abstract Riemannian space, one no longer can rely on this strategy, as no 'outer' space exists to define such a normal. The definition of the rotation axes must thus keep completely intrinsic to the abstract space. In 2-D, one does not need such a rotation axis, as a unique point is necessary to define the centre of rotation. Rotations must thus be made in 2-D with the precaution as before of ortho-normalizing the basis beforehand. In 3-D, the turtle may turn naturally according to its local reference frame $(\mathbf{H}, \mathbf{L}, \mathbf{U})$ that, respectively, locally defined 3 axes of rotations in the curved space. Here again, the local covariant basis carried by the turtle has to be ortho-normalised before applying the corresponding rotations, according to Eq. (50) with updated axes of rotation.

To represent the forms constructed in these abstract spaces, we will finally display the $u^\alpha$ coordinates of the abstract Riemannian spaces in our 2-D or 3-D Euclidean world. However, while we apparently observe a 2-D or 3-D world, the Euclidean flat metric at each point of this space is replaced by a metric $g_{\alpha\beta}$ defined by the user that locally distorts space. While the turtle interprets the Lstrings, it produces uv-parameters corresponding to the form description in the curved abstract space coordinates.

The following sections give examples of how to program shapes in 2-D abstract Riemannian spaces using Riemannian L-systems.

### 6.2. Examples in 2-D abstract Riemannian spaces

**6.2.1. Defining an abstract Riemannian space and drawing geodesics.** To illustrate the modelling of abstract Riemmanian spaces with Riemannian L-systems, we will first use the Beltrami–Poincaré half plane. Historically, this space has played a fundamental role in the development of non-Euclidean geometry (e.g. Needham, 2021). Many of its properties have been thoroughly studied, and we will make use of this knowledge to test our Riemannian L-system algorithms and constructs. In particular, this space has a constant negative Gaussian curvature $\kappa_G = -1$ (hyperbolic space).

At first sight, the Beltrami–Poincaré half plane looks like a Euclidean half plane with Cartesian coordinates $u^\alpha$, Figure 22a and c. However, the metric is non-Euclidean and is defined at each point as

$$ds^2 = \frac{1}{(u^2)^2} \delta_{\alpha\beta} du^\alpha du^\beta, \qquad (62)$$

---

[3]The Levi-Civita connection is for instance the one selected by default in various developments in GR (Carroll, 2014), unless otherwise specified.

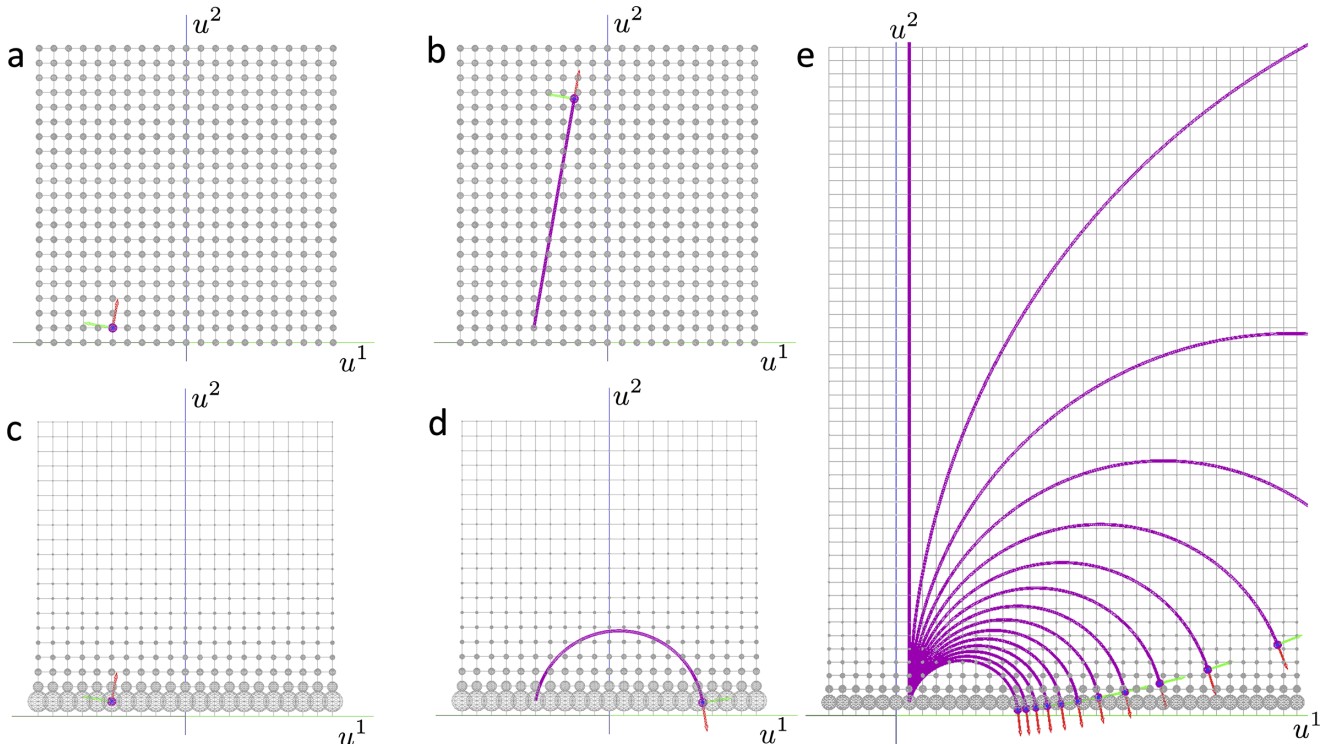

**Figure 22.** Beltrami–Poincaré half plane. (a) and (b) Euclidean half plane with uniform isotropic metric, represented by small discs with equal size at every point of the space (representing themselves $ds^2$). A geodesic is a straight line (b). (c) and (d) Beltrami–Poincaré half plane. The metric varies vertically ($u^2$ coordinate), and the geodesic generated at the same point and with the same orientation as in (a) and (b) is a portion of a circle (d). (e) In the Beltrami–Poincaré half plane, all geodesics are circles centred on the $u^1$-axis. Geodesic starting with a vertical orientation are vertical lines (degenerated circles).

where $\delta$ is the Kronecker delta, $\delta_{\alpha\beta} = 1$ if $\alpha = \beta$, and 0 otherwise. The metric spatial distribution is represented in Figure 22c at every point of coordinates $u^\alpha$ by a small disc representing the $ds^2$ for small local variations $(du^1, du^2)$ of the coordinates such that $(du^1)^2 + (du^2)^2 = \epsilon^2$, epsilon being a small real constant. One can observe that as Eq. (62) specifies, the circles gets bigger as points approach the $u^1$-axis. This intuitively means that a line crossing this region will be much longer than a line with identical coordinate variations crossing a region further up in the space (i.e., with higher $u^2$ coordinates). Note that a dual representation of the metric, where $ds$ would be considered as a constant at every point, is possible and would provide increasing circle size for increasing values of $du^2$, reflecting the size of the coordinate space to cross for a constant amount of $ds$ at different locations of the Beltrami–Poincaré plane.

A geodesic in the Euclidean space is a straight line, Figure 22b. In the Beltrami–Poincaré half plane, the geodesic starting at the same coordinates and in the same direction is a portion of a circle whose centre is located on the $u^1$-axis, Figure 22d. It can be shown that any geodesic in this hyperbolic plane of constant negative curvature is either a circle centred on the $u^1$-axis or a vertical line, as is illustrated in Figure 22e for different geodesics starting at the same point with different orientations and of the same length (the four uppermost geodesics go beyond the scope of the visible grid).

To compute these geodesics with Riemannian L-systems is no more difficult than drawing straight lines in a Euclidean space, see List. 17. First, the metric is defined by functions $g_{\alpha\beta}$ of the coordinates (here called $u, v$ instead of $u^1, u^2$), lines 2–4. The metric is then assembled as a dictionary of functions (the metric is symmetric and then $g_{21}$ need not be defined), line 5. A special type of space

```
1  lunit = 1                                    # unit of length
2  def g11(u,v,*args): return 1./v**2
3  def g12(u,v,*args): return 0.                # g12 == g21
4  def g22(u,v,*args): return 1./v**2
5  metric = {'g11':g11,'g12':g12,'g22':g22}    # metric matrix
6  Axiom:
7    nproduce SetSpace(RiemannianSpace2D(**metric,umin=-1.0,umax=1.0,vmin=0.0,vmax=2))
8    nproduce InitTurtle([-0.5,0.10,0,1])       # initial position and orientation
9    nproduce -(10)                             # reorient the turtle
10   nproduce A(0)
11
12 derivation length: N                         # Make N derivations
13 production:
14 A(n):
15   nproduce F(lunit)A(n+1)                     # geodesic segments of 1 unit of length
```

**Listing 17.** Geodesics in the Beltrami–Poincaré half plane (see Figure 22d).

object, `RiemannianSpace2D`, has been defined to represent 2-D abstract Riemannian spaces. At its construction, this object must be given the dictionary of metric functions (line 7). Then after positioning and orienting the turtle (lines 8 and 9), a series of $N$ segments of unit length is drawn, giving rise to a geodesic of length `N lunit` (lines 12, 14 and 15, here N = 5).

### 6.2.2. Immersion of an intrinsic geometry in an abstract space.

A form specified using L-systems is usually defined in an intrinsic way as only lengths and relative angles between consecutive components are used in its description. The turtle interpretation then immerses this intrinsic description in a specific space, leading to a form with explicit geometry.

In classical L-systems, the turtle embeds the L-system's intrinsic forms in the Euclidean 3-D space. The form implicit geometry is thus always associated, in a one-to-one way, with a default explicit geometry. In Riemannian L-systems, we can use abstract spaces to embed intrinsic geometries specified by L-systems into various types of curved spaces which will associate different explicit geometries to the original form. While preserving the intrinsic shape, this new embedding results in different explicit geometric shapes, reflecting the curvature the embedding spaces.

To illustrate this fact, in Figure 23, we use a form consisting of a fractal von Koch flake, and move it progressively to the right through an abstract curved space. The space metric linearly depends on the distance to the central point of coordinates $(u_0^1, u_0^2)$ (small reference dot close to each form on figure Figure 23): the metric is small close to the point and increases with the distance to the central point.

$$ds^2 = r^2 \delta_{\alpha\beta} du^\alpha du^\beta. \tag{63}$$

with

$$r^2 = (u^\alpha - u_0^\alpha)(u^\beta - u_0^\beta)\delta_{\alpha\beta}. \tag{64}$$

Due to curvature spatial inhomogeneity, the geometric embedding varies continuously as a function of the position of the flake in space (Figure 23), Supplementary Movie 4. This suggests that abstract curved spaces could be used to model the deformation of natural objects. We explore this idea in the next section by modelling different types of tropisms on plants.

### 6.2.3. Modelling growth within heterogeneous substrates, tropism.

Various systems, such as roots, veins or pollen tubes, may grow within substrates that are not uniform (i.e., not homogeneous or not isotropic or both) in their physical or chemical contents. Plant shoots for instance, growing within plant canopies, usually progress in a non-uniform light environment and are also subject to the anisotropy of the gravity field. Such interactions are usually mediated by chemical or bio-physical forces, that affect the direction and/or size of the growing elements. Plants respond in general to these forces with a high degree of plasticity, which is responsible for example of tropism phenomena, such as gravitropism (response to gravity), phototropism (response to light), thigmotropism (response to touch or contact), etc.

Models of growth in non-uniform media usually rely on a mechanistic description of the system's interaction with the substrate. These models take various forms depending on the spatial and temporal scales considered. Recently, for example, multiscale models of tropisms have been proposed (Moulia et al., 2022; Moulton et al., 2020) and make it possible to integrate various types of tropisms in unified approaches. The models rely on the expression of differential growth at different scales controlled by the sensing of different environmental signals or fields (Jensen, 2021). In this approach, forces of different nature are affecting the plant components during growth, which in turn affect the growth intensity or directions of the branching system itself.

In contrast, we consider here the possibility to model branching system growth (roots, veins and aerial branching system) within a non-uniform substrate (soil, leaf blade and aerial space) in an effective rather than in a mechanistic manner. For this, instead of describing how the substrate non-uniformity impacts the growth through force interaction, we seek to describe a system developing straight axes in a substrate curved by the presence of non-uniformity. This non-uniformity modifies the substrate's metric, which becomes non-flat, which in turn impacts the development of the growing components, that follows substrate geodesics instead of growing in straight lines.

Let us consider a simple branching structure growing in a flat Euclidean space (Figure 24a). This tree structure is made of straight branches that are represented in the figure as geodesics of the Euclidean plane, i.e., straight lines. At each point of the Euclidean space, the metric is Euclidean and in Cartesian coordinates, this can be expressed as

$$ds^2 = \delta_{\alpha\beta} du^\alpha du^\beta. \tag{65}$$

Now, let us illustrate this with a simple branching system model, in which the metric is a function of points in spaces. We wish to understand how this affects the form being constructed in this space. In Figure 24b, the metric was changed to that of the Beltrami–Poincaré half plane introduced above (Eq. (62)). We observe that the 'straight' branches now become portions of circles as they correspond to straightest lines in the curved space, i.e., geodesics of the Beltrami–Poincaré half plane. This mimics (here in a caricatured way) the bending of the branches under the effect of gravity. We observe also that as a side effect the metric

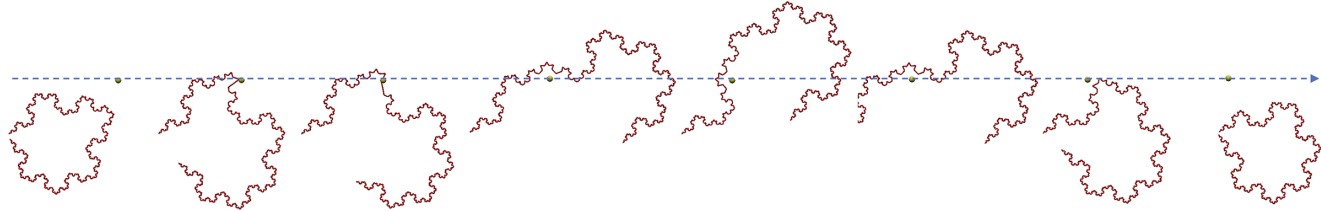

**Figure 23.** Different geometric embeddings of the same intrinsic von Koch flake curve. Sequence of snapshots showing a von Koch flake moving progressively from left to right (the fixed point in each snapshot serves as a position reference) in an abstract 2-D space where the metric linearly depends on the distance at the origin (small yellow dot). During this move, the flake is deformed by the metric. Geodesics forming the segments close to the origin tend to be strongly curved, thus deforming the entire flake. At the end of the sequence, the metrics becomes more homogeneous over the entire flake, which is less and less distorted. The dotted line indicate the temporal progression of the snapshots during the move.

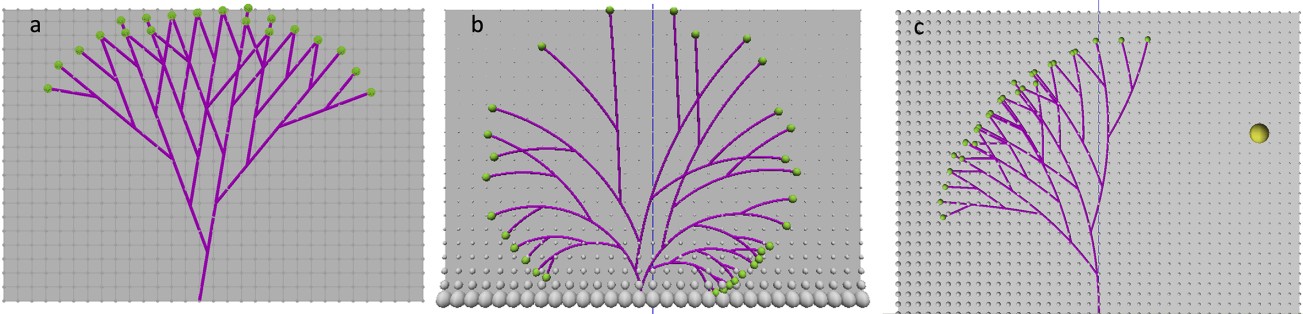

**Figure 24.** Modelling tropism using abstract Riemannian spaces. (a) Simple branching system in a Euclidean plane. (b) Same branching system interpreted in a Beltrami–Poincaré half plane. This simulates a form of 'gravity attraction' of the branches. (c) Abstract Riemannian space with point-like source of metric distorsion, simulating shadow-avoiding behaviour of trees.

'compresses' or 'stretches' the length of the branches depending on the intensity of the $ds^2$: curves at the bottom cross high values of $ds^2$, represented by the size of the local grey circles, and are much compressed. Curves at the top on the contrary cross small $ds^2$ values and get stretched. This is due to the fact that for a given length $L$ to draw, more increments of coordinates $du^\alpha$ will be needed in the upper part of the half plane than in the lower part to achieve the same length $L$. Supplementary Movie 5.1 shows the action of a continuous change of the metric (augmenting the strength of the gravity attraction) on the tree structure and that 'bends' the tree branches downwards.

By changing the spatial distribution of the metric, one can simulate qualitatively different types of tropisms. Figure 24c and Supplementary Movies 5.2 and 5.3 for instance show how a metric can be defined as a field emanating from a particular source point (yellow sphere on the right) according to Eq. (64). The metric is inversely proportional to the distance from the source point. This curves the space in the way that geodesics bend away from the source point as illustrated by the branches of the trees. This simulates a shadow-avoiding tropism, where the source point could represent a source of shadow in space (due to another plant, to a wall, etc.). Branches then bend away from the shadow to seek for more light. Here again, the length of the branch segments is affected as well by the metric, with the branches in areas with small metric being longer than branches in areas with large metric.

These simulations show that modifying the metric spatially may curve the space such that growing forms 'look like' being attracted or repulsed by different type of sources. This bending deformation is coupled with a compression or stretching of the branch segments. Further work must be carried out to explore how these properties can be used to construct effective models of growth in non-uniform substrate, where potentially, similarly to spacetime deformation by the matter it contains in GR, the plant processes themselves could be a source of substrate's curvature that contributes to locally orient fluxes and/or growth (Jaeger et al., 2008).

## 7. Conclusions

In this article, we have presented an extension of the classical formalism of L-systems to parametric Riemannian spaces (2-D surfaces in $\mathbb{R}^3$ and 2-D abstract non-Euclidean spaces). This is made by extending turtle geometry to these curved spaces and by developing high-level language constructs to program form development in these spaces. Both theoretical and applied examples have been presented to illustrate concepts of differential geometry

as well as the potential of this new framework to model realistic phenomena in biology.

Riemannian L-systems provide an intuitive high-level programming language to program the development of a large variety of forms in a wide spectrum of curved spaces. By using turtle geometry and its locality principle, such programming is, most of the time, not more difficult than programming in flat Euclidean spaces as Riemannian spaces are locally Euclidean. All the complexity related to the handling of complex differential geometry algorithms is mostly hidden to the user. It should allow modellers to easily address new modelling problems where the growth of forms takes places in various sorts of curved embedding spaces in either plant or animal systems. For example, the recently identified ability of animal cells to move on a surface according to the local curvature of the supporting surface, called curvotaxis (Pieuchot et al., 2018; Werner et al., 2019), could naturally be modelled using Riemannian L-systems. Finally, Riemannian L-systems also offer interesting new possibilities to learn and teach differential geometry from a natural and easy perspective.

The modelling of growth phenomena in non-uniform substrates using intrinsically curved spaces (here called abstract Riemannian spaces) is a promising avenue for the development of Riemannian L-systems. Curved trajectories of growing forms in abstract Riemannian spaces are reminiscent of trajectories of massive objects in the intrinsically curved spacetime of GR (Carroll, 2014; D'Inverno, 1992). However, both situations are not quite identical. Here, we consider a curved space and not a *curved spacetime* as in *GR*, and the forces that are responsible for plant form growth are both of gravitational and electromagnetic nature, while the curvature of spacetime corresponds to gravitation only in *GR*. As expected, the framework of *GR* thus does not apply as such to the morphogenesis of living forms. On the other hand, the situation has strong similarities with other physical systems. For example, the propagation of light rays in media with varying optical index, in which the rays follow geodesics that bend according to the local changes of optical index (Needham, 2021). This phenomenon is for instance at the origin of astronomical refraction and results in the fact that astronomic objects like the sun for instance appear higher above the horizon than they actually are (Thomas & Joseph, 1996), which itself has connections with *GR* (Hui & Zhu, 2024). By providing computational concepts and tools to manipulate curved spaces and information propagation inside, this approach can also be viewed as a first step towards a concrete implementation of the concept of a general relativistic theory of positional information introduced by Jaeger et al. (2008). While still at an early stage, abstract Riemannian L-systems open up a new possibility of formalizing growth

in non-uniform fields. These effective fields can in principle be generated by the plant environment or by the plant matter itself, for instance to grow or to bend locally, thus making yet another connection with the concept of general relativistic positional information (Jaeger et al., 2008).

Based on the current system, we have shown that a whole new set of biological questions, such as the growth of pollen tubes on papillae, or the joint growth of a leaf blade and its venation network, are becoming easily accessible to modelling. However, Riemannian L-systems could be further extended in several directions. Addressing more complex biological systems would require to integrate gene regulation networks, molecular transport and tissue mechanics within the curved spaces, and explore feedback regulation loops between these factors. To model such multi-physics systems, we could possibly enrich Riemannian L-systems to endow a given topological space with several overlaying metrics corresponding to constraints imposed on growth by the different chemical/physical processes, and compute trajectories realizing some trade-off between geodesics from these different metrics. On the computational side, the currently implemented notion of manifold relies on a unique coordinate map. This restricts the possibility to deal with degenerated coordinates and with complex manifold topologies. In the current implementation, degenerated points are handled as exceptions with dedicated algorithms (see Supplementary technical documentation of Riemannian L-systems in L-Py). To overcome these limits, a complete implementation of the notion manifold, relying on collections of coordinate maps called atlases (Carroll, 2014) would be needed. Another direction of research is the development of abstract Riemannian spaces in 3-D or even 4-D (3-D + time). Such an implementation would make it possible to model growth phenomena taking place in volumes (such as the formation of vascular tissues growing stems for instance). Finally, generalisations of such combination of rewrite rules and dynamic/differential modelling of geometry could certainly be developed in other declarative (rule-based) morphodynamic systems beyond L-systems and turtle geometry, such as MGS (Giavitto & Spicher, 2008), Dynamical Grammars (Mjolsness, 2010) or Graph Grammars (Hemmerling et al., 2008).

**Open peer review.** To view the open peer review materials for this article, please visit http://doi.org/10.1017/qpb.2025.10014.

## Acknowledgements

The authors would like to thank the reviewers for their detailed feedback and their various constructive comments that helped improve the article. They would also like to acknowledge the journal editors who accepted to publish a theoretical paper of unconventional size and structure.

**Competing interests.** The authors declare none.

**Data availability statement.** The L-Py software is freely available through the conda environment at https://anaconda.org/fredboudon/openalea.lpy. The L-Py Riemannian L-system software plugin and the notebooks describing the examples displayed in this article are freely available at https://github.com/fredboudon/RiemannianLsystems.

**Author contributions.** C.G. conceived the study and implemented the first prototype of Riemannian L-systems, C.G. and F.B. designed and developed the examples, F.B. modified the L-Py API to host Riemannian L-systems, C.G. and F.B. developed the final Riemannian L-systems module integrated in L-Py, F.B. finalised the notebooks presenting the examples, C.G. and F.B. wrote this article.

**Funding statement.** The authors would like to thank their research institutes Inria and Cirad for background funding. This research received no specific grant from any funding agency, commercial or not-for-profit sectors.

**Supplementary material.** The supplementary material for this article can be found at https://doi.org/10.1017/qpb.2025.10014.

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
