## [Reviewer Report]

<b>Review of: “Riemannian L-systems: modeling growing forms in curved spaces”, QPB-2024-0029</b>

July 2024

The MS by Godin and Boudon extends the definitions of L-systems and procedural “turtle geometry” to background geometries that are not flat Euclidean spaces, but are instead modeled by Riemannian manifolds, first those embedded within ordinary flat space and then those that are not so embedded. The MS provides examples that seem relevant to modeling plant growth, as might be expected from the heritage of L-systems as plant growth models, but now extended to a more general geometric setting. The MS also has a tutorial flavor, in explaining the standard mathematics of the geometry of curved surfaces and spaces, so that for example the key concept of a covariant derivative can be used to generalize the dynamics of L-systems so that they apply equally to finite displacements in curved spaces as in ordinary flat Euclidean spaces.

The MS is both imaginative in the methods proposed and well connected to plant biology, with numerous potentially instructive examples (veined leaves, branches, stigmas, and pollen tubes are featured). The methods comprise an appealing combination of rewrite rule modeling with dynamic geometry. This combination could be obtained for other rewrite rule based spatial modeling systems besides L-systems as well, although that point is not made in the text. There is no doubt in my mind that the MS should be published.

Of course I have some suggestions, preceded by locations in the subitted MS.

Section 1, p. 1: I argue that Equation 2 is largely a red herring in the current context. The inclusion of “dt” as an argument in F is required in stochastic differential equations (SDEs) due to the presence of both drift terms (the usual RHS of an ordinary differential equation) that scale linearly in dt, summed with stochastic noise terms which scale as sqrt(dt). Both kinds of differential equations, deterministic and stochastic, can exist also on graphs and, with partial spatial derivatives, in spatial continuua such as Euclidean spaces and manifolds. But no SDEs are present in the MS, and the distinction between deterministic and stochastic dynamics thus encompassed is completely independent of the distinction between discrete and continuous spatial structures made in the MS. In this regard I think Equation 2 expresses an overly general form (general dependence on dt) of the wrong distinction. In addition, Equation 2 puzzles because like Equation 1 it contains spatial partial derivatives despite “the lack of a notion of derivative on discrete structures” in the systems it is intended to describe.

Section 1, p. 1: The phrase “Dynamical systems with dynamical structures” is a bit cumbersome. Briefer alternatives are “variable-structure systems” or “morphodynamics”.

Section 2.2, p. 5, end of Section 2.2: Regarding the “procedural” aspect of Riemann L-systems (and perhaps ordinary L-systems too): For computer graphics it doesn’t matter, but for biological modeling the procedures should be limited to computations for which one can imagine a finer-grained biological model. So, “if” statements are fine but (just for example) for-loops with integer-valued loop variables that index into arrays would be more problematic. They, and most complex procedures, would be better rendered using more L-system rules that are each constrained to be biologically interpretable. This issue could use some discussion.

Section 3.1, p.10:

One might also mention the relationship of the two principle curvatures to the eigenvalues of the Hessian of the local height function.

Section 3.1, p.10:

It is good to be reminded up front of the two kinds of 2D surface curvature, intrinsic (Gaussian) vs. embedding-dependent (mean). The covariant derivatives used later on depend on the intrinsic geometry only, as developed most fully in Section 6. But it can be a mistake to concentrate too much on just the intrinsic geometry. Biologically the embedding geometry matters too. Examples of this are the Helfrich energy function for fluid membranes which depends on both curvatures but due to the Gauss-Bonnet theorem mainly on the mean curvature; uncontrolled self-intersection in 3D of 2D surfaces specified soley by intrinsic properties; tissue invagination models such as [Odell et al. 1981, Fig. 5] which rely on raising a network of biomechanical structures (modeled e.g. as springs with nonzero resting lengths) a short distance above or below the idealized 2D surface of a tissue, thus gaining access to embedding-dependent mean curvature data; and the resulting ubiquitous biomechanical formation of roughly cylindrical tubes, which have zero intrinsic curvature and are thus invisible to intrinsic geometry alone. Another example may be the morphogenesis by “kirigami” steric constraints due to collision of leaf tissues within a bud [Couturier et al. 2011].

On the other hand with the exception of such self-intersection issues, which are important, one can possibly locally factor the 3D embedding of 2D geometry into the intrinsic 2D geometry together with one extra scalar field for the mean curvature. I think it would be good to discuss this topic a bit in order to provide some conceptual clarity on why focusing on the intrinsic geometry, and on the covariant derivative, is still important even though intrinsic geometry alone doesn’t suffice for modeling many biological systems and subsystems.

Section 3.2: Each of the manifolds parameterized in this MS seem to be parameterized by a single coordinate patch. For general manifolds this is not possible without introducing singularities, so the general theory of manifolds deals with smooth changes of coordinate system between overlapping patches. Even the sphere has coordinate singularities at the north and south poles where for example longitude becomes many-valued. How do the authors propose to deal with singular and/or multiple coordinate patches in a Riemannian L-system? This point should probably be discussed.

Section 3.3, p. 13: The first and third unnumbered equations after Equation 24 could refer back to Equation 8.

Section 3.3, p. 15: Eq. 27: note that the Gammas depend on p and q in general, fully coupling the system.

Section 3.3, p. 15: The combination of rewrite rule modeling and covariant derivative dynamics could be obtained for other rewrite rule based spatial modeling systems besides L-systems (such as the cited references [18,44]) as well, perhaps more naturally in the case of those like [44] that incorporate general differential equation dynamics explicitly.

p. 21: Don’t you have to turn again by -alpha at the end, to get H’ ?

also does it say F(l) or F(1)? Courier font makes it hard to say.

p. 27, Fig 13d; p. 30, Fig 15b: Veins should influence shape of leaf as much as vice versa, since veins are strong structural elements that tend to bend less than other parts of the leaf. How could one model this effect with the proposed system?

p. 29: “the corresponding geodesic follows naturally the ridge of the curved leaf blade”: Why “naturally”? If I draw a fan of lines emanating from a single point on a flat piece of paper, and then curl the paper into a tube imprinted with a fan of geodesics, there is no elevated probability for a geodesic to follow the main axis of the tube in particular. Is something else, having to do with Gaussian curvature, going on and if so what? Same question for p. 30, “The simulated trajectory then follows naturally the ridge along the trunk surface.”

“Interestingly, the disposition of ivy leaves is not confined to the surface of the shape….” With multiple embedding spaces it seems we are reverting from biological modeling to the L-systems’ previous culture as a tool of computer graphics. What would be the biological analog of turning off an embedding space with Endspace? It seems purely procedural - a matter of programming, rather than modeling from the local point of view of the small dumb molecules that must ultimately do all this work.

Section 6.2, p. 41, text following equation 45, and Figure 22c-d, representation of Poincare-Beltrami half-plane:

The small disks that grow as one approaches u^2 = 0 from above are visualized *exactly backwards* from actual neighborhood sizes. A small (in geodesic distance) neighborhood of each point would have formula ds^2 < epsilon^2, i.e.

(du^1)^2 + (du^2)^2 < epsilon^2 (u^2)^2, so the radii of the circular disks should get *smaller* not bigger as one approaches the u^1 axis. This is how the small disks should be drawn! The inferior alternative would be to draw disks of constant small radius using distance as defined in the parameter space, not geodesically. Then the little disks would look exactly like Figures 22a-b, which would be uninformative.

Correcting this visualization would also explain the curved-looking geodesics: the shortest path between two points at the same elevation should not be through a large number of small-appearing neighborhood disks at the same elevation, but rather should take a short-cut through just a few large-appearing (but geodesically equal-sized) neighborhood disks at a higher elevation and then come back down.

Tropisms:

I’m not convinced this isn’t just a “looks like” phenomenon. The figures are suggestive of gravitropism, phototropism, and wind-blown modification of plant morphology. But as to *why* these mechanisms should be well-modeled by curved geometry, mechanistically, I can’t find a reason. Can the authors?

<b>Smaller suggested corrections</b>

p. 9 : non-ambiguous —> unambiguous

p. 18: transport coordinates —> change coordinates

p. 21, end of first paragraph:

“turtle at the point P’ “ —> “turtle to the point P’ ” ??

p.28:

“out downward the papillae” —> “out down the papilla”

“the papillae surface” —> “the papilla surface” (two occcurences)

p. 28, Fig 14: pin-formed —> pin-shaped

Reason: avoid ambiguity with the “PIN-FORMED” auxin efflux carriers in plants.

p. 28:

“Leaves vascular networks are essential to gaz,”

—> “Leaves’ vascular networks are essential to gas,”

[two corrections here]

p. 45:

“relativistic theory of positional information”

—> “ ‘general relativistic’ theory of positional information” [or similar]

Reason: There is a tricky problem with the phrase “relativistic theory of positional information”. In physics I think “relativistic” strongly connotes “very fast”, so fast that the speed of light effects (Lorenz transformations) of special relativity come into play. Special relativity plays out in flat Minkowski space, not flat Euclidean space, so it has nothing to do with the kind of geometry studied in the MS since all metrics here are positive definite. And of course nothing in plant science is obligatorily fast on the scale of the speed of light. “General relativistic” on the other hand means roughly “very fast *or* in very strong gravity” so that it is either “relativistic” or it is in a strongly curved space (but locally Minkowski rather than locally Euclidean) or both. So the title of Ref. [27] “… a general relativistic theory of positional information” is just barely correct, but only if the word “general” is taken to modify “relativistic” and not “relativistic theory of positional information”. So the word “general” cannot be dropped from “general relativistic” in the present MS either. I further suggest scare quotes for ‘general relativistic’ because actual General Relativity requires the local flat Minkowski spaces as tangent spaces, which leaves don’t ordinarily require.

<b>References</b>

G.M. Odell, G. Oster, P. Alberch, B. Burnside

The mechanical basis of morphogenesis: I. Epithelial folding and invagination

Developmental Biology

Volume 85, Issue 2, 30 July 1981, Pages 446-462

Couturier E., Courrech du Pont S., Douady S.

The filling law: a general framework for leaf folding and its consequences on leaf shape diversity.

J. Theor. Biol. 2011; 289: 47-64

https://doi.org/10.1016/j.jtbi.2011.08.020 ; https://arxiv.org/abs/1003.4756

---

## [Reviewer Report]

I’m excited to see the work described in this paper,

and especially to see the new biological models it enables,

which is why I’m still recommending acceptance.

However, the paper is much, much too long

(45 pages in manuscript, plus references)

and spends most of that space introducing mathematical background

in an excessively formal way

(5 pages on L-systems, and 17 on differential geometry).

No biological model is even mentioned until section 4.3 on page 28!

For this reason, I can’t suggest acceptance with anything but major revisions.

I understand that some mathematical background is necessary

to motivate the the advances of the work.

However, when so much is standard textbook derivations

I wonder if it’s really needed here.

Admittedly, I am familiar with all of this background already,

and a newcomer to the area might require a more in-depth introduction.

At the same time, though, I can’t help but feel that any reader

that can (for example) follow the derivation of the geodesic equation

you present in section 3.3

wouldn’t be better served by reading about it in a textbook

on differential geometry (or, indeed, the Wikipedia page on the topic).

Section 2, the L-systems overview, covers the topic in quite a bit of detail,

starting from the introduction of D0L-systems and introducing branching,

parameters, context sensitivity, procedural rules, turtle geometry,

interpretation rules, and environmental sensing one at a time.

This recapitulation of the history of L-systems is almost entirely unnecessary.

In addition, the formalism with which the topic is introduced

(starting with string homomorphisms) seems unwarranted;

even the paper originally introducing L-Py doesn’t go to such formal detail.

Productions are important, and turtle geometry even more so,

and branches and environmental sensing are referenced in some of the examples,

but surely they could be introduced more succinctly,

with a reference to some other text on L-systems

for those interested in the formal details?

(In fact, I could find only one example L-system which uses

interpretation rules (p.22), which otherwise take up four paragraphs

in this section).

Section 3 takes up more than a third of the text

and deals with both the background in differential geometry

and how moving on parametric surfaces is performed in L-Py.

Again, the background is introduced in significant detail,

including coordinate lines, Gaussian and mean curvature,

the covariant derivative, geodesics, exponential maps,

and parallel transport.

I understand that this background is harder to compress

than that on L-systems;

encoding this mathematics in a usable way is the entire point

of the paper, after all.

However, the detail with which it is introduced seems excessive.

The section deriving the covariant derivative and the geodesic equation,

three pages in the manuscript,

seems like it could be reduced to something like

Geodesics, locally straight lines in curved space [citation of text],

follow the differential equation [geodesic equation]. Here the

Γs are so-called Christoffel symbols, which on a parametric surface

have the form [...].

and then go on to the more salient information,

namely how geodesics are implemented in L-Py by solving this equation,

as an initial-value or boundary-value problem.

Curvature is introduced over a whole page in the manuscript,

including a two-panel figure,

and with the exception of the curvature-sensing models in section 5.1

(which do not require an in-depth mathematical basis of curvature to understand)

is never brought up again.

Parallel transport is described, along with L-Py modules which implement it,

but never used in any example model at all.

At the same time, though, some of this background is clearly necessary,

at least in some detail; for instance, it’s not immediately obvious

that turning on curved surfaces has to involve transformation

through the embedding space.

(As a matter of science rather than presentation,

the Lane-Riesenfeld algorithm is described on pages 21-23.

While in flat space this algorithm produces successive approximations to

B-spline curves, it’s unclear (and you don’t prove) that the same is

true in curved space. Indeed, the sequence in figure 10l,m,n doesn’t

look like a convergence to a B-spline curve.)

The next sections cover example models.

The first few (in sections 4.1-2) seem more like elaborations

of the primitives they illustrate than models in their own right:

geodesic trajectories illustrating F,

turning and branching illustrating the turning operations,

the ‘venation’ on the leaf surface illustrating LineTo.

Section 4.3, where biological models are introduced, is mostly fine;

the growing ivy model is interesting and could probably use some sample code

to illustrate how the EndSpace module is used.

The models in Section 5 are much more interesting.

It’s not clear to me why the ?T module is needed,

versus the existing query modules like ?P or ?H,

for extracting position and heading information

specific to the manifold.

The difference between forms ‘floating on’ versus

’living at' the surface is well-observed,

but the section could use a code sample

to show how this reactive development

is specified and performed by L-Py.

I don’t understand the kidney fern leaf model, however,

or at least why it is interesting in this context.

Finally, section 6 introduces L-systems in manifolds

defined by an abstract metric, rather than on a parametric surface.

This is introduced by another two-page recapitulation of

differential geometry, much of which follows directly from

the material introduced in section 3.

Surely you could just introduce the metric tensor,

then quickly describe how all of the operations described there

rely only on the metric, not the embedding surface?

The models illustrating geodesics and embeddings on such a surface

are well-chosen.

The final models, considering tropism as development in a non-Euclidean space,

are an interesting experiment but ultimately a diversion.

You don’t show that the deformations are anything more than

suggestive of tropisms, let alone

"result[ing] from the nature of the ... abstract space in which the plant grows,

reminiscent of how gravity ... [is] interpreted as resulting from

space-time curvature in general relativity".

---

Once more, I really appreciate the advance represented by this work,

but I think the paper has too much mathematical background described

too formally.

The descriptions should be made cleaner,

and the examples either simpler or more applicable.

If you feel you can’t make these changes,

then this manuscript might also work as a submission

to a computer graphics journal.

---

## [Reviewer Report]

I am a plant modeler, but not a specialist of L-systems, and, before reading this manuscript I had only limited knowledge in Riemannian geometry. I thus have the position of an interested, but only partially educated reader.

The authors “chose not to assume that the reader is familiar with concepts in differential geometry” and “therefore introduce the necessary fundamental concepts and notations used in this domain to keep the text as self-contained as possible”. I must say that the challenge was successfully met. The explanations are progressive and clear. I learnt a lot about L-systems and how they can be implemented in Riemannian geometry. The relevance and usefulness of this approach is demonstrated through a variety of beautifully illustrated examples. I specially enjoyed the last section, which proposes modeling plant tropisms using concepts borrowed from General Relativity. This was worth a technical digression through abstract Riemannian spaces!

On the whole, this a very interesting and inspiring manuscript, combining advanced theoretical developments with a ready-to-use modeling software platform and stimulating ideas for future research. I expect it to be influential in the field of plant development modeling (and maybe even for other system than plants).

A few minor comments for the authors:

In all listings generating a von Koch flake, you are using the L-Py statements ‘+’ and ‘-’ without an angle value. If I understand correctly, this is short for ‘+(60)’ and ‘-(60)’. This should be stated explicitly. I can not find it in the L-Py documentation.

P.11: “xi and uα are often considered as the coordinates of the surface point P expressed in either U or R3 respectively”

To get the correct respective order, it should be “uα and xi”, not “xi and uα”. And in the next sentence, X and u could also be swapped.

P.20: “the module ParallelTransportReset that reiniliatizes”

⇒ reinitializes

P.22: “encapsulated within StartBSpline and EndBSpline modules 6.”

I guess you mean the reference [6[. Same in the caption of Fig 10.

Fig. 16: It looks the ivy branching system is not the same on the left and on the right.

In Section 5, you could mention another biological example, ‘curvotaxis’, i.e. the influence of curvature on cell migration. See for instance:

* Pieuchot, L., Marteau, J., Guignandon, A. et al. Curvotaxis directs cell migration through cell-scale curvature landscapes. Nat Commun 9, 3995 (2018). https://doi.org/10.1038/s41467-018-06494-6

* M. Werner, A. Petersen, N. A. Kurniawan, C. V. C. Bouten, Cell-Perceived Substrate Curvature Dynamically Coordinates the Direction, Speed, and Persistence of Stromal Cell Migration. Adv. Biosys. 2019, 3, 1900080. https://doi.org/10.1002/adbi.201900080

P.44: I believe that “Movie #2” should be replaced with “Movie #5.1”. By the way, Movie #5.2 and Movie #5.3 are never mentioned.

---

## [Editor Report]

Dear Christophe and Frederic, 

Please accept my sincere apologies for the time it has taken me to secure appropriate reviewers for your manuscript. 

I consider your manuscript to be of excellent quality and my enthusiasm is shared by all reviewers. There are, however, several points raised by the reviewers that I would encourage you to consider. Whilst the current manuscript is clearly (and successfully) aimed at being accessible and didactical, this results in a rather long article that takes many pages to get to what many readers will be most interested in (actual models). Perhaps based on the reviewers suggestions there are elements you can consider shortening or cutting without losing the current excellent accessibility. I will, however, leave this to your judgement. It is a recommendation, not a requirement.

Please look also at the other points raised by the reviewers and address these as best you can. There are some important points here that warrant (minor) changes. 

Thank you for submitting such an imaginative and important contribution to QPB! I look forward to seeing your manuscript in press. 

With best wishes

Richard

---

## [Reviewer Report]

All my comments have been taken into account. The ‘general-relativistic’ approach to tropism modeling is now better discussed. I recommend this revised manuscript for publication.

---

## [Reviewer Report]

Riemannian L-systems: modeling growing forms in curved spaces”, QPB-2024-0029

Re-review

The revision is acceptable. I provide below some replies to replies which may suggest a few remaining minor points to correct, optionally, in the paper or in the software documentation that the paper refers to.

Section 3.1: Couldn’t find strings “embed” or “getNormalAt”in the documentation.md site using its search tool. Instead the documentation “has turtle.space.normal(u,v)” which takes the same parameters.

Section 3.2: Couldn’t find strings “degen” or “singular” in the documentataion.md site using its search tool. Couldn’t find it manually by other names.

p. 29: “ ‘the corresponding geodesic follows naturally the ridge of the curved leaf blade’: Why ‘naturally’? If I draw a fan of lines emanating from a single point on a flat piece of paper, and then curl the paper into a tube imprinted with a fan of geodesics, there is no elevated probability for a geodesic to follow the main axis of the tube in particular. Is something else, having to do with Gaussian curvature, going on and if so what? Same question for p. 30, ‘The simulated trajectory then follows naturally the ridge along the trunk surface.’

The Reviewer is right. There is no general property of curved surfaces that we are aware of that would explain this behavior.

→ We simply removed the word naturally to avoid any confusion.”:

The p. 30 “naturally” was removed, but the p. 29 one was missed. It still says “the corresponding geodesic follows naturally the ridge of the curved leaf blade”.

Section 6.2 on the half plane: The “dual” representation makes little sense to me, but OK. I think one would prefer the disk radius to correspond to something *locally measurable* e.g. by a turtle, like painting the set of all places it could get to, starting from home and proceeding at a fixed speed for a fixed short time. That’s operationally defined. (Which is why this visualization method works in Escher Poincare disk artwork.) Representing an “amount of coordinates” doesn’t seem to be locally, operationally, measurable because it depends on the coordinate system chosen. But the new text mentioning the alternative is OK.

[Parenthetically: I could not follow the derivation the authors offer in the rebuttal text. A better explanation of the “dual” alternative might be:

a = arclength in u,v parameter space

da = sqrt(du^2 + dv^2)

s = arclength in actual, Reimannian geometry

ds = sqrt(du^2 + dv^2)/v

path length = \int_{path} ds = \int (ds/da) da = \int (1/v) da

so magnification factor along a path = 1/v .

That magnification factor is what the authors try to visualize as a varying circle radius, sort of like visualizing city populations as smaller or larger disks on a geographic map.

Then if you imagine the (u,v) coordinate system is also an embedding space, one is trying to calculate a geodesic distance as a weighted sum of parametric distances with weight ds/da = 1/v. For an embedding at least parametric distances da correspond to something real: distance in an embedding space. What is weird about this visualization, though, is that the magnification factor has the wrong units to be plotted on the (u,v) parametric plane at all. Its units are length_Reimann/length_parametric, so the numerator is the wrong kind of length to plot as a disk radius (dimension length_parametric) on the (u,v) plane. But confusingly, it is still related to distance measures.

The more physical visualization I prefer is to approximate the integral along the path as a sum of small ds = epsilon line segments, which do not overlap one another, and just count up the number of line segments, i.e. non-overlapping small disks traversed, times epsilon. So these kinds of disks do make sense on the (u,v) plane - not surprising since their radii are proportional to v.

However, this aesthetic disagreement doesn’t affect the validity of the paper.]

Futher improvements look good.

---

## [Editor Report]

Dear Christophe and Frederic, 

Thank you for your excellent revisions. I am in agreement with the reviewers and am very happy to accept your manuscript. 

Thank you for choosing QPB! 

There are several minor points raised in one of the reviews that I would recommend you consider and, if you agree, address. But I leave this to your judgement. 

With best wishes

Richard